# Private learning implies quantum stability

**Srinivasan Arunachalam**
IBM Quantum, IBM T.J. Watson Research Center, Yorktown Heights, USA
`Srinivasan.Arunachalam@ibm.com`

**Yihui Quek**
Information Systems Laboratory, Stanford University, USA *
`yquek@stanford.edu`

**John Smolin**
IBM Quantum, IBM T.J. Watson Research Center, Yorktown Heights, USA
`Smolin@us.ibm.com`

## Abstract

Learning an unknown $n$-qubit quantum state $\rho$ is a fundamental challenge in quantum computing. Information-theoretically, it is known that tomography requires exponential in $n$ many copies of $\rho$ to estimate its entries. Motivated by learning theory, Aaronson et al. introduced many (weaker) learning models: the PAC model of learning states (Proc. of Royal Society A'07), shadow tomography (STOC'18) for learning "shadows" of a state, a model that also requires learners to be differentially private (STOC'19) and the online model of learning states (NeurIPS'18). In these models it was shown that $\rho$ can be learned "approximately" using *linear* in $n$ many copies of $\rho$. But is there any relationship between these models? In this paper we prove a sequence of (information-theoretic) implications from differentially-private PAC learning to online learning and then to quantum stability. Our main result generalizes the recent work of Bun, Livni and Moran (Journal of the ACM'21) who showed that finite Littlestone dimension (of Boolean-valued concept classes) implies PAC learnability in the (approximate) differentially private (DP) setting. We first extend their work to the real-valued setting, and further extend to the setting of learning quantum states. Key to our results is our generic quantum online learner, Robust Standard Optimal Algorithm (RSOA), which is robust to adversarial imprecision. We then show information-theoretic equivalences between DP learning quantum states in the PAC model, learnability of quantum states in the one-way communication model, online learning of quantum states, quantum stability, various combinatorial parameters and give further applications to gentle shadow tomography and noisy quantum state learning.

## 1 Introduction

Learning an unknown quantum state $\rho$, given copies of the state is a fundamental task in quantum computing, often referred to as *tomography*. Tomography is of great practical interest since it helps in tasks such as verifying entanglement, understanding correlations, and is useful for calibrating, understanding and controlling noise in quantum devices. In the last few years, questions about the fundamental limits of this task have gained a lot of theoretical attention: how many copies of an $n$-qubit quantum state $\rho$ are required to estimate the density matrix of $\rho$ up to small error? Recent

---

*https://stanford.edu/~yquek

35th Conference on Neural Information Processing Systems (NeurIPS 2021).

breakthrough results of [1, 2, 3] showed that $\Theta(2^{2n}/\varepsilon^2)$ copies of $\rho$ are *necessary and sufficient* to learn $\rho$ up to trace distance $\varepsilon$. Unfortunately, this exponential scaling in complexity hampers practical applications of tomography; the best known experimental implementation of full-state quantum tomography has been for a 10-qubit quantum state [4].

Simultaneously, the use of *machine learning* for quantum states has gained lot of traction in quantum computing. In this area, the goal is to use heuristics to learn an unknown quantum state, often with the motivation of physical implementation. Provable results in this area are very few, but interest is widespread due to the number of emerging quantum hardware devices. Given the fundamental importance of tomography and the *natural* connection to learning, Aaronson [5] formally defined a variant of tomography: *PAC learning quantum states*. This problem brings together two ripe exciting topics: classical machine learning and quantum computing. Following his work, there have been a few other papers that have introduced models of learning quantum states with various constraints.

In this work, our main contribution is to show implications between several seemingly different quantum learning models. The main technicality is extending known results in *classical* learning theory from Boolean functions to real-valued functions with noisy labels – a setting motivated by learning quantum states, but with potential extensions to learning other physical systems (e.g. [6]). Along the way we also show how certain implications that hold for classical Boolean functions do *not* hold true in the real-valued setting (which could be of independent classical interest). See Figure 1 for a summary of our results.

## 1.1 Background: Models of interest

To explain our main results, we start by introducing some learning models of interest. We state them in the language of quantum state learning. However, these are at their core, classical learning models for real-valued functions with imprecise feedback, and the translation is stated precisely in Section 2.2. We also formally translate them to the classical learning setting in Supplementary material (Section A), which we omit here due to space constraints.

**PAC learning.** Valiant's Probably Approximately Correct (PAC) learning, lays the foundation for computational learning theory. Aaronson [5] considered learning quantum states in the PAC model. In this model, let $\rho \in \mathcal{C}$ be an unknown quantum state (picked from a *known concept class* $\mathcal{C}$ of states) and let $D : \mathcal{E} \to [0,1]$ be an *arbitrary* unknown distribution over all possible 2-outcome measurements $E$. Suppose a quantum learner obtains *training examples* $(E_i, \mathsf{Tr}(\rho E_i))$ where $E_i$ is drawn from $D$, and the goal is to output $\sigma$ such that with probability $\geq 0.99$, $\sigma$ satisfies $\Pr_{E \sim D}[|\mathsf{Tr}(\sigma E) - \mathsf{Tr}(\rho E)| \leq \zeta] \geq 1 - \alpha$. How many training examples suffice for such a $(\zeta, \alpha)$-PAC learner? In answer, he showed that the number of examples necessary and sufficient to learn $\mathcal{C}$ is captured by the *fat-shattering dimension* of $\mathcal{C}$.

**PAC learning with Differential privacy.** A well-studied area of computer science is differential privacy (DP) (which says that an algorithm should behave "approximately" the same given two datasets that differ in one element). This notion can be extended to the quantum realm, where we ask that the quantum PAC learner proposed above is also *differentially private*, wherein given two datasets $S = \{(E_i, \mathsf{Tr}(\rho E_i))\}_i$, $S' = \{(E_i', \mathsf{Tr}(\rho E_i'))\}_i$ such that there exists a unique $i$ such that $E_i \neq E_i'$, then a quantum $(\gamma, \delta)$-DP PAC learning algorithm needs to satisfy $\Pr[\mathcal{A}(S) = \sigma] \leq e^\gamma \Pr[\mathcal{A}(S') = \sigma] + \delta$, where $\mathcal{A}(S)$ is the output of $\mathcal{A}$ on input $S$. [2]

**Communication complexity.** Consider the standard one-way communication model between Alice and Bob. Suppose Alice has a quantum state $\rho$ (unknown to Bob) and Bob has an unknown (to Alice) measurement $E$. The goal of Bob is to output an approximation of $\mathsf{Tr}(\rho E)$ if only Alice is allowed to communicate to Bob. A trivial strategy for this communication task is for Alice to send a classical description of $\rho$, but can we do better? If so, how many bits of communication suffice for this task?

**Online learning.** Several features of the PAC learning model and tomography are somewhat artificial: first, the assumption that the measurements (training examples) are drawn from the same unknown distribution $D$ that the learner will be evaluated on, which does not account for adversarial or changing environments, and secondly, it may be infeasible to possess $T$-fold tensor copies of the unknown quantum state $\rho$, rather we may only be able to obtain sequential copies of it. The quantum

---

[2]Our notion of DP differs from the notion of DP proposed by [7]. They consider DP *measurements* with respect to a class of *product states*, whereas here we require DP with respect to the dataset $\{(E_i, \mathsf{Tr}(E_i\rho)\}_i$.

online learning model addresses these aspects. Online learning consists of repeating the following rounds of interaction: a learner obtains a copy of $\rho$, maintains a local $\sigma$ which is its guess of $\rho$, obtains a description of measurement operator $E_i$ (possibly adversarially) and predicts the value of $y_i = \text{Tr}(\rho E_i)$. Subsequently it receives as feedback an $\varepsilon$-approximation of $y_i$. On every round, if the learner's prediction satisfies $|\text{Tr}(\sigma E_i) - y_i| \leq \varepsilon$ then it is *correct*, otherwise it has made a *mistake*. The goal of the learner is the following: minimize $m$ so that after making $m$ mistakes (not necessarily consecutively), it makes a correct prediction on *all* future rounds.

Importantly, while the goal in the above model is to make real-valued predictions $y_i$, it departs from the real-valued online learning literature in allowing for $\varepsilon$-imprecision in the feedback. This imprecision is inherent to all learning settings where the feedback is generated by a statistical algorithm (in our setting, the feedback arises from processing the outcomes of quantum measurements), and this generalization has non-trivial implications, as we show. Working in this model, Aaronson [8] showed that for learning the class of all quantum states, it suffices to let $m$ be at most *sequential fat-shattering dimension* of $\mathcal{C}$ (a combinatorial parameter which was originally introduced by Rakhlin et al. [9]).

**Motivating question.** All these learning models can be seen as variants of full-state tomography, and are known to require *exponentially* fewer resources than tomography. A natural question is:

*Is there a relation between these learning models, communication and combinatorial parameters?*

Understanding this question classically in the context of Boolean functions has received tremendous attention in computational learning theory and theoretical computer science in the last two years. There have been several papers establishing various connections [10, 11, 12, 13, 14, 15, 16, 17, 18], just to cite a few. However, these papers leave two important questions open:

1. *Do these results apply to learning quantum states?*
2. *Do the classical results (for Boolean functions) also hold for real-valued functions?*

Ours is the first work that studies the first question. In the process, we not only explore how implications between Boolean learning models translate over to the quantum setting, but we also introduce new notions of (classically) privately learning quantum states and quantum stability (linked to privacy), contributing another perspective to an ongoing discussion of what differentially-private quantum computation could look like [19, 20, 7]. The second question indirectly received attention recently in NeurIPS'20 by Jung et al. [11] for *multi-class* learning, but their work left several open ends (when translated to real-valued learning) which we tie up here (see Section 3.4 for a more in-depth comparison).

## 2 Overview of main results

### 2.1 Quantum results

To condense our (affirmative) answer to the first question above, we derive a series of implications going through all these models, starting from differentially-private PAC learning to online learning to quantum stability (our conceptual contribution which we define and discuss below).

Taking a step back, quantum online learning and DP PAC quantum learning seem very different on the surface. Online learning ensures that *eventually* (after $m$ mistakes) we learn the state approximately, but makes no guarantees on when the last of $m$ mistakes occurs with respect to the series of examples seen. DP PAC learning is *not* online – it separates the learning into train (offline) and test (online) phases, and also introduces a distribution, $D$, from which measurements are drawn. Ultimately DP PAC learning says that after seeing $T$ measurements from $D$, we have (privately) learned the state. We show that in fact DP PAC learning sample complexity can be lower bounded by the *sequential fat-shattering dimension* which also characterizes the complexity of online learning [9]. We give a summary of our results in Figure 1. Also, we say an algorithm is *pure* DP (resp. *approximate* DP) when $\delta = 0$ in our definition (resp. $\delta > 0$). We remark that only a few arrows are efficient in both sample and time complexity, otherwise these implications are primarily information-theoretic.

**Conceptual contribution.** The main center piece in establishing these connections is the concept of *quantum stability*, which is the new conceptual contribution in this work. Intuitively, we say a quantum learning algorithm is stable if, for an unknown state $\rho$, given a set of noisy labelled

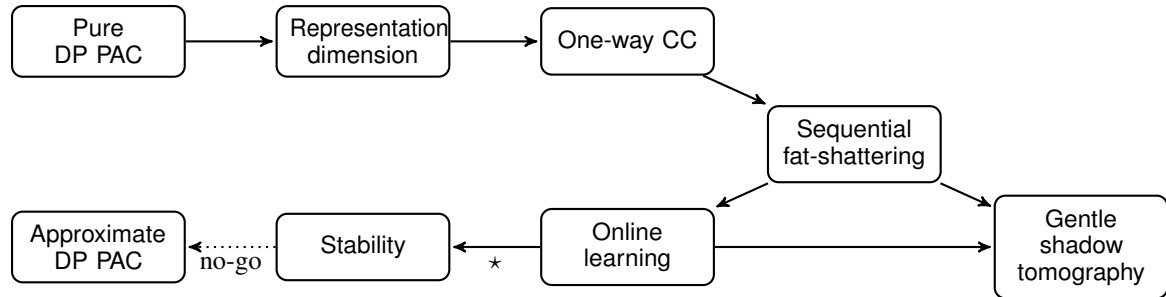

Figure 1: Summary of results for learning real-valued concept classes and quantum states with imprecise feedback. Except for the $\star$-arrow, an arrow $\mathsf{A} \to \mathsf{B}$ implies that, if the sample complexity of learning in model $\mathsf{A}$ or the combinatorial parameter $\mathsf{A}$ is $S_\mathsf{A}$, then the complexity of learning in model $\mathsf{B}$ or the combinatorial parameter $\mathsf{B}$ is $S_\mathsf{B} = \mathrm{poly}(S_\mathsf{A})$. The dotted arrow signifies that a technique used to prove that arrow for Boolean functions is a no-go for our quantum learning setting.

examples from a distribution $D$, there exists one state $\sigma$ such that, with "high" probability, the output of the learning algorithm is "close" to $\sigma$. More formally, we say a quantum learning algorithm $\mathcal{A}$ is $(T, \varepsilon, \eta)$-*stable* with respect to distribution $D$ over a set of orthogonal 2-outcome measurements if, given $T$ many labelled examples $S$ consisting of $E_i$ drawn from $D$ and $\zeta$-approximations of $\mathsf{Tr}(\rho E_i)$, there exists a state $\sigma$ such that

$$\Pr[\mathcal{A}(S) \in \mathcal{B}_\mathcal{M}(\varepsilon, \sigma)] \geq \eta, \tag{1}$$

where the probability is taken over the examples in $S$ and $\mathcal{B}_\mathcal{M}(\varepsilon, \sigma)$ is the ball of states $\varepsilon$-close to $\sigma$ with respect to $\mathcal{M}$, i.e., $\mathcal{B}_\mathcal{M}(\varepsilon, \sigma) = \{\sigma' : |\mathsf{Tr}(E\sigma) - \mathsf{Tr}(E\sigma')| < \varepsilon \text{ for every } E \in \mathcal{M}\}$. In other words, quantum stability means that up to an $\varepsilon$-distance, there is some hypothesis state $\sigma$ that is output by $\mathcal{A}$ with "high" (at least $\eta$) probability.

A classical analog of stability can be written as: a classical learning algorithm for concept class $\mathcal{C} : \mathcal{X} \to [0,1]$ (run on examples $\{(x_i, \hat{c}(x_i))\}$ where $|\hat{c}(x_i) - c(x_i)| < \zeta, c \in \mathcal{C}$) is *stable* if there exists a hypothesis $f$ such that

$$\Pr[\mathcal{A}(S) \in \mathcal{T}(\varepsilon, f)] \geq \eta,$$

where $\mathcal{T}(\varepsilon, f) := \{g \in \mathcal{C} : |g(x) - f(x)| < \varepsilon \text{ for every } x \in \mathcal{X}\}$.

*Consistent stability implies learnability.* While we will make this precise later, the significance of an algorithm $\mathcal{A}$ being *stable* is that $\sigma$, the output state at the 'center of the ball', is a good hypothesis for estimating measurement probabilities (and hence $\mathcal{A}$ is a good learner). This is not at all obvious from the definition of stability, which does not inherently require that this $\sigma$ is a good approximation of $\rho$. Yet, it turns out that if $\mathcal{A}$ is a stable and consistent learner (i.e., its output does not contradict any of the training examples it has seen), $\sigma$ has low loss with respect to $D$. This means that, using hypothesis $\sigma$ to predict outcomes of future measurements drawn from distribution $D$ as $\mathsf{Tr}(E\sigma)$ will yield $\varepsilon$-accurate predictions with high probability.

## 2.2 Classical results

In order to see the connection to classical learning theory we first observe the following. Learning $n$-qubit quantum states over an orthogonal basis of $n$-qubit quantum measurements, $\mathcal{M}$, is *equivalent* to learning – with imprecise adversarial feedback – an arbitrary real-valued function in the class $\mathcal{D} = \{f : \mathcal{X} \to [0,1]\}$ when $\mathcal{X} = \mathcal{M}$: there is a one-to-one mapping between the set of all quantum states and the set of bounded real-valued functions on $\mathcal{M}$.[3] To elaborate, for every $\sigma$, one can clearly associate a function $f_\sigma : \mathcal{M} \to [0,1]$ defined as $f_\sigma(M) = \mathsf{Tr}(M\sigma)$ and for the converse direction, given an arbitrary $c : \mathcal{M} \to [0,1]$, one can find a density matrix $\sigma$ for which $c(M) = \mathsf{Tr}(M\sigma)$ for all $M \in \mathcal{M}$ (and this uses the fact that $\mathcal{M}$ is an orthogonal measurement basis crucially). Hence, if one can learn $\mathcal{D}$ when we fix the $\mathcal{M}$ to be an arbitrary orthogonal basis of 2-outcome measurements then one can learn the class of quantum states $\mathcal{C}$, and the converse is also true.

---

[3]Indeed, in Section 4 of this paper, we slightly abuse notation; when $\mathcal{C}$ is a class of quantum states, we use $\mathsf{sfat}(\mathcal{C})$ to mean $\mathsf{sfat}(\mathcal{D})$.

So, our results for quantum learning naturally follow from our study of learning from noisy examples real-valued concept classes over arbitrary domain $\mathcal{X}$, which are the focus of this work. As we mentioned in the introduction, some of the implications we study are well-known for the case of Boolean functions. However, motivated by the setting of learning quantum states, we were confronted with two questions: (i) do these Boolean learning results also hold for real-valued learning? (ii) in the real-valued class learning problem, the learner is not provided with exact data, but is given *imprecise* feedback which could correspond to a different class label altogether, so are these results robust?

Our main contribution is thus in establishing all the arrows in Figure 1 for real-valued concept classes. As far as we are aware, even *classically* establishing equivalences between online learning, stability and approximate differential privacy for *real-valued* functions with *imprecise* feedback was not explored before. In the precise feedback model, a few of the arrows in Figure 1 were proven implicitly in [11], when they considered multi-class learnability but they left several open ends which we tie up here (see Section 3.4 for more). Strikingly, when considering real-valued functions, a technique due to [10] showing stability implies approximate DP PAC for Boolean functions, is a no-go in our setting, as we show. We give a counterexample concept class which can be reduced to fingerprinting codes, for which we show that stability does not imply approximate DP PAC without a domain size dependence.

# 3 Proof sketches for our main results (Fig. 1)

We break down the proofs of the arrows in Figure 1 into four steps and discuss them below. Before this, we provide a quick introduction to necessary quantum information theory.

**Quantum information preliminaries.** An $n$-qubit quantum state $\rho$ is a positive semidefinite matrix with trace 1. An arbitrary valid quantum operation on quantum states can be expressed as a *unitary matrix* $U$ (which satisfies $UU^* = U^*U = \mathbf{1}$). An application of a unitary $U$ to the state $\rho$ results in the quantum state $U\rho U^*$. Finally, in order to extract meaningful *classical* information out of a quantum state, one can perform a POVM (positive-operator valued measurement) which is specified by a set of $m$ positive semidefinite matrices $E_1, \ldots, E_m$ satisfying $\sum_i E_i = \mathbf{1}$. A measurement on a state $\rho$ using such a POVM returns a classical outcome $i \in [m]$ with probability $\mathsf{Tr}(E_i\rho)$.

## 3.1 Pure DP PAC learnability implies finite sfat dimension

It is known that if there is a DP PAC learning algorithm for a Boolean function class $\mathcal{C}$ then the *representation dimension* of the class is bounded by the sample complexity of said algorithm. Representation dimension, itself, upper bounds classical communication complexity as well as a combinatorial dimension of the concept class known as the *Littlestone dimension*.

All of the above connections and combinatorial parameters pertain to learning Boolean functions, but we show that they can be robustly ported to our 'quantum-inspired' DP PAC setting of learning real-valued functions and with adversarial imprecision. Analogous to the Littlestone dimension, the $\zeta$-*sequential fat shattering dimension* [9] (denoted $\mathsf{sfat}_\zeta(\cdot)$) is a combinatorial parameter that can be associated to every real-valued concept class. In this direction, a key contribution is: prior to our work, [21] showed that Littlestone dimension lower bounds one-way *classical* communication complexity, but we show that for learning real-valued functions, the sfat dimension lower bounds both *classical* and *quantum* communication complexity. We omit the proofs of these implications and refer the reader to Section D in Supplementary material.

## 3.2 Finite sfat($\cdot$) implies online learnability and resolving an open question of Aaronson

In the second step, the goal is to go from a concept class $\mathcal{C}$ having finite $\mathsf{sfat}(\mathcal{C})$ to design an online learning algorithm for $\mathcal{C}$ that makes at most $\mathsf{sfat}(\mathcal{C})$ mistakes even in the presence of imprecise feedback. In this direction, one of our technical contributions is to construct a *robust* standard optimal algorithm (denoted RSOA) which satisfies this mistake-bound. This RSOA algorithm and result below will be crucial for the following steps.

**Result 3.1** (Informal). *Let $\mathcal{C}$ be a concept class with $\mathsf{sfat}_\zeta(\mathcal{C}) = d$. There is an explicit robust standard optimal algorithm RSOA that makes at most $d$ mistakes in online learning $\mathcal{C}$.*

Indeed, the word 'robust' in the name of the algorithm indicates that it is robust to $\zeta$-imprecise adversarial feedback. This robustness property allows RSOA to be relevant in contexts where the feedback is generated by some physical measurement process. In this case, our end goal is to use RSOA for learning quantum states, where typically, the feedback is generated by the learner itself by measuring $E$ repeatedly on copies of the quantum state $\rho$. Averaging these measurements will provide a $\zeta$-approximation of $\mathsf{Tr}(\rho E)$ instead of its exact value.

To accommodate this imprecision, we introduce the notion of an *interleaved $\zeta$-cover* of the $[0, 1]$ interval, $\tilde{\mathscr{I}}_\zeta$, as the set of overlapping half-open intervals ('super-bins') of width $2\zeta$ given by $\big\{[0, 2\zeta), [\zeta, 3\zeta), \ldots, [1 - 2\zeta, 1]\big\}$ with the midpoints $\tilde{\mathscr{I}}_\zeta = \{\zeta, 2\zeta, \ldots, 1 - \zeta\}$ where $|\tilde{\mathscr{I}}_\zeta| = 1/\zeta - 1$. We also define the following: given a set of functions $V \subseteq \{f : \mathcal{X} \to [0, 1]\}$, $r \in \tilde{\mathscr{I}}_{2\zeta}$ and $x \in \mathcal{X}$, define a (possibly empty) subset $V(r, x) \subseteq V$ as $V(r, x) = \big\{f \in V : f(x) \in [r - 2\zeta, r + 2\zeta]\big\}$, i.e., $V(r, x)$ are the set of functions $f \in V$ such that $f(x)$ is within a $2\zeta$-ball around $r$.

We present our RSOA algorithm in Algorithm 1. We now make a few comments regarding this result. Classically, for the Boolean setting, it is well-known that the so-called Standard Optimal Algorithm is an online learner for *any* concept class, that makes at most Littlestone dimension-many mistakes [22]. Eventually, [9] generalized the work of Littlestone for real-valued functions, showing that real-valued concept classes can be learned using their FAT-SOA algorithm, with at most $\mathsf{sfat}(\mathcal{C})$ many mistakes. Now, our RSOA algorithm generalizes this, showing that real-valued concept classes can be learned with $\mathsf{sfat}_\zeta(\mathcal{C})$ many mistakes, even in the presence of adversarial imprecision of magnitude $\zeta$.

---

**Algorithm 1** Robust Standard Optimal Algorithm

---

**Input:** Concept class $\mathcal{C} \subseteq \{f : \mathcal{X} \to [0, 1]\}$, target (unknown) concept $c \in \mathcal{C}$, and $\zeta \in [0, 1]$.

**Initialize**: $V_1 \leftarrow \mathcal{C}$

1: **for** $t = 1, \ldots, T$ **do**
2:     The learner receives $x_t$ and maintains set $V_t$, a set of "surviving functions".
3:     For every super-bin midpoint $r \in \tilde{\mathscr{I}}_{2\zeta}$, compute the set of functions $V_t(r, x_t)$.
4:     The learner finds the super-bin which achieves the maximum $\mathsf{sfat}(\cdot)$ dimension

$$R_t(x_t) := \left\{ \arg\max_{r \in \tilde{\mathscr{I}}_{2\zeta}} \mathsf{sfat}_{2\zeta}\left(V_t(r, x_t)\right) \in \tilde{\mathscr{I}}_{2\zeta} \right\}$$

5:     Learner computes the mean of the set $R_t(x_t)$, i.e., $\hat{y}_t := \frac{1}{|R_t(x_t)|} \sum_{r \in R_t(x_t)} r$.
6:     The learner outputs $\hat{y}_t$ and receives feedback $\widehat{c}(x_t)$.
7:     Update $V_{t+1} \leftarrow \{g \in V_t \mid |g(x_t) - \widehat{c}(x_t)| \le \zeta\}$
8: **end for**

**Outputs:** The intermediate predictions $\hat{y}_t$ for $t \in [T]$ and a hypothesis $f(x) := R_{T+1}(x)$.

---

The basic principle that underlies our RSOA algorithm is, keep track of a set of 'surviving' functions, and after every round of learning, eliminate those which were grossly inconsistent with the adversary's feedback. In more detail: beforehand, the learner discretizes the function range $[0, 1]$ into $1/\zeta - 1$-many $2\zeta$-sized overlapping bins. During learning, upon receiving the domain point $x$, the learner evaluates all remaining functions at $x$ and 'counts' (using $\mathsf{sfat}(\cdot)$ dimension as a proxy) the number of functions mapping to each bin, and finally outputs the bin with the highest $\mathsf{sfat}(\cdot)$ dimension. The intuition is that quantifying the number of functions in each bin with the $\mathsf{sfat}(\cdot)$ dimension (of only the functions in the bin) instead of their raw count, allows the learner to reduce the surviving set of functions efficiently. Furthermore, the learner never makes more than $\mathsf{sfat}(\mathcal{C})$ prediction mistakes in the course of learning. For a proof of correctness, refer to Section B in the supplementary material.

**Resolving an open question.** Prior to our work, Aaronson [8] considered quantum online learning and showed that $\mathsf{sfat}(\cdot)$ of all $n$-qubit quantum states is at most $n$, which in particular shows the *existence* of a quantum online learning algorithm for the class of all quantum states that makes at most $n$ mistakes. However, their focus was on online learning with *regret* bounds, and so they didnn't provide an explicit algorithm that achieves the $\mathsf{sfat}(\cdot)$ mistake bound, but raised this question in their work. Our Result 3.1 resolves their question, by showing that our RSOA is such an algorithm.

## 3.3 Online learnability implies stability

We now show that if a set of quantum states has sfat dimension $d$ (i.e., it can be online-learned with noisy measurements with $d$-many mistakes), then it can be learned by an $(T = \varepsilon^{-d}, \varepsilon, \varepsilon^d)$ quantum-stable algorithm $\mathcal{A}$ satisfying Eq. (1)). That is, for unknown state $\rho$ from this online-learnable set, given $O(\varepsilon^{-d})$-many examples which consist of 2-outcome measurements $E_i$ from an orthogonal set $\mathcal{M}$ and $\varepsilon$-accurate $\mathsf{Tr}(\rho E_i)$, there exists a $\sigma$ having low loss, such that $\Pr_{S \sim D^T}[\mathcal{A}(S) \in \mathcal{B}_{\mathcal{M}}(\varepsilon, \sigma)] \geq \varepsilon^d$. We now state this formally in terms of real-valued learning.

**Result 3.2.** *Let $\alpha, \zeta \in [0, 1]$. Let $\mathcal{C} \subseteq \{f : \mathcal{X} \to [0, 1]\}$ be a concept class with $\mathsf{sfat}_{2\zeta}(\mathcal{C}) = d$. Let $D : \mathcal{X} \to [0, 1]$ be a distribution and let $S = \{(x_i, \widehat{c}(x_i))\}$ be a set of $T = \zeta^{-d}/\varepsilon$-many examples where $x_i \sim D$ and $|\widehat{c}(x_i) - c(x_i)| < \zeta$ where $c \in \mathcal{C}$ is an unknown concept. There is a $(T, \zeta^{-d}, \zeta)$-stable learner $\mathcal{G}$, that outputs $g \in \mathcal{B}(f, \zeta)$ such that $\Pr_x[||f(x) - c(x)| \leq \zeta] \geq 1 - \alpha$.*

**Proof sketch.** To prove this theorem we borrow the high-level idea from [10] (for the case of Boolean functions). Before our work we didn't have an RSOA algorithm which could be used as a black-box in order to emulate the proof-technique of [10] for the quantum/robust real-valued setting. Our stable learner in the result above is essentially the RSOA algorithm that we designed earlier, but run on a carefully tailored input distribution over the examples, with $T$ being the overall sample complexity of our algorithm. Most of the work in the proof arises in explaining how to tailor the set of examples drawn from the original distribution $D$ into a new set $S$ on which RSOA is guaranteed to succeed.

We prove this theorem in two steps: in step (1) we provide a tailoring algorithm that *defines* distributions $\mathsf{ext}(D, 1), \ldots, \mathsf{ext}(D, d)$ (where $\mathsf{sfat}_\zeta(\mathcal{C}) = d$) as a function of the unknown target distribution $D$ (to which we have black-box access). Just as in [10], the key idea for the tailoring is to inject examples into the sample that would force RSOA to make mistakes (we give more details about this below). We adapt this idea for the robust, real-valued setting. Unfortunately, this tailoring algorithm uses an unbounded number of examples (in the worst case). To handle this, step (2) is to compute the expected number of examples drawn by the tailoring algorithm, then use Markov's inequality to compute what the cutoff should be. This will be our final stable algorithm.

We now give more details about step (1). We sample many labelled examples from the unknown distribution $D$ and instead of feeding these examples directly to the black-box RSOA, we plant amongst them some "mistake examples" before giving the processed sample to RSOA. A "mistake example" is an example which is correctly labelled, but on which RSOA would make the wrong prediction. That is to say, from a large pool of $T = \zeta^{-d}$ examples drawn from $D$, craft a short sequence of $O(1/\zeta)$ examples that include at most $d$ mistake examples; now feed the short sequence into RSOA. This works, because RSOA satisfies the guarantee that after making $d = \mathsf{sfat}_\zeta(\mathcal{C})$ mistakes, it would have identified the target concept up to $O(\zeta)$ prediction error.

Our technique for creating mistake examples differs from that of [10]. In the Boolean case, to insert a mistake, it suffices to do the following: Let $c : \mathcal{X} \to \{0, 1\}$ be the unknown target function. First, take two candidate samples $S_1, S_2$, feed them into two parallel runs of SOA, and obtain two different output hypothesis functions $f_1, f_2$ respectively. Next, identify a point in the domain $x$ at which $f_1(x) \neq f_2(x)$. Say $f_1(x) = c(x)$ and $f_2(x) = \overline{c(x)}$, then it suffices to append an example (which will be the mistake example) of the form $(x, c(x))$ to $S_2$, so that when SOA is now run on $S_2 \circ (x, c(x))$, SOA's hypothesis function just before seeing the last example is $f_2$, which it then uses to make a (wrong) prediction on the last example. To generate the bit $c(x)$ for their mistake example, [10] simply flip a coin $b \in \{0, 1\}$ and with probability $1/2$, $b = c(x)$. For us this does not work because our target function is real-valued, i.e. $c(x) \in [0, 1]$. Instead, we discretize $[0, 1]$ into $1/\zeta$ many $\zeta$-intervals, pick a uniformly random interval and let $b$ be the center of this interval. Clearly now, with probability $1/\zeta$, $c(x)$ lies in the $\zeta$-ball around $b$ and is thus valid adversarial feedback. Overall, the construction of our quantum stable learner is more involved in order to work around the looser approximation guarantees of RSOA due to the possibility of imprecise adversarial feedback. We skip the proofs here and refer the reader to Section C in Supplementary material.

## 3.4 Stability does not imply approximate DP PAC learnability

So far we showed that quantum/robust real-valued online learning $\mathcal{C}$ implies a quantum/robustly stable learner for $\mathcal{C}$. For Boolean-valued $\mathcal{C}$s, [10] went one step further and created a *approximately differentially-private* learner from a stable learner; in this sense, stability can be viewed as an

intermediate property between online learnability and differential privacy. A natural question here is, can we extend this result to our setting, i.e., does quantum/robust stability in turn imply differential privacy? If so, then Figure 1 would start and end with differential privacy (albeit starting from pure DP and resulting in approximate DP).

Alas, we show that the technique of [10] does not go through for real-valued learning. More precisely, one cannot go from a stable learner (in the sense of Result 3.2) to a differentially private learner without a domain-size dependence. First observe that our "stability" guarantees on $\mathcal{G}$ (Result 3.2) are somewhat unusual: there exists some *function ball* (around the target concept) such that the collective probability of $\mathcal{G}$ outputting its member functions is high, in contrast to the Boolean setting [10], where global stability means that a *single* function is output with high probability. The stability guarantees differ because, in our setting, the learner only obtains $\varepsilon$-accurate feedback from the adversary. Since all functions that are in the $\varepsilon$-ball of $c$ are consistent with the feedback of the adversary, the learner cannot uniquely identify the target concept $c$. We thus allow the learner to output a function in the $\varepsilon$-ball around the target concept. This difference, however, also prevents us from applying the generic transformation from a stable learner to a private learner employed in [10], as we now explain.[4]

That transformation first generates a list of hypothesis functions by running the stable learner $\mathcal{G}$ of Result 3.2, $n$ many times, each of which outputs a function $f_i$. By Result 3.2 and a Chernoff bound, one can show that with high probability, an $\eta = \zeta^d$-fraction of the list should lie in $\mathcal{B}(\zeta, f)$ for some $f$. Next one would like to *privately* output some function in $\mathcal{B}(\zeta, f)$, i.e. solve the following problem:

**Problem 3.3** (Query release for function balls). *Given as input a list of $n$ functions $\{f_i : \mathcal{X} \to \mathbb{R}\}_{i \in [n]}$, an $\eta$-fraction of which satisfy $|f_i(x) - f^*(x)| \leq \zeta$ for all $x$, $f^* : \mathcal{X} \to \mathbb{R}$ output some function $g$ such that $|g(x) - f^*(x)| \leq \zeta$ for all $x$.*

We also introduce the following related problem:

**Problem 3.4** (Clique identification on a discrete domain). *For domain $\mathcal{Y} = [4]^d$, given as input a relation $R = \{(x, y) \in \mathcal{Y} \times \mathcal{Y} : \|x - y\|_\infty \leq 1\}$ and a dataset $D \in \mathcal{Y}^n$ under the promise that $(x, y) \in R$ for all $x, y \in D$, output a $z \in \mathcal{Y}$ such that $(x, z) \in R$ for every $x \in D$.*

Observe that Problem 3.4 reduces to Problem 3.3. We give a lower bound on the number of samples necessary to solve problem 3.4, which gives a lower bound for our query release on functions balls. In order to do this one can use a slightly non-standard reduction to the well-studied 1-way marginal release problem [24]. Subsequently, one can use a modified argument from fingerprinting codes [25, 26] and show that for $\delta < 1/1500$, every $(1, \delta < 1/n)$-DP algorithm solving Problem 3.4 with probability at least $1499/1500$ requires $n \geq \tilde{\Omega}(\sqrt{d})$. This implies that every $(1, \delta)$-DP [5] algorithm for Problem 3.3 requires $n \geq \tilde{\Omega}(\sqrt{|\mathcal{X}|})$. In particular, this lower bound is unbounded if $|\mathcal{X}|$ is unbounded, which is precisely the case if we are performing quantum learning on the domain of *all* possible 2-outcome measurements. See Section C in the Supplementary material for more.

Linking this discussion back to our motivating setting of quantum learning (see Section 2.2), when $M$ is the orthogonal basis of $n$-qubit Paulis, then $|\mathcal{M}| = 4^n$ and so the above implies that one needs sample complexity $\tilde{\Omega}(4^{n/2})$ in order to go from stability to approximate differential privacy.

We can also give a quadratically-worse upper-bound on the sample complexity of *pure* private PAC learning in our quantum/robust setting (which naturally also upper bounds its *approximate* version). We simply invoke the generic private learner of [27, 10], without going through the intermediate step of producing a stable learner. Given any set of functions $\mathcal{H}$, this learner simply draws $O(\log |\mathcal{H}|)$ examples, and samples a function from $\mathcal{H}$ with probability inverse exponential in the function's loss on the examples. To learn quantum states in $\mathcal{C}$ – again with $\mathcal{M}$ the orthogonal basis of $n$-qubit Paulis – with the Generic Private Learner, we discretize the $[0, 1]$-range of $\mathcal{D}$ into bins of size $\zeta$, obtaining a resulting function class $\mathcal{H}$ with size $(1/\zeta)^{4^n}$. Therefore the sample complexity of differentially-private quantum learning is $\tilde{O}(4^n)$.

**Comparison to prior work [11].** After completion of this work, we were made aware of the work by [11] (in NeurIPS'20). While they extended the work of [10] to multi-class functions, they claim that their results also apply to real-valued learning by discretizing the range of the functions (we

---

[4]The following argument was communicated to us by Mark Bun [23].

[5]In differential privacy, we are typically interested only in values of $\delta < 1/n$, in order to rule out algorithms that simply pick a point in their input uniformly and then process them in a possibly non-private manner.

couldn't find a version of the paper that spells out the proof that online learnability implies a stable real-valued learner, but this seems implicit from their proofs). While they state a notion of stability for learning real-valued functions that, similar to ours, invokes the notion of function balls, in order to go from an online learner to stable learner they crucially rely on a *alternate* Littlestone dimension. In this work, we use the standard notion of sequential fat-shattering dimension – which we also bound for the case of quantum states. Secondly, while [11]'s online and PAC learning settings assume that the learner receives *exact* real-valued labels, in all our learning models, we allow for $\varepsilon$-imprecise labels. Thus, all our implications are robust to such adversarial imprecision (making our work closer to "classification-noise learning"). This additional consideration bars the usage of [10]'s technique, developed for Boolean functions as a black-box. We discuss further differences between our works in the paragraph above Section D in the Supplementary material.

# 4 Faster algorithms for quantum state learning

Our previous sections were written for the general setting of robustly learning a real-valued function class. Now we switch gears, applying these results to our motivating problem: tightly characterizing the complexity of quantum state learning. See section E in the Supplementary material for details.

*1.* **Faster shadow tomography for classes of states.** Aaronson [28] introduced a learning model called *shadow tomography*. Here, the goal is to learn the "shadows" of an unknown quantum state $\rho$, i.e., given $m$ measurements $E_1, \ldots, E_m$, how many copies of $\rho$ suffice to estimate $\mathsf{Tr}(\rho E_i)$ for all $i \in [m]$. Aaronson surprisingly showed that $O(n, \log m)$ copies of $\rho$ suffice for this task (in contrast to the trivial tomography or sampling bound of $O(\exp(n), m)$), and an important open problem is can we get rid of the $n$ dependence (even for a subclass of quantum states)? Subsequently, there were many works that tried to improve the complexity of general shadow tomography [29, 30]. We can now answer this question affirmatively: the complexity of shadow tomography (assuming that the unknown state $\rho$ comes from a set $\mathcal{C}$) can be made $O(\mathsf{sfat}(\mathcal{C}), \log m)$.

**Result 4.1** (Informal). *There is a $(\alpha, \delta)$-gentle $(1 - \varepsilon)$-accurate shadow tomography procedure for a set of states $\mathcal{C}$ using* $\mathrm{poly}\big(\mathsf{sfat}(\mathcal{C}), \log m, 1/(\alpha\varepsilon), \log(1/\delta)\big)$ *copies of $\rho$.*

This result relies on an observation of [7] that a mistake-bounded online learning algorithm (we plug in RSOA) can be used as a subroutine in *gentle* shadow tomography, where gentle can be viewed as a proxy for privacy (albeit, the notion of DP here is with respect to states and different from ours).

*2.* **A better bound on** $\mathsf{sfat}(\cdot)$**.** Let $\mathcal{C}_n$ be the class of *all* $n$-qubit states. As we mentioned earlier, [8] showed that $\mathsf{sfat}(\mathcal{C}_n)$ is at most $O(n)$, but clearly for a subset $\mathcal{C} \subseteq \mathcal{C}_n$ of quantum states it is possible that $\mathsf{sfat}(\mathcal{C}) \ll \mathsf{sfat}(\mathcal{C}_n)$. In this direction, using techniques from quantum random access codes (which was also used before [5, 8, 31, 32]) we first give a general upper bound on the sequential fat shattering dimension of a class of quantum states in terms of Holevo information of an ensemble.

**Result 4.2** (Informal). *Let $\mathcal{C}$ be a set of states on $n$ qubits. Then $\mathsf{sfat}_p(\mathcal{C}) \leq \frac{1}{1-H(p)} \max_{G \subseteq \mathcal{C}} \chi(\mathcal{E}_G)$, where $\chi$ is the Holevo information and $\mathcal{E}_G$ is a uniform ensemble over the states in $G$.*

Now, an immediate consequence of this result is a class of states for which $\mathsf{sfat}(\cdot)$ is much smaller than $n$. Consider the set $\mathcal{C}$ of "$k$-juntas",[6] i.e., each $n$-qubit state lives in the same *unknown* $k$-dimensional subspace. In this case it is not hard to see that $\chi(\mathcal{E}_G)$ is at most $\log k$, which improves upon the trivial upper bound of $n$ on $\mathsf{sfat}(\mathcal{C})$. We discuss more such classes of states below.

*3.* **Learning noisy quantum states.** Let $\mathcal{C}$ again be a class of quantum states and let $\mathcal{N}$ be a quantum channel. Let $\mathcal{C}' = \{\mathcal{N}(U) : U \in \mathcal{C}\}$ be the set of states obtained after passing through quantum channel $\mathcal{N}$. Suppose the goal is to learn $\mathcal{C}'$ (i.e., to learn states that have passed through a noisy channel $\mathcal{N}$). This connects to the question of experimental learning of quantum states, i.e., can we learn the unknown noise (or states) in a noisy-quantum device. Here we show that $\mathsf{sfat}(\mathcal{C}') \leq C(\mathcal{N})$, i.e., $\mathsf{sfat}(\mathcal{C}')$ is upper bounded by the classical capacity of $\mathcal{N}$. Fortuitously, recent advances in quantum Shannon theory enable upper bounds on these previously hard-to-compute capacities, and we use them to bound the complexity of learning states corrupted by noise from depolarizing channels, Pauli channels and bosonic channels (e.g. fiber optic cables). As far as we are aware, this is the first work to consider learnability of continuous-variable states.

---

[6]$k$-juntas are well-studied in computational learning theory, wherein a Boolean function on $n$ bits is a $k$-junta if it depends on an *unknown* subset of $k$ input bits.

### 4.1 Open questions

Our work opens up several questions relevant to both classical and quantum machine learning: (1) Can we extend our work to the agnostic setting (which is more applicable in the setting of implementing quantum algorithm in a device with unknown noise)? (2) Recently, [12] improved upon [10] by showing that one needs a *polynomial* blow-up in sample complexity in going from online learning to differential privacy, can we use similar techniques to improve our results? (3) Classically, [27] established connections between statistical query learning and local differential privacy, can we extend this also to the quantum regime, using the recently defined notion of quantum statistical query learning [33]? (4) Recently, [15] showed that the equivalence between private learning and online learning cannot be made computationally efficient assuming existence of one-way functions, do these extend to the quantum setting?

## Acknowledgments and Disclosure of Funding

We thank Mark Bun for various clarifications and also for showing us how to prove that stability does not imply approximate DPPAC. SA was partially supported by the IBM Research Frontiers Institute and acknowledges support from the Army Research Laboratory, the Army Research Office under grant number W911NF-20-1-0014. YQ was supported by the Stanford QFARM fellowship and an NUS Overseas Graduate Scholarship. JS and SA acknowledge support from the IBM Research Frontiers Institute.

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
