# Private learning implies quantum stability

We show the following web of implications:

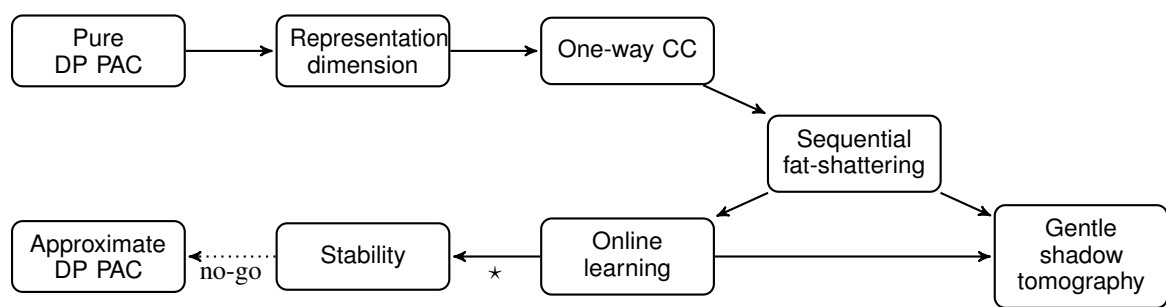

Figure 1: Summary of results for learning real-valued concept classes and quantum states with imprecise feedback. Except for the $\star$-arrow, an arrow $\mathsf{A} \to \mathsf{B}$ implies that, if the sample complexity of learning in model $\mathsf{A}$ or the combinatorial parameter $\mathsf{A}$ is $S_\mathsf{A}$, then the complexity of learning in model $\mathsf{B}$ or the combinatorial parameter $\mathsf{B}$ is $S_\mathsf{B} = \mathrm{poly}(S_\mathsf{A})$. The dotted arrow signifies that a technique used to prove that arrow for Boolean functions is a no-go for our quantum learning setting.

1

## A   Preliminaries

**Notation.**   Throughout this paper we will use the following notation. We let $\mathcal{X}$ be the input domain of real-valued functions (eventually when instantiating to quantum learning, we will let $\mathcal{X}$ be the set of all possible 2-outcome measurements denoted by $\mathcal{M}$). We will let $\mathcal{C}$ be a concept class of real valued functions, i.e., $\mathcal{C} \subseteq \{f : \mathcal{X} \to [0,1]\}$ and let $\mathcal{H}$ be a *collection* of concept classes $\mathcal{C}$. For a distribution $D : \mathcal{X} \to [0,1]$, two functions $h, c : \mathcal{X} \to [0,1]$ and a distance parameter $r \in [0,1]$, we define loss as

$$\mathsf{Loss}_D(h, c, r) := \Pr_{x \sim D}\big[|h(x) - c(x)| > r\big]. \tag{1}$$

**The quantum learning setting.**   While we are interested in the quantum learning setting – learning $n$-qubit quantum states in the class $\mathcal{U}$ over an orthogonal basis of $n$-qubit quantum measurements, $\mathcal{M}$ – our results apply more generally to learning an arbitrary real-valued function class $\mathcal{C} = \{f : \mathcal{X} \to [0,1]\}$ with imprecise adversarial feedback. Therefore the learning models we introduce, and our theorems in the rest of this paper, will be for the more general real-valued setting.

For $\mathcal{X} = \mathcal{M}$, these two problems are equivalent: there is a one-to-one mapping between the set of all quantum states and real-valued functions on $\mathcal{M}$, i.e., for every $\sigma$, one can clearly associate a function $f_\sigma : \mathcal{M} \to [0,1]$ defined as $f_\sigma(M) = \mathsf{Tr}(M\sigma)$ and for the converse direction, given an arbitrary $c : \mathcal{M} \to [0,1]$, one can find a density matrix $\sigma$ for which $c(M) = \mathsf{Tr}(M\sigma)$ for all $M \in \mathcal{M}$ (and this uses the orthogonality of $\mathcal{M}$ crucially). Hence, if one can learn $\mathcal{C}$ for $\mathcal{X} = \mathcal{M}$ then one can learn the class of quantum states $\mathcal{U}$, and the converse is also true. When $\mathcal{U}$ is a subset of the set of all

20  $n$-qubit states, the learner we construct is an improper learner, i.e., it could output $\sigma$ not in $\mathcal{U}$, which
21  nevertheless is useful for prediction.

## A.1  Learning models of interest

23  PAC **learning.**   We first introduce the PAC learning model for the real-valued concept classes.

24  **Definition A.1** (PAC learning)**.**  *Let $\alpha, \zeta \in [0, 1]$. An algorithm $\mathcal{A}$ $(\zeta, \alpha)$-PAC learns $\mathcal{C}$ with sample*
25  *complexity $m$ if the following holds: for every $c \in \mathcal{C}$, and distribution $D : \mathcal{X} \to [0, 1]$, given*
26  *$m$ labelled examples $\{(x_i, \widehat{c}(x_i))\}_{i=1}^m$ where each $x_i \sim D$ and $|c(x_i) - \widehat{c}(x_i)| \leq \zeta/5$, then with*
27  *probability at least $3/4$ (over random examples and randomness of $\mathcal{A}$) outputs a hypothesis $h$*
28  *satisfying*[1]

$$\Pr_{y \sim D}\left[|c(y) - h(y)| \geq \zeta\right] \leq \alpha. \tag{2}$$

29  We remark that in the definition above, we assume the success probability of the algorithm is $3/4$ for
30  notational simplicity. With an overhead of $O(\log(1/\beta))$, we can boost $3/4$ to $1 - \beta$ using standard
31  techniques as mentioned in [1].

32  **Online learning**   Let us now introduce the online learning setting in the form of a game between
33  two players: the learner and the adversary. As always, we shall be concerned with learning real-valued
34  concept classes $\mathcal{C} := \{f : \mathcal{X} \to [0, 1]\}$ and we let the target function be $c \in \mathcal{C}$. In the rest of this
35  paper, we will use the term "online learning" to refer to *improper* online learning[2], also known
36  in the literature as online *prediction*, where the learner's objective is to make predictions for $c(x)$
37  given some point $x \in \mathcal{X}$, and it may do so using a hypothesis function $f(x)$ not necessarily in $\mathcal{C}$.
38  Importantly, we also depart from the real-valued online learning literature in allowing the adversary
39  to be imprecise; that is, for the adversary to respond to the learner with feedback that is $\varepsilon$-away from
40  the true value (this is made more precise below). This generalization allows for the case when the
41  feedback is generated by a randomized algorithm with approximation guarantees, a statistical sample,
42  or a physical measurement.

43  The following setting, which we also call the *strong feedback* setting, was introduced by [2] to model
44  online learning of quantum states. The following procedure repeats for $T$ rounds: at the $t$-th round,

45      1. Adversary provides input point in the domain: $x_t \in \mathcal{X}$.
46      2. Learner has a local prediction function $f_t$ which may not necessarily be in $\mathcal{C}$, and predicts
47         $\hat{y}_t = f_t(x_t) \in [0, 1]$.
48      3. Adversary provides strong feedback $\widehat{c}(x_t) \in [0, 1]$ satisfying $|\widehat{c}(x_t) - c(x_t)| < \varepsilon$.
49      4. Learner suffers loss $|\hat{y}_t - c(x_t)|$.

50  At the end of $T$ rounds, the learner has computed a function $f_{T+1}$, which functions as its prediction
51  rule. If the learner is such that $f_{T+1}$ is not guaranteed to be in $\mathcal{C}$, we call the learner an 'improper
52  learner'. Such a learner can, however, still make predictions $f_{T+1}(x)$ on any given input $x \in \mathcal{X}$.
53  Alternatively, we could also require that the learner be 'proper', that is, it must output some $f_{T+1} \in \mathcal{C}$.
54  Generally, the goal of the learner is either to make as few prediction mistakes as possible within
55  $T$ rounds (where a 'mistake' is defined as $|f(x_t) - c(x_t)| > \varepsilon$, to be discussed more below); or to
56  minimize *regret* for a given notion of loss, which is the total loss of its predictions compared to the
57  loss of the best possible prediction function that could be found with perfect foresight. Because the
58  former, 'mistake-bound' setting is the one relevant to quantum states, we focus on that and will define
59  it next.

60  Some variants of our strong feedback setting could also be considered, and we now explain how they
61  are related to our setting. Firstly, [3] and [4] consider an alternative setting for online prediction of
62  real-valued functions that differs from ours in step (3). There, the adversary's feedback is $c(x_t)$ itself
63  and is infinitely precise; to recover that setting from ours, we merely set $\varepsilon = 0$. Since in our setting
64  we allow $\varepsilon$ arbitrary, we accommodate the possibility of a precision-limited adversary, for instance if

---

[1]An alternative definition of the PAC model of learning is the following: a learner obtains $(x_i, b)$ where
$b \in \{0, 1\}$ satisfies $\Pr[b = 1] = c(x_i)$. Both these models are equivalent up to poly-logarithmic factors.
[2]However, whenever we are concerned with online learning of *quantum states*, we take special care to ensure
that our algorithms are proper, so that the learner's hypothesis function corresponds to an actual quantum state.

the adversary's feedback comes from some estimation process or physical measurement. A second alternative setting is where the adversary only commits to providing *weak feedback*: $\widehat{c}(x_t) = 0$ if $|\hat{y}_t - c(x_t)| < \varepsilon$ and $\widehat{c}(x_t) = 1$ otherwise. Additionally, the adversary specifies if $c(x_t) > \hat{y}_t + \varepsilon$, or $c(x_t) < \hat{y}_t - \varepsilon$ to the learner. We have termed this 'weak feedback' because it contains only two bits of information, whereas for the strong feedback setting considered above, the feedback contains $O(\log(1/\varepsilon))$ bits of information.[3]

**Mistake bound for online learning.** We now introduce the notion of 'mistake bound' of an online learner. Before defining the model, we first define an $\varepsilon$-*mistake* at step $(3)$ of the the $T$-step procedure we mentioned above.

**Definition A.2** ($\varepsilon$-mistake). *Let the target concept be $c$. At a given round, let the input point be $x_t$ and the learner's guess be $\hat{y}_t$. The learner has made a mistake if $|\hat{y}_t - c(x_t)| \geq \varepsilon$.*

We now define the mistake-bound model of online learning.

**Definition A.3** (Mistake bound ). *Let $\mathcal{A}$ be an online learning algorithm for class $\mathcal{C}$. Given any sequence $S = (x_1, \widehat{c}(x_1)), \ldots, (x_T, \widehat{c}(x_T))$, where $T$ is any integer, $c \in \mathcal{C}$ and $\widehat{c}$ is the feedback of the online learner on point $x_i$. Let $M_{\mathcal{A}}(S)$ be the number of mistakes $A$ makes on the sequence $S$.*

*We define the* mistake bound of learner $\mathcal{A}$ *(for $\mathcal{C}$) as $\max_S M_{\mathcal{A}}(S)$ where $S$ is a sequence of the above form. We say that class $\mathcal{C}$ is online learnable if there exists an algorithm $A$ for which $M_{\mathcal{A}}(\mathcal{C}) \leq B < \infty$. We further define the* mistake bound of a concept class *as $M(\mathcal{C}) := \min_{\mathcal{A}} M_{\mathcal{A}}(\mathcal{C})$ where the minimization is over all valid online learners $A$ for $\mathcal{C}$.*

The mistake bound of class $\mathcal{C}$, $M(\mathcal{C})$ is one way to measure the online learnability of $\mathcal{C}$. For learning Boolean function classes, [5] showed that this bound gives an operational interpretation to the Littlestone dimension of the function class: $\min_{\mathcal{A}} M_{\mathcal{A}}(\mathcal{C}) = \mathsf{Ldim}(\mathcal{C})$. For showing that there exists $\mathcal{A}$ such that $M_{\mathcal{A}}(\mathcal{C}) \leq \mathsf{Ldim}(\mathcal{C})$, Littlestone constructed a generic algorithm – the *Standard Optimal Algorithm* – to learn any class $\mathcal{C}$ that makes at most $\mathsf{Ldim}(\mathcal{C})$-many mistakes on any sequence of examples.

The mistake-bounded online learning model outlined in the previous few paragraphs recovers the 'online learning of quantum states' model, proposed by [2], once we specialize to learning quantum states using the translation we provide at the start of this section. Whereas [2]'s focus was on regret bounds for online learning, we instead focus on online learning with bounded mistakes. While this can be viewed as a special case of bounding regret (with an indicator loss function), the mistake-bound viewpoint opens up windows to other models of learning, as we will see in the rest of this paper.

## A.2 Other tools of interest

### A.2.1 Differentially-private learning

The task of designing randomized algorithms with privacy guarantees has attracted much attention classically with the motivation of preserving user privacy [6]. Below we formally introduce *differential privacy*, one way of formalizing privacy. Let $\mathcal{A}$ be a learning algorithm. Let $S$ be a sample set consisting of labelled examples $\{(x_i, \ell_i)\}_{i \in [n]}$ where $x_i \in \mathcal{X}, \ell_i \in [0, 1]$, that is fed to a learning algorithm $\mathcal{A}$. We say two sample sets $S, S'$ are *neighboring* if there exists $i \in [n]$ such that $(x_i, \ell_i) \neq (x'_i, \ell'_i)$ and for all $j \neq i$ it holds that $(x_j, \ell_j) = (x'_j, \ell'_j)$. Additionally, we define $(\varepsilon, \delta)$-indistinguishability of probability distributions: for $a, b, \varepsilon, \delta \in [0, 1]$ let $a \approx_{\varepsilon, \delta} b$ denote the statement $a \leq e^{\varepsilon} b + \delta$ and $b \leq e^{\varepsilon} a + \delta$. We say that two probability distributions $p, q$ are $(\varepsilon, \delta)$-indistinguishable if $p(E) \approx_{\varepsilon, \delta} q(E)$ for every event $E$.

**Definition A.4** (Differentially-private learning). *A randomized algorithm*

$$\mathcal{A} : (\mathcal{X} \times [0, 1])^n \to [0, 1]^X$$

*is $(\varepsilon, \delta)$-differentially-private if for every two neighboring examples $S, S' \in (X \times [0, 1])^n$, the output distributions $\mathcal{A}(S)$ and $\mathcal{A}(S')$ are $(\varepsilon, \delta)$-indistinguishable.*

---

[3]That is to say, a learner that works in the strong feedback setting can also work in the weak feedback setting, by mounting a binary search of the range $[0, 1]$ to obtain for itself an $\varepsilon$-approximation of strong feedback at every round. Conversely, a learner that works for the weak feedback setting also works in the strong feedback setting, by throwing away some information in the strong feedback.

**Definition A.5** (Differentially-private PAC learning)**.** *Let* $\mathcal{C} \subseteq \{f : \mathcal{X} \to [0,1]\}$ *be a concept class. Let* $\zeta, \alpha \in [0,1]$ *be accuracy parameters and* $\varepsilon, \delta$ *be privacy parameters. We say* $\mathcal{C}$ *can be learned with sample complexity* $m(\zeta, \alpha, \varepsilon, \delta)$ *in a private* PAC *manner if there exists an algorithm* $\mathcal{A}$ *that satisfies the following:*

- PAC *learner — Algorithm* $\mathcal{A}$ *is a* $(\zeta, \alpha)$-PAC *learner for* $\mathcal{C}$ *with sample size* $m$ *(as formulated in Definition A.1).*

- *Privacy — Algorithm* $\mathcal{A}$ *is* $(\varepsilon, \delta)$-*differentially private (as formulated in Definition A.4).*

*We shall say such a learner is* $(\zeta, \alpha, \varepsilon, \delta)$-PPAC.

### A.2.2 Communication complexity.

In this section, we introduce one-way classical and quantum communication complexity. Different from the usual setting, here we consider communication protocols that compute real-valued and not just Boolean functions. In the one-way classical communication model, there are two parties Alice and Bob. Let $\mathcal{C} \subseteq \{f : \{0,1\}^n \to [0,1]\}$ be a concept class. We consider the following task which we call $\mathsf{Eval}_\mathcal{C}$: Alice receives a function $f \in \mathcal{C}$ and Bob receives an $x \in \mathcal{X}$. Alice and Bob share random bits and Alice is allowed to send classical bits to Bob, who needs to output a $\zeta$-approximation of $f(x)$ with probability $1 - \varepsilon$. We let $R^{\to}_{\zeta,\varepsilon}(c, x)$ be the *minimum* number of bits that Alice communicates to Bob, so that he can output a $\zeta$-approximation of $f(x)$ with probability at least $1 - \varepsilon$ (where the probability is taken over the randomness of Alice and Bob). Let $R^{\to}_{\zeta,\varepsilon}(\mathcal{C}) = \max\{R^{\to}_{\zeta,\varepsilon}(c, x) : c \in \mathcal{C}, x \in \mathcal{X}\}$.

We will also be interested in the quantum one-way communication model. The setting here is exactly the same as above, except that now Alice and Bob can apply quantum unitaries locally and Alice is allowed to send qubits instead of classical bits to Bob. Like before, we let $Q^{\to}_{\zeta,\varepsilon}(c, x)$ be the *minimum* number of qubits that Alice communicates to Bob, so that he can output a $\zeta$-approximation of $c(x)$ with probability at least $1 - \varepsilon$ (where the probability is taken over the randomness of Alice and Bob). Let $Q^{\to}_{\zeta,\varepsilon}(\mathcal{C}) = \max\{Q^{\to}_{\zeta,\varepsilon}(c, x) : c \in \mathcal{C}, x \in \mathcal{X}\}$.

### A.2.3 Stability of algorithms

An important conceptual contribution in this paper is the concept of *stability* of algorithms. The notion of stability has been used in several previous works [6, 7, 8, 9, 10]. In the context of real-valued functions we are not aware of such a definition. We naturally extend previous definitions of stability from Boolean-valued functions to real-valued functions as follows.

**Definition A.6** (Stability)**.** *Let* $\mathcal{C} \subseteq \{f : X \to [0,1]\}$ *be a concept class and* $\eta, \zeta \in [0,1]$. *Let* $\mathcal{D} : \mathcal{X} \to [0,1]$ *be a distribution and* $c \in \mathcal{C}$ *be a target unknown concept. We say a learning algorithm* $\mathcal{A}$ *is* $(T, \eta, \zeta)$-*stable with respect to* $\mathcal{D}$ *if: given* $T$ *many labelled examples* $S = \{(x_i, c(x_i))\}$ *when* $x_i \sim \mathcal{D}$, *there exists a hypothesis* $f$ *such that*

$$\Pr[\mathcal{A}(S) \in \mathcal{T}(\zeta, f)] \geq \eta,$$

*where the probability is taken over the randomness of the algorithm* $\mathcal{A}$ *and the examples* $S$, *and* $\mathcal{T}(\zeta, f)$ *is the function ball of radius* $\zeta$ *around* $f$, *i.e* $\mathcal{T}(\zeta, f) = \{g : |g(x) - f(x)| < r \text{ for every } x \in \mathcal{X}\}$.

It is worth noting that in the standard notion of global stability (for example the one used in [7]), we say an algorithm $\mathcal{A}$ is stable if a *single* function is output by $\mathcal{A}$ with high probability. In the real-valued robust scenario, one cannot hope for similar guarantees because the adversary is allowed to be $\zeta$-off with his feedback at every round. In particular, the adversary's feedback could correspond to a different function from the target concept $c$. However, the intuition is that any adversarially-chosen alternative function cannot be "too" far from $c$.

Inspired by the definition above we also define quantum stability as follows.

**Definition A.7** (Quantum Stability)**.** *Let* $S$ *be a class on* $n$-*qubit quantum states and* $\eta, \zeta \in [0,1]$. *Let* $\mathcal{D} : \mathcal{X} \to [0,1]$ *be a distribution over orthogonal 2-outcome measurements and* $\rho \in S$ *be an unknown quantum state. We say a learning algorithm* $\mathcal{A}$ *is* $(T, \eta, \zeta)$-*stable with respect to* $D$ *if: given* $T$ *many labelled examples* $Q = \{(E_i, \mathsf{Tr}(\rho E_i))\}$ *when* $E_i \sim \mathcal{D}$, *there exists a quantum state* $\sigma$ *such that*

$$\Pr[\mathcal{A}(Q) \in \mathcal{B}(\varepsilon, \sigma)] \geq \eta, \tag{3}$$

154 *where the probability is taken over the examples in $Q$ and $\mathcal{B}(\varepsilon, \sigma)$ is the ball of states $\varepsilon$-close to $\sigma$*
155 *with respect to $\mathcal{X}$, i.e., $\mathcal{B}(\varepsilon, \sigma) = \{\sigma' : |\mathsf{Tr}(E\sigma) - \mathsf{Tr}(E\sigma')| < \varepsilon \text{ for every } E \in \mathcal{X}\}$.*

### A.2.4 Combinatorial parameters.

157 We define some combinatorial parameters used in PAC learning and online learning real-valued
158 function classes $\{f : \mathcal{X} \to [0, 1]\}$. These are the fat-shattering (for PAC learning) and sequential
159 fat-shattering dimension (for online learning). They can be viewed as the real-valued analogs of
160 the VC dimension and Littlestone dimension respectively for PAC learning and online learning
161 Boolean function classes $\{f : \mathcal{X} \to \{0, 1\}\}$. Below we define the combinatorial parameters for
162 real-valued functions.

163 **Fat-Shattering dimension**    The set $\{x_1, \ldots, x_k\} \subseteq \mathcal{X}$ is $\gamma$-fat-shattered by concept class $\mathcal{C}$ if there
164 exists real numbers $\{\alpha_1, \ldots, \alpha_k\} \in [0, 1]$ such that for all $k$-bit strings $y = (y_1 \cdots y_k)$ there exists a
165 concept $f \in \mathcal{C}$ such that *if $y_i = 0$ then $f(x_i) \leq \alpha_i - \gamma$ and if $y_i = 1$ then $f(x_1) \geq \alpha_i + \gamma$.*

166 The fat-shattering dimension of $\mathcal{C}$, or $\mathsf{fat}_\gamma(\mathcal{C})$ is the largest $k$ for which: there exists $\{x_1, \ldots, x_k\} \in \mathcal{X}$
167 that is $\gamma$-fat-shattered by $\mathcal{C}$. We remark that if the functions in $\mathcal{C}$ have range $\{0, 1\}$ and $\gamma > 0$, then
168 $\mathsf{fat}_\gamma(\mathcal{C})$ is just the standard VC dimension.

169 **Sequential Fat-Shattering dimension**    We also define an analog of the fat-shattering dimension
170 for online learning. The presentation of this dimension closely follows [2]. We say a depth-$k$ tree $T$
171 is an $\varepsilon$-*sequential fat-shattering tree* for $\mathcal{C}$ if it satisfies the following:

172 1. For every internal vertex $w \in T$, there is some domain point $x_w \in U$ and threshold
173 $a_w \in [0, 1]$ associated with $w$, and

174 2. For each leaf vertex $v \in T$, there exists $f \in \mathcal{C}$ that causes us to reach $v$ if we traverse $T$ from
175 the root such that at any internal node $w$ we traverse the left subtree if $f(x_w) \leq a_w - \varepsilon$ and
176 the right subtree if $f(x_w) \geq a_w + \varepsilon$. If we view the leaf $v$ as a $k$-bit string, the function $f$ is
177 such that for all ancestors $u$ of $v$, we have $f(x_u) \leq a_u - \varepsilon$ if $v_i = 0$, and $f(x_u) \geq a_u + \varepsilon$
178 if $v_i = 1$, when $u$ is at depth $i - 1$ from the root.

179 The $\varepsilon$-*sequential fat-shattering dimension* of $\mathcal{C}$, denoted $\mathsf{sfat}_\varepsilon(\mathcal{C})$, is the largest $k$ such that we can
180 construct a complete depth-$k$ binary tree $T$ that is an $\varepsilon$-sequential fat-shattering tree for $\mathcal{C}$. Again,
181 we remark that if the functions in $\mathcal{C}$ have range $\{0, 1\}$ and $\gamma > 0$, then $\mathsf{sfat}_\gamma(\mathcal{C})$ is just the standard
182 Littlestone dimension [5].

183 **Representation dimension.**    The representation dimension of concept class $\mathcal{C}$ roughly considers the
184 collection of all distributions over sets of hypothesis functions (not necessarily from the class $\mathcal{C}$) that
185 "cover" $\mathcal{C}$. We make this precise below. This dimension is known to capture the sample complexity
186 of various models of differential private learning Boolean functions [11, 12]. Because we shall be
187 concerned with learning real-valued concept classes, we define these notions below with an additional
188 'tolerance' parameter $\zeta$.

189 **Definition A.8** (Deterministic representation dimension DRdim, real-valued analog of [12]). *Let*
190 $\mathcal{C} \subseteq \{f : \mathcal{X} \to [0, 1]\}$ *be a concept class. A class of functions $\mathcal{H}$ deterministically $(\zeta, \varepsilon)$-represents $\mathcal{C}$*
191 *if for every $f \in \mathcal{C}$ and every distribution $\mathcal{D} : \mathcal{X} \to [0, 1]$, there exists $h \in \mathcal{H}$ such that*

$$\Pr_{x \sim D} \left[ |h(x) - f(x)| > \zeta \right] \leq \varepsilon. \tag{4}$$

192 *The deterministic representation dimension of $\mathcal{C}$ (abbreviated $\mathsf{DRdim}(\mathcal{C})$) is*

$$\mathsf{DRdim}_{\zeta, \varepsilon}(\mathcal{C}) = \min_{\mathcal{H}} \log |\mathcal{H}| \tag{5}$$

193 *where the minimization is over $\mathcal{H}$ that deterministically $(\zeta, \varepsilon)$-represent $\mathcal{C}$.*

194 **Definition A.9** (Probabilistic representation dimension PRdim, real-valued analog of [13]). *Let*
195 $\mathcal{C} \subseteq \{f : \mathcal{X} \to [0, 1]\}$ *be a concept class. Let $\mathscr{H}$ be a collection of concept classes of real-valued*
196 *functions, and $\mathcal{P} : \mathscr{H} \to [0, 1]$. We say $(\mathscr{H}, \mathcal{P})$ is $(\zeta, \varepsilon, \delta)$-representation of $\mathcal{C}$ if for every $f \in \mathcal{C}$*
197 *and distribution $D : \mathcal{X} \to [0, 1]$, with probability at least $1 - \delta$ (over the choice of $\mathcal{H} \sim \mathcal{P}$), there*
198 *exists $h \in \mathcal{H}$ such that*

$$\Pr_{x \sim D} \left[ |h(x) - f(x)| > \zeta \right] \leq \varepsilon. \tag{6}$$

199 *The probabilistic representation dimension of $\mathcal{C}$ (abbreviated $\mathsf{PRdim}(\mathcal{C})$) is*

$$\mathsf{PRdim}_{\zeta,\varepsilon,\delta}(\mathcal{C}) = \min_{(\mathscr{H},\mathcal{P})} \max_{\mathcal{H}\in\mathsf{supp}(\mathscr{H})} \log|\mathcal{H}|, \tag{7}$$

200 *where the outer minimization is over all sets $(\mathscr{H},\mathcal{P})$ of valid $(\zeta,\varepsilon,\delta)$-representations.*

## B  Robust standard optimal algorithm and mistake bounds

202 In this section, we present an algorithm that improperly online-learns a real-valued function class $\mathcal{C}$,
203 making at most $\mathsf{sfat}(\mathcal{C})$ many mistakes (see Definition A.3). This algorithm is an important tool for
204 results in the rest of the paper. All results in this section are presented for the general case of online-
205 learning arbitrary real-valued function classes, with imprecise adversarial feedback. Ultimately, we
206 will use this algorithm as a subroutine for the specific setting of quantum learning.

207 For learning a Boolean function class $\mathcal{C}$, [5] showed that the mistake bound $M(\mathcal{C})$ is equal to the
208 Littlestone dimension $\mathsf{Ldim}(\mathcal{C})$, thus giving an operational interpretation to this dimension. The
209 aim of this section is to examine if the same operational interpretation holds for the sequential
210 fat-shattering dimension, in the context of online-learning real-valued functions with strong feedback
211 (which is also the setting most relevant to quantum learning).

212 Our algorithm's learning setting generalizes that of [3] and [4], who also studied online learning of
213 real-valued and multi-class functions (i.e. functions mapping to a finite set), albeit, the former in
214 the case of precise adversarial feedback ($\varepsilon = 0$). [4] defined several extensions of the Littlestone
215 dimension $\mathsf{Ldim}_\tau$ for $\tau \in (0,2)$ and showed that for learning a multi-class function class $\mathcal{C}$, $\mathsf{Ldim}_\tau <$
216 $M(\mathcal{C}) < \mathsf{Ldim}_{2\tau}$. They also showed that for a real-valued function class $\mathcal{C}$, $\mathsf{sfat}(\mathcal{C})$ is linked to the
217 $\mathsf{Ldim}_\tau$ of a discretization of the function class, thus effectively transforming any real-valued learning
218 problem into a multi-class learning problem. However, their approach does not work for our setting,
219 for the following reason: if $c$ is the target real-valued function, and the true value of $c(x)$ is $\varepsilon$-close
220 to a boundary of some class within the discretized range, our $\varepsilon$-imprecise adversary could choose a
221 value of the feedback $\widehat{c}(x)$ that falls in the neighboring class. Hence the resulting multi-class learner
222 has to deal with the adversary reporting the wrong class, which is beyond the scope of what they
223 considered.

224 In Section B.1, we first construct an algorithm Robust Standard Optimal Algorithm (RSOA) whose
225 mistake bound satisfies $M_{\mathsf{RSOA}}(\mathcal{C}) \leq \mathsf{sfat}(\mathcal{C})$ for online-learning with strong feedback. In Section B.2,
226 we prove some of the properties of this algorithm, which are essential for proving later results in
227 this paper. Moreover, for online learning with weak feedback, we show that $M(\mathcal{C}) \geq \mathsf{sfat}(\mathcal{C})$.
228 However, since the type of feedback differs in these two models we consider, we cannot yet state
229 that $M(\mathcal{C}) = \mathsf{sfat}(\mathcal{C})$ when $\mathcal{C}$ is a real-valued function class (this would be the real-valued analog of
230 the relation $M(\mathcal{C}) = \mathsf{Ldim}(\mathcal{C})$ for Boolean function classes). It is an open question whether we can
231 close this gap, but for the rest of this paper, we are concerned solely with online learning with strong
232 feedback and hence the implication $M_{\mathsf{RSOA}}(\mathcal{C}) \leq \mathsf{sfat}(\mathcal{C})$ is sufficient.

### B.1  Robust Standard Optimal Algorithm

234 In this section, we give an algorithm to to online-learn real-valued functions with strong feedback.
235 In order to handle subtleties caused by learning functions with output in $[0,1]$ instead of $\{0,1\}$, we
236 define the notion of an $\zeta$-cover. This was introduced by [3] and in order to handle inaccuracies in the
237 output of an adversary, we extend their notion to define an *interleaved $\zeta$-cover*.

238 **Definition B.1** ($\zeta$-cover and interleaved $\zeta$-cover)**.** *Let $0 < \zeta < 1$ be such that $1/\zeta$ is an integer.*
239 *A $\zeta$-cover of the $[0,1]$ interval is a set of non-overlapping half-open intervals ('bins') of width $\zeta$*
240 *given by $\big\{[0,\zeta),[\zeta,2\zeta),\ldots,[1-\zeta,1]\big\}$ with the midpoints $\mathscr{I}_\zeta = \big\{\zeta/2,3\zeta/2,\ldots,1-\zeta/2\big\}$ where*
241 *$|\mathscr{I}_\zeta| = 1/\zeta$. Given a $\zeta$-cover $\mathscr{I}_\zeta$, the corresponding interleaved $\zeta$-cover $\tilde{\mathscr{I}}_\zeta$ is the set of overlapping*
242 *half-open intervals ('super-bins') of width $2\zeta$ (each consisting of two adjacent bins in $\mathscr{I}_\zeta$) given by*
243 *$\big\{[0,2\zeta),[\zeta,3\zeta),\ldots,[1-2\zeta,1]\big\}$ with the midpoints $\tilde{\mathscr{I}}_\zeta = \big\{\zeta,2\zeta,\ldots,1-\zeta\big\}$ where $|\tilde{\mathscr{I}}_\zeta| = |\mathscr{I}_\zeta|-1$.*
244 *We denote a super-bin with midpoint $r$ as $\mathsf{SB}(r)$.*

245 We will also need the definition of a $\zeta$-ball.

246 **Definition B.2** ($\zeta$-ball)**.** *An $\zeta$-ball around an arbitrary point $x \in [0,1]$ (denoted $B(\zeta,x)$) is the open*
247 *interval of radius $\zeta$ around $x$, i.e., $B(\zeta,x) := (x-\zeta,x+\zeta)$*

As we mentioned earlier, the FAT-SOA algorithm of [3] used $\alpha$-covers to understand real-valued online learning, however, it does not suffice in the setting of quantum learning since the output of the adversary could be imprecise. To account for this, we use interleaved $\alpha$-covers defined above. Our learning algorithm will take advantage of the following property enjoyed by the interleaved $\alpha$-cover: the $\zeta$-ball of any point is guaranteed to be *entirely* contained inside some super-bin, i.e., for every $x \in (\zeta, 1 - \zeta)$, $\alpha > 2\zeta$ and $r = \arg\min_{r \in \tilde{\mathscr{I}}_{2\zeta}} \{|x - r|\}$, we have $B(\zeta, x) \subset \mathsf{SB}(r)$. Finally, we need one more notation: given a set of functions $V \subseteq \{f : \mathcal{X} \to [0, 1]\}$, $r \in \tilde{\mathscr{I}}_{2\zeta}$ and $x \in \mathcal{X}$, define a (possibly empty) subset $V(r, x) \subseteq V$ as

$$V(r, x) = \{f \in V : f(x) \in B(2\zeta, r)\},$$

i.e., $V(r, x)$ are the set of functions $f \in V$ for which $f(x)$ is within a $2\zeta$-ball around $r$ or $f(x) \in [r - 2\zeta, r + 2\zeta]$. We are now ready to present our mistake-bounded online learning algorithm for learning real-valued functions. Our algorithm is Algorithm 1.

---

**Algorithm 1** Robust Standard Optimal Algorithm, RSOA$_\zeta$

---

**Input:** Concept class $\mathcal{C} \subseteq \{f : \mathcal{X} \to [0, 1]\}$, target (unknown) concept $c \in \mathcal{C}$, and $\zeta \in [0, 1]$.

**Initialize**: $V_1 \leftarrow \mathcal{C}$

1: **for** $t = 1, \ldots, T$ **do**
2:     A learner receives $x_t$ and maintains set $V_t$, a set of "surviving functions".
3:     For every super-bin midpoint $r \in \tilde{\mathscr{I}}_{2\zeta}$ the learner computes the set of functions $V_t(r, x_t)$.
4:     A learner finds the super-bin which achieves the maximum $\mathsf{sfat}(\cdot)$ dimension

$$R_t(x_t) := \left\{ \arg\max_{r \in \tilde{\mathscr{I}}_{2\zeta}} \mathsf{sfat}_{2\zeta}\left(V_t(r, x_t)\right) \in \tilde{\mathscr{I}}_{2\zeta} \right\}$$

5:     The learner computes the mean of the set $R_t(x_t)$, i.e., let

$$\hat{y}_t := \frac{1}{|R_t(x_t)|} \sum_{r \in R_t(x_t)} r.$$

6:     The learner outputs $\hat{y}_t$ and receives feedback $\widehat{c}(x_t)$.
7:     Learner makes the update $V_{t+1} \leftarrow \{g \in V_t \mid g(x_t) \in B(\zeta, \widehat{c}(x_t))\}$
8: **end for**

**Outputs:** The intermediate predictions $\hat{y}_t$ for $t \in [T]$, and a final prediction function/hypothesis which is given by $f(x) := R_{T+1}(x)$.

---

We first provide some intuition about this algorithm. At round $t$, the set of functions that has 'survived' all previous rounds is $V_t$: in particular, $V_t$ consists of functions which are consistent with the feedback received in the previous $t - 1$ iterations. Here, 'consistent' means that suppose $x_1, \ldots, x_{t-1}$ were presented to a learner previously, then, for every $g \in V_t$, $g(x_i) \in B(\zeta, \widehat{c}(x_i))$ for $i \in [t - 1]$. This is clear from Line 7 of the algorithm; indeed, notice that $V_t$ either stays the same as $V_{t-1}$ or shrinks at every round. At round $t$, once a learner receives $x_t$, it always replies with $\hat{y}_t$ that is either $\zeta$-close to the true $c(x_t)$ else, aims to reduce $V_{t-1}$ as much as possible. In particular, for every super-bin $r \in \tilde{\mathscr{I}}_{2\zeta}$, the learner identifies the subset of surviving functions that map to that super-bin at $x_t$, i.e., $f \in V_t$ that satisfy $f(x_t) \in B(2\zeta, r)$. This forms the set $V_t(r, x_t)$. The learner then computes $\mathsf{sfat}_{2\zeta}$ of the set of functions $V_t(r, x_t)$ and picks out the super-bins $r \in \tilde{\mathscr{I}}_{2\zeta}$ that maximize this combinatorial quantity, and output the mean of their midpoints as the prediction $\hat{y}_t$. Intuitively, the parameter $\mathsf{sfat}(\cdot)$ serves as a surrogate metric for the number of functions mapping to a certain interval. Using $\mathsf{sfat}(\cdot)$ to define this prediction rule thus maximizes the number of eliminated functions for every mistake of the learner. Once it receives the feedback $\widehat{c}(x_t)$, the learner updates $V_t$ to $V_{t+1}$ and this process repeats for $T$ steps. We now list a few properties of this algorithm.

### B.2 Properties and guarantees of RSOA

**Lemma B.3.** RSOA$_\zeta$ *(denoted* RSOA*) has the following properties:*

1. *ζ-consistency: at the t-th iteration every $f \in V_t$ satisfies $|f(x_i) - \widehat{c}(x_i)| \le \zeta$ for $i \in [t-1]$.*

2. *Correctness: the target function $c$ is never eliminated, i.e., $c \in V_t$ for every $t \in [T]$.*

3. *For every $t \in [T], x \in \mathcal{X}$, any pair of points $r, r' \in \tilde{\mathscr{I}}_{2\zeta}$ for which*

$$\mathsf{sfat}_{2\zeta}\left(V_t(r, x)\right) = \mathsf{sfat}_{2\zeta}\left(V_t(r', x)\right) = \mathsf{sfat}_{2\zeta}\left(V_t\right) \tag{8}$$

   *also satisfies $|r - r'| < 4\zeta$. Additionally for all $r \in \tilde{\mathscr{I}}_{2\zeta}$, $\mathsf{sfat}_{2\zeta}\left(V_t(r, x)\right) \le \mathsf{sfat}_{2\zeta}\left(V_t\right)$.*

4. RSOA *is deterministic, i.e., for the same sequence of inputs $(x_1, \widehat{c}(x_1)), \ldots, (x_T, \widehat{c}(x_T))$ provided by the adversary to the learner (each of which is followed by a response $\widehat{y}_1, \ldots, \widehat{y}_T$ of the learner), the* RSOA *algorithm produces the same function $f$.*

*Proof.* The first item follows by construction. At the end of $i$th round, the following update is performed: $V_{i+1} \leftarrow \{g \in V_i \mid g(x) \in B(\zeta, \widehat{c}(x_i)))\} \subseteq V_i$. This eliminates all functions $g$ for which $g(x_i) \notin B(\zeta, \widehat{c}(x_i))$ from the set $V_{i+1}$, hence all functions for which $|f(x_i) - \widehat{c}(x_i)| > \zeta$ are eliminated.

The second item follows trivially: by assumption $y_t = c(x_t)$ is in the $\zeta$-ball of $\widehat{c}(x_t)$. Thus the target concept $c$ is never eliminated in the update $V_{t+1} \leftarrow \{g \in V_t \mid g(x) \in B(\zeta, \widehat{c}(x_t))\}$.

We now show the third item. Suppose by contradiction, there is a pair $r, r' \in \tilde{\mathscr{I}}_{2\zeta}$ such that

$$\mathsf{sfat}_{2\zeta}\left(V_t(r, x)\right) = \mathsf{sfat}_{2\zeta}\left(V_t(r', x)\right) = \mathsf{sfat}_{2\zeta}\left(V_t\right)$$

and $|r - r'| > 4\zeta$. Let $\mathsf{sfat}_{2\zeta}\left(V_t\right) = d$. Without loss of generality, we assume $r > r'$. Then let $s = (r + r')/2$. Clearly, for every $f \in V_t(r, x)$ we have $f(x) \ge s + \zeta$ and $g \in V_t(r', x)$ we have $g(x) \le s - \zeta$. This means that, given a sequential fat-shattering tree of depth $d$ for $V_t(r, x)$, and the tree also of depth $d$ for $V_t(r', x)$, we may join them together by adding a root node with the label $x$ and the threshold $s$, and this new tree of depth $d+1$ is sequentially fat-shattered by $V_t(r, x) \cup V_t(r', x)$ and hence by $V_t$ (which is a superset). This contradicts the assumption that $\mathsf{sfat}_{2\zeta}(V_t) = d$, because by definition of $\mathsf{sfat}(\cdot)$ dimension, $d$ is the depth of the *deepest* tree for the functions in $V_t$. The "additionally" part follows immediately because $V_t(r, x) \subseteq V_t$.

The final item of the lemma is clear because steps 3 to 7 in the RSOA algorithm are deterministic and involve no randomness from a learner. $\square$

Having established these properties, are now ready to prove our main theorem bounding the maximum number of prediction mistakes that RSOA makes.

**Theorem B.4** (RSOA mistake bound)**.** *Let $\mathcal{C} \subseteq \{f : \mathcal{X} \to [0, 1]\}$ be a concept class and $\zeta > 0$. Given the setting of online learning with strong feedback, i.e., at every round $t \in [T]$, the feedback $\widehat{c}(x_t)$ is $\zeta$-close to the true value $|c(x_t) - \widehat{c}(x_t)| \le \zeta$, $\mathsf{RSOA}_\zeta$ (described in Algorithm 1) is such that, for every $T$, the algorithm makes a predictions $\hat{y}_t$ satisfying*

$$\sum_{t=1}^{T} \mathbb{I}\left[|\hat{y}_t - c(x_t)| > 5\zeta\right] \le \mathsf{sfat}_{2\zeta}(\mathcal{C})$$

*Proof.* The intuition is that whenever the learner makes a mistake, functions are eliminated from the 'surviving set', such that $\mathsf{sfat}(\cdot)$ of the remaining functions decreases by 1. Since the true function $c$ is never eliminated from $V_t$, and the $\mathsf{sfat}(\cdot)$ dimension of a set consisting of a single function is 0, no more than $\mathsf{sfat}(\cdot)$ mistakes can be made.

First observe that, whenever the algorithm makes a mistake, i.e., $|\hat{y}_t - c(x_t)| > 5\zeta$, it also follows that $|\hat{y}_t - \widehat{c}(x_t)| > 4\zeta$ because $\widehat{c}(x_t)$ is an $\zeta$-approximation of $c(x_t)$. Below we show that on every round where $|\hat{y}_t - \widehat{c}(x_t)| > 4\zeta$, $\mathsf{sfat}(V_{t+1}) \le \mathsf{sfat}(V_t) - 1$. Together with property 2 of Lemma B.3 and the fact that $V_1 = \mathcal{C}$ this already implies that no more than $\mathsf{sfat}(\mathcal{C})$ mistakes are made by RSOA.

Suppose $|\hat{y}_t - \widehat{c}(x_t)| > 4\zeta$. Fix $t$ and $x_t$. Observe that by property 3 Eq. (8) (in Lemma B.3) there are at most three super-bins whose midpoints $r$ satisfy $\mathsf{sfat}_{2\zeta}\left(V_t(r, x)\right) = \mathsf{sfat}_{2\zeta}\left(V_t\right)$, i.e., between 0 and 3 super-bins achieve the upper-bound on $\mathsf{sfat}(\cdot)$ at each round, which we now call $\mathsf{UB}_t := \mathsf{sfat}_{2\zeta}(V_t)$. We now analyze each of four cases for the number of upper-bound-achieving super-bins.

**Case 1**: $\mathsf{sfat}_{2\zeta}(V_t(r,x_t)) < \mathsf{UB}_t$ for every $r \in \tilde{\mathscr{I}}_{2\zeta}$, i.e., no super-bins achieve $\mathsf{UB}_t$. Every update of $V_t$ updates it to the functions within some $\zeta$-ball, $\bigcirc := B(\zeta, \widehat{c}(x_t))$. Observe that $\bigcirc$ is entirely contained within some super-bin, call it $\mathsf{SB}$ (note that even if $\widehat{c}_t$ is at the boundary of two super-bins, it would still be inside the super-bin that is in-between the two, by definition of the interleaved $\zeta$-cover). Hence, $\mathsf{sfat}(\bigcirc) \leq \mathsf{sfat}(\mathsf{SB}) < \mathsf{UB}_t$ where the second inequality is by the assumption of the case.

**Case 2**: There exists exactly one $r \in \tilde{\mathscr{I}}_{2\zeta}$ such that

$$\mathsf{sfat}_{2\zeta}(V_t(r,x_t)) = \mathsf{UB}_t,$$

i.e., exactly one super-bin (centered at $r = 2k\zeta$ for some $k \in \mathbb{Z}_+$) achieves $\mathsf{UB}_t$, let's call this $\mathsf{SB}^* = [2(k-1)\zeta, 2(k+1)\zeta)$. Since the super-bin's midpoint is at some bin boundary, the prediction is $\hat{y}_t = 2k\zeta$. Similar to the previous case, the update step retains only the functions in some $\bigcirc := B(\zeta, \widehat{c}(x_t))$. However, since $|\hat{y}_t - \widehat{c}(x_t)| > 4\zeta$, we either have $\widehat{c}(x_t) < 2(k-2)\zeta$ or $\widehat{c}(x_t) > 2(k+2)\zeta$. $\bigcirc$, therefore, is entirely contained within some super-bin $\mathsf{SB} \neq \mathsf{SB}^*$. Since there is only one maximizing super-bin $\mathsf{SB}^*$, we have $\mathsf{sfat}(\bigcirc) \leq \mathsf{sfat}(\mathsf{SB}) < \mathsf{sfat}(\mathsf{SB}^*) = \mathsf{UB}_t$.

**Case 3**: There exists $r_1, r_2 \in \tilde{\mathscr{I}}_{2\zeta}$ such that

$$\mathsf{sfat}_{2\zeta}(V_t(r_1,x_t)) = \mathsf{sfat}_{2\zeta}(V_t(r_2,x_t)) = \mathsf{UB}_t,$$

i.e., two super-bins (centered at $r_1, r_2$ respectively) achieve $\mathsf{UB}_t$, call them $\mathsf{SB}_1^*, \mathsf{SB}_2^*$. Using Property 3 of Lemma B.3, these two super-bins must either be touching at a boundary (hence $\hat{y}_t = 2k\zeta$ where $\mathsf{SB}_1^* = [2k\zeta, 2(k+2)\zeta)$, $\mathsf{SB}_2^* = [2(k-2)\zeta, 2k\zeta)$) or intersecting at one bin (hence $\hat{y}_t = (2k+1)\zeta$ where $\mathsf{SB}_1^* = [2k\zeta, 2(k+2)\zeta)$, $\mathsf{SB}_2^* = [2(k-1)\zeta, 2(k+1)\zeta)$). In the former case, $\widehat{c}(x_t) < 2(k-2)\zeta$ or $\widehat{c}(x_t) > 2(k+2)\zeta$ and thus neither $\mathsf{SB}_1^*$ nor $\mathsf{SB}_2^*$ entirely contains $\bigcirc$, though there is some super-bin that does. In the latter case, $\widehat{c}(x_t) < (2k-3)\zeta$ or $\widehat{c}(x_t) > (2k+5)\zeta$ and thus neither $\mathsf{SB}_1^*$ nor $\mathsf{SB}_2^*$ entirely contains $\bigcirc$, though there is some super-bin that does. Identical reasoning to the previous two cases shows that the update thus decreases $\mathsf{sfat}(\cdot)$ on the remaining functions.

**Case 4**: There exists $r_1, r_2, r_3 \in \tilde{\mathscr{I}}_{2\zeta}$ such that

$$\mathsf{sfat}_{2\zeta}(V_t(r_1,x_t)) = \mathsf{sfat}_{2\zeta}(V_t(r_2,x_t)) = \mathsf{sfat}_{2\zeta}(V_t(r_3,x_t)) = \mathsf{UB}_t,$$

i.e., three super-bins (centered at $r_1$, $r_2$, $r_3$ respectively) achieve $\mathsf{UB}_t$. Call them $\mathsf{SB}_1^*, \mathsf{SB}_2^*, \mathsf{SB}_3^*$. By Property 3 of Lemma B.3, there is only one configuration these three super-bins could be in, namely two super-bins have to be touching at a boundary, with the last super-bin straddling them: $\mathsf{SB}_1^* = [2k\zeta, 2(k+2)\zeta)$, $\mathsf{SB}_2^* = [2(k-1)\zeta, 2(k+1)\zeta)$, $\mathsf{SB}_3^* = [2(k-2)\zeta, 2k\zeta)$. Then $\hat{y}_t = 2k\zeta$ and $\widehat{c}(x_t) < 2(k-2)\zeta$ or $a - t > 2(k+2)\zeta$. None of $\mathsf{SB}_1^*, \mathsf{SB}_2^*, \mathsf{SB}_3^*$ entirely contains $\bigcirc$, though there is some super-bin that does, and identical reasoning to the previous three cases shows that the update thus decreases $\mathsf{sfat}(\cdot)$ on the remaining functions. $\qquad\square$

Theorem B.4 says that the RSOA algorithm for a concept class $\mathcal{C}$ in the strong feedback model, makes at most $\mathsf{sfat}(\mathcal{C})$ mistakes. This is also the setting in the rest of the paper as well as most of the real-valued online learning literature. A natural question is, can we make fewer mistakes than the RSOA algorithm? Below we consider the *weak* feedback model of online learning and show no learner can do better than making $\mathsf{sfat}(\cdot)$ mistakes. An interesting open question is, can we even improve the lower bound in the theorem below for the strong feedback model setting?

**Theorem B.5.** *Let $\zeta \in [0,1]$ and $\mathcal{C} \subseteq \{f : \mathcal{X} \to [0,1]\}$. Every online learner $\mathcal{A}$ (in the weak feedback setting) for the class $\mathcal{C}$, satisfies $M_{\mathcal{A}}(\mathcal{C}) \geq \mathsf{sfat}_\zeta(\mathcal{C})$.*

*Proof.* We construct an adversary that can always force at least $\mathsf{sfat}(\mathcal{C})$ mistakes in the weak model of learning (where the adversary only gives two bits of feedback to the learner). To do so, the adversary traverses the $\zeta$-fat-shattered tree starting at the root node, at every round interacting with the learner based on the information at the current node, *always* claiming the learner made a mistake, and then moving to one of the two daughter nodes. In particular, the interaction at node $v$ of the tree, which is associated with $(x_v, a_v)$, is as follows: The adversary gives the learner the point $x_v$. If the learner predicts $\hat{y}_t < a_v$, claim the learner is wrong and go to the right daughter node, thus committing the adversary to the subset of functions $f \in \mathcal{C}$ such that $f(x_v) \geq a_v + \zeta$. Go to the opposite node if the learner predicts $\hat{y}_t \geq a_v$. After $\mathsf{sfat}_\zeta(\mathcal{C})$ rounds, the adversary will have reached a leaf node. At this point, by the definition of the $\mathsf{sfat}(\cdot)$ tree, there is at least one function consistent with all previous commitments of the adversary. This becomes the target function, which the adversary then commits to in the first place. Since the depth of the tree is by definition $\mathsf{sfat}_\zeta(\mathcal{C})$, the learner will have made $\mathsf{sfat}_\zeta(\mathcal{C})$ mistakes by the time the adversary reaches a leaf and has to commit to a function. $\qquad\square$

## C Online learning implies stability

In this section we show that online learnability of a real-valued function class implies that there exists a real-valued DP PAC learner for the same class. More precisely, we will assume that the $\mathsf{sfat}(\cdot)$ dimension of the function class is bounded (which implies its online learnability, as discussed in Section B); then we will explicitly describe an algorithm that uses this learner to learn in a globally-stable manner.

This, however, is only half of the implication shown in [7]. There, they go one step further and turn their stable learner into an approximately DP PAC learner, concluding overall that online learning implies approximate DP PAC learning. Supposing we could prove the same for our learning model, then combining this with the implication shown in Section D (that pure DP PAC learning implies online learning) would make for almost a complete chain of implications starting at pure DP PAC learning, implying online learning, and finally implying approximate DP PAC learning. However, in the second half of this section, we use an argument from fingerprinting codes to show that the transformation in [7] from a stable learner to a DP PAC learner does not work with the stability guarantees we obtain for our real-valued learning setting.

We will use the following notation throughout this section. Let $\mathcal{C} \subseteq \{f : \mathcal{X} \to [0, 1]\}$ be a concept class and $c \in \mathcal{C}$ be a target concept. Let $D : \mathcal{X} \to [0, 1]$ be a distribution. In a slight abuse of notation, we use the notation $(x, \widehat{c}(x)) \sim D$ to mean that $x$ is drawn from the distribution $D$ and $\widehat{c}(x)$ satisfies $|\widehat{c}(x) - c(x)| < \zeta$. Also, we say $B \sim D^m$ to mean that a learner receives $m$ such examples $\{(x_i, \widehat{c}(x_i))\}_{i=1}^m$. We say that the learner has made a *mistake* on input $x$ if he has made a $5\zeta$-mistake (refer to Definition A.2). Finally, because we are concerned with *real-valued* learning, it is often the case that functions in the vicinity of the target function are considered "close enough" as hypotheses, and so we will make use of the following notion of *function ball*:

**Definition C.1** (Function ball of radius $r$ around $c$)**.** *Given a set of functions $\mathcal{H} \subseteq \{f : \mathcal{X} \to [0, 1]\}$, a function ball of radius $r$ around $c \in \mathcal{H}$ is the set of all functions $f \in \mathcal{H}$ such that*

$$|f(x) - c(x)| < r \quad \text{for every } x \in \mathcal{X}, \tag{9}$$

*and we denote such a function ball by $\mathcal{T}(r, c)$.[4] Moreover, for a set of functions $\mathcal{E} = \{f_1, \ldots, f_k\}$, we let $\mathcal{T}(r, \mathcal{E}) = \cup_{i=1}^k \mathcal{T}(r, f_i)$.*

In Section C.1, we prove that given a mistake-bounded online learner, there exists a stable learner. In Section C.2, we prove that stability does not, in turn, imply approximate DP learning using the transformation of [7], without a domain size dependence in the sample complexity. In Section C.3, we turn our attention to how our results apply to learning quantum states.

### C.1 Online learning implies stability

In this subsection we prove the following theorem:

**Theorem C.2.** *Let $\alpha, \zeta \in [0, 1]$. Let $\mathcal{C} \subseteq \{f : \mathcal{X} \to [0, 1]\}$ be a concept class with $\mathsf{sfat}_{2\zeta}(\mathcal{C}) = d$. Let $D : \mathcal{X} \to [0, 1]$ be a distribution and let $S = \{(x_i, \widehat{c}(x_i))\}$ be a set of*

$$T = O\left(\zeta^{-d} \cdot \frac{d}{\alpha}\right)$$

*examples where $x_i \sim D$ and $|\widehat{c}(x_i) - c(x_i)| < \zeta$ where $c \in \mathcal{C}$ is a unknown concept. There exists a $(T, \zeta^{-O(d)}, O(\zeta))$-stable learning algorithm $\mathcal{G}$, that outputs $f$ satisfying $\mathsf{Loss}_D(f, c, O(\zeta)) \leq \alpha$.*

The algorithm $\mathcal{G}$ is the RSOA run on a carefully tailored input distribution over the examples, with $T$ being the overall sample complexity of our algorithm. Most of the work in the proof arises in explaining how to tailor the set of examples drawn from the original distribution $D$ into a new set $S$ on which RSOA is guaranteed to succeed. In this section, when we write $\mathsf{RSOA}_\zeta(S)$ where $S$ is a sample, i.e., $S = \{(x_i, \widehat{c}(x_i))\}$, we mean that we feed the examples in $S$ into RSOA sequentially, as in the online learning setting. We will prove this theorem in three parts, corresponding to the subsequent three sub-subsections:

---

[4]The symbol $\mathcal{T}$ stands for 'tube' since for a member of the function ball, closeness to $c$ must be satisfied at not just a single point but all points in the domain. We usually omit mentioning the function class $\mathcal{C}$, which is usually taken to be $\mathcal{R}$, the set of all functions output by RSOA. Because RSOA is an improper learner, $\mathcal{R}$ is not the same as $\mathcal{C}$.

- Our Algorithm 2, is a tailoring algorithm that *defines* distributions $\text{ext}(\mathcal{D}, k)$ for $k \in [d]$ as a function of the distributions $\mathcal{D}$, to which we have black-box access. Just as in [7], the key idea for the tailoring is to inject examples into the sample that would force mistakes. We have adapted this idea for the robust, real-valued setting. Unfortunately, this algorithm could potentially use an unbounded number of examples (in the worst case), which we handle next.

- Next, we seek to impose a cutoff on the number of examples drawn in the algorithm above. In Lemma C.5, we compute the expected number of examples drawn by Algorithm 2. Then, we use Markov's inequality to compute what the cutoff should be. The final tailoring algorithm is simply Algorithm 2, cut off when the number of examples drawn exceeds this threshold.

- Finally, we state the globally-stable learning algorithm Algorithm 3, which essentially invokes Algorithm 2 with the cutoff we defined above. In Theorem C.6 we prove the correctness and sample complexity of Algorithm 3.

### C.1.1 Sampling from the distributions $\text{ext}(D, k)$

In the following, the symbol $S \circ T$ between two sets of examples means the concatenation of the two sets $S, T$. Intuitively our learning algorithm is going to obtain $T$ examples overall and break these examples into blocks of size $m$ (a parameter which will be fixed later in Theorem C.6), each block followed by a single mistake example, all of which which are fed to an online learner. Additionally, below we can think of $k \leq \text{sfat}(\mathcal{C})$ as the number of mistakes we want to inject into the examples we feed to an online learner.

---

**Algorithm 2** An algorithm to sample from distributions $\text{ext}(D, k)$.

**Input:** Distribution $D : \mathcal{X} \to [0, 1]$, $m \geq 1$, $k \in \{0, \ldots, d\}$.

**Output:** A sample from the distribution $\text{ext}(D, k)$.

For $k \geq 0$, the distributions $\text{ext}(D, k) : \mathcal{X}^{k(m+1)} \times [0, 1] \to [0, 1]$ are defined inductively as follows:

1. $\text{ext}(D, 0)$ : output the empty sample $\emptyset$ with probability 1.

2. Sampling from $\text{ext}(D, k)$ involves recursively sampling from $\text{ext}(D, k - 1)$ as follows:

   (a) Draw $S^{(0)}, S^{(1)} \sim \text{ext}(D, k - 1)$ and two sets of $m$ examples $B^{(0)}, B^{(1)} \sim D^m$.

   (b) Let $f_0 = \text{RSOA}_\zeta \left( S^{(0)} \circ B^{(0)} \right), f_1 = \text{RSOA}_\zeta \left( S^{(1)} \circ B^{(1)} \right)$.

   (c) If $|f_0(x) - f_1(x)| \leq 11\zeta$ for every $x \in \mathcal{X}$ then go back to step (i).

   (d) Else pick $x'$ such that $|f_0(x') - f_1(x')| > 11\zeta$ and sample $\alpha \sim \mathscr{I}_\zeta$ uniformly.[5]

   (e) Let $M_k := (x', \alpha) \in \mathcal{X} \times [0, 1]$. If $|\alpha - f_0(x')| < |\alpha - f_1(x')|$, output $S^{(1)} \circ B^{(1)} \circ M_k$, else output $S^{(0)} \circ B^{(0)} \circ M_k$

---

**Intuition of the algorithm.** We first explain Algorithm 2 on an intuitive level. Recall the goal: using our RSOA online learning algorithm for $\mathcal{C}$, we would like to design a *globally stable* PAC learner for $\mathcal{C}$. To this end, let $\mathcal{D}$ be the unknown distribution (under which we need the PAC learner to work).

Algorithm 2 'tailors' a sample (fed to the online learner) as follows: in the $k$th iteration it repeatedly draws pairs of batches of $(k - 1)(m + 1)$ examples from $\text{ext}(D, k - 1)$ and then decides whether to keep or discard each batch based on the outcome of running RSOA on the batches. If some batch is kept, it is appended with a *single* example which is guaranteed to force a mistake on RSOA, and the resulting sample $S$ is output by the algorithm. This process of outputting $S$ can be regarded as drawing sample $S$ from the distribution $\text{ext}(D, k)$. The structure of $S$ is illustrated in Figure 2. Each $B_i$ is a block of $m$ examples each drawn i.i.d. from $D$. Each $M_i = (x_i, \alpha_i)$, forces a mistake when $S$ is fed to RSOA. $S$ has $k$ blocks and $k$ mistake examples in total.

We now focus on explaining steps $2(i)$ to $2(v)$ which 'force a mistake'. In step $2(i)$ we draw two examples, $S^{(0)} \circ B^{(0)}$ and $S^{(1)} \circ B^{(1)}$. In $2(ii)$, we feed $S^{(0)} \circ B^{(0)}$ into RSOA, which returns

---

[5]Recall the definition of the $\zeta$-cover, $\mathscr{I}_\zeta = \{\zeta/2, 3\zeta/2, \ldots, 1 - \zeta/2\}$

Figure 2: Structure of curated sample $S$ obtained resulting from Algorithm 2. Each $B_i$ is a block of $m$ examples $(x, c(x))$ where $x \sim D$ and $M_i = (x, b)$ is an example which forces a *mistake*.

function $f_0$, and do the same for $S^{(1)} \circ B^{(1)}$, returning $f_1$. There are now two possibilities, either $f_0, f_1$ are "close" or $f_0$ and $f_1$ differ significantly at some $x \in \mathcal{X}$ and step $2(iii)$ checks which is the case as follows.

1. $f_0, f_1$ agree to within $11\zeta$ on every point in $\mathcal{X}$: then draw a new pair $S^{(0)} \circ B^{(0)}$ and $S^{(1)} \circ B^{(1)}$ afresh, going back to step 2i).

2. $|f_0(x) - f_1(x)| > 11\zeta$ for some $x \in \mathcal{X}$. Note that this $x$ need not be from an example previously given to the learner. Intuitively, in this case, the predictions $f_0$ and $f_1$ are so far apart at $x$ that they cannot both be $5\zeta$-correct, and so at least one of them is a mistake. More precisely, in the $\zeta$-cover, let $b_c \in \mathcal{I}_\varepsilon$ be the midpoint of the bin (of width $\zeta$) that contains $c(x)$. Since $|f_0(x) - f_1(x)| > 11\zeta$, at least one of the predictions $f_0(x), f_1(x)$ is $5\zeta$-far from $b_c$ (though we don't know which it is, since we don't know $c$!)

Steps $2(i)$ to $2(iii)$ are repeated until we are in the second case. Note that steps $2(i)$ to $2(iii)$ could be repeated an unbounded number of times, each repetition drawing fresh examples. For the remainder of this section, we assume that steps $2(i)$ to $2(iii)$ terminate eventually so that we may argue about the final output sample. In Section C.1.2, we show it suffices to "impose" a cut-off of $T$ examples so that with high probability the algorithm (with an appropriate value of $k$) terminates before drawing $T$-many examples.

In order to create $M_k$, we uniformly draw some $\alpha \sim \mathcal{I}_\zeta$ (the set of all possible bin midpoints), which means $\alpha = b_c$ with probability $\zeta$.[6] If $\alpha = b_c$, we are guaranteed that $f_i$ is a mistake for $i := \arg\max_i |\alpha - f_i(x)|$. Therefore, we concatenate our mistake example with $S^{(i)} \circ B^{(i)}$, eventually outputting $S := S^{(i)} \circ B^{(i)} \circ (x, \alpha)$ as the output of Algorithm 2. By the end of these steps, we will have a sample $S' \circ B' \circ M_k$ where $S' \sim \text{ext}(D, k-1)$, $B' \sim D^m$ and $M_k$ is a single 'mistake' example with the following two properties: (i) $M_k = (x', \alpha)$ is a valid example (i.e., $|\alpha - c(x')| \le \zeta$). (ii) If RSOA is fed $S' \circ B' \circ M_k$, RSOA will make a mistake upon seeing the example $M_k$, i.e., at the round corresponding to $M_k$, RSOA predicts $\hat{y}$ such that $|\hat{y} - c(x')| > 5\zeta$.

**Key Lemma.** We now prove our key lemma on global stability. Let $\mathcal{R}$ be the set of all possible functions that could be output by the RSOA algorithm when run for arbitrarily many rounds.

**Lemma C.3** (Some function ball is output by RSOA with high probability). *Let* $\text{sfat}_{2\zeta}(\mathcal{C}) = d$. *There exists $k \le d$ and some $f \in \mathcal{R}$ such that*

$$\Pr_{\substack{S \sim \text{ext}(D,k), \\ B \sim D^m}}[\text{RSOA}_\zeta(S \circ B) \in \mathcal{T}(5\zeta, f)] \ge \zeta^d. \tag{10}$$

*Proof.* Towards contradiction, suppose for every $k \le d$ and $f \in \mathcal{R}$, we have

$$\Pr_{\substack{S \sim \text{ext}(D,d), \\ B \sim D^m}}[\text{RSOA}_\zeta(S \circ B) \in \mathcal{T}(5\zeta, f)] < \zeta^d. \tag{11}$$

In particular, Eq. (11) holds for $f = c$ where $c$ is the target concept.

In Step $2(iv)$, Algorithm 2 picks $\alpha$ uniformly from the set of midpoints in $\mathcal{I}_\zeta$. Call a mistake example $(x, \alpha)$ 'valid' if $|\alpha - c(x)| \le \zeta$. Notice there are actually two midpoints in $\mathcal{I}_\zeta$ which are less than $\zeta$

---

[6]Note that this step crucially differs from [7] since for them the true value of $f_0(x)$ or $f_1(x)$ is always 0 or 1, so they can flip a coin and force a mistake with probability at least $1/2$.

away from any $c(x)$, and hence, the probability that a mistake example is valid is $2\zeta > \zeta$. Hence the probability that all $d$ mistake examples are valid is at least $\zeta^d$. In the event that all mistake examples are valid, $S$ is a valid sample. Since $S$ contains $d$ mistake examples, and Theorem B.4 guarantees that $\mathsf{RSOA}_\zeta$ on a valid sample always outputs some hypothesis function in $\mathcal{T}(5\zeta, c)$ after making $d$ mistakes, this contradicts Eq. (11). $\qquad\square$

**Lemma C.4** (Generalization). *Let $\mathsf{ext}(D, \ell)$ be such that $\ell \geq 1$ and there exists $f$ such that*

$$\Pr_{\substack{S \sim \mathsf{ext}(D,\ell), \\ B \sim D^m}} [\mathsf{RSOA}_\zeta(S \circ B) \in \mathcal{T}(5\zeta, f)] \geq \zeta^d. \tag{12}$$

*(The above property is the analog of the distribution $\mathsf{ext}(D, \ell)$ being 'well-defined' in [7].)*

*Then, every $f$ satisfying Eq. (12) also satisfies $\mathsf{Loss}_D(f, c, 6\zeta) \leq d \ln(1/\zeta)/m$.*

*Proof.* Let $S \sim \mathsf{ext}(D, \ell)$ and $B \sim D^m$. Suppose $\mathsf{RSOA}_\zeta(S \circ B)$ outputs a function $f' \in \mathcal{T}(5\zeta, f)$. Now, for $f' \in \mathcal{R}$, let $E_{f'}$ be the event that $\mathsf{RSOA}_\zeta(S \circ B)$ outputs $f'$. Then observe that

$$\begin{aligned}
\Pr_{\substack{S \sim \mathsf{ext}(D,\ell), \\ B \sim D^m}} [\mathsf{RSOA}_\zeta(S \circ B) \in \mathcal{T}(5\zeta, f)] &= \sum_{f': f' \in \mathcal{T}(5\zeta, f)} \Pr_{\substack{S \sim \mathsf{ext}(D,\ell), \\ B \sim D^m}} [E_{f'}] \\
&\leq \sum_{f': f' \in \mathcal{T}(5\zeta, f)} \Pr_{\substack{S \sim \mathsf{ext}(D,\ell), \\ B \sim D^m}} [B \text{ is } \zeta\text{-consistent with } f'] \\
&\leq \Pr_{\substack{S \sim \mathsf{ext}(D,\ell), \\ B \sim D^m}} [B \text{ is } 6\zeta\text{-consistent with } f],
\end{aligned} \tag{13}$$

where the first inequality follows from combining two observations:

1. Since $B$ is a subset of the examples fed to $\mathsf{RSOA}_\zeta$, by Property 1 in Lemma B.3, if $\mathsf{RSOA}_\zeta(S \circ B)$ outputs $f'$ then $f'$ is $\zeta$-consistent with all $m$ examples in $B$;

2. By Property 4 of Lemma B.3 (for a fixed sample, no two different functions can be output by RSOA), $\{E_{f'}\}_{f' \in \mathcal{R}}$ are disjoint on the sample space;

and the last inequality used that $f'$ is in a $5\zeta$-ball of $f$, hence $f$ is $\zeta + 5\zeta = 6\zeta$ consistent with $B$. Recall that Eq. (12) shows that the LHS of Eq. (13) is lower-bounded by $\zeta^d$. If we define $\mathsf{Loss}_D(f, c, 6\zeta) := \alpha$, then by the definition of loss, since $B$ is a sample of $m$ i.i.d. examples drawn from $D$, the RHS of the inequality above is $(1 - \alpha)^m$. Putting together the lower and upper bound $\zeta^d \leq (1 - \alpha)^m \leq e^{-\alpha m}$, proves the lemma statement. $\qquad\square$

#### C.1.2 A Monte Carlo version of the tailoring algorithm

Algorithm 2 that we described in the previous section could potentially run steps $(i) - (iii)$ forever. Apriori it is not clear why this algorithm terminates. In this section, we compute the expected number of examples drawn by Algorithm 2 and eventually use Markov's inequality to define a "stopping criterion" (a sample complexity cutoff) on Algorithm 2 so that the algorithm eventually stops drawing a certain number of examples. The reason the number of examples drawn is a random variable is that steps $2(i)$ to $2(iii)$ of Algorithm 2 must be repeated until there is one round where $f_0, f_1$ are distance more than $11\zeta$ apart, i.e., there exists $x \in \mathcal{X}$ satisfying $|f_0(x) - f_1(x)| > 11\zeta$.

**Lemma C.5** (Expected number of examples drawn in Steps $2(i)$ to $2(iii)$). *Let $\zeta \in [0, 1/2]$ and let $k^*$ be the smallest value (guaranteed to exist by Lemma C.3) for which*

$$\Pr_{\substack{S \sim \mathsf{ext}(D,k^*), \\ B \sim D^m}} [\mathsf{RSOA}_\zeta(S \circ B) \in \mathcal{T}(11\zeta, f)] \geq \zeta^d \tag{14}$$

*holds. Let $\ell \leq k^*$ and $M_\ell$ denote the number of examples drawn from $D$ in order to generate a sample $S \sim \mathsf{ext}(D, \ell)$. Then*

$$\mathbb{E}[M_\ell] \leq 4^{\ell+1} \cdot m,$$

*where the expectation is taken over the random sampling process in Algorithm 2.*

*Proof.* Because we have chosen $k^*$ to be the smallest value for which Eq. (14) is true, this implies that for every $\ell' < k^*$ and $f \in \mathcal{R}$, we have

$$\Pr_{\substack{S \sim \text{ext}(D, \ell'), \\ B \sim D^m}} [\text{RSOA}_\zeta(S \circ B) \in \mathcal{T}(11\zeta, f)] < \zeta^d$$

which is equivalent to

$$\Pr_{\substack{S \sim \text{ext}(D, \ell'), \\ B \sim D^m}} [\text{RSOA}_\zeta(S \circ B) \notin \mathcal{T}(11\zeta, f)] \geq 1 - \zeta^d.$$

Now consider sampling from $\text{ext}(D, \ell)$ such that $0 \leq \ell \leq k^*$. Call each round of $2(i)$ to $2(iii)$ 'successful' if it results in $f_0$, $f_1$ such that $|f_0(x) - f_1(x)| > 11\zeta$ for some $x$. Upon success, the algorithm proceeds to step $2(iv)$. Let us assume that the probability of success for the $\ell$th round is $\theta$. Then one can express $\theta$ as follows:

$$\theta = \sum_{f_0 \in \mathcal{R}} \Pr_{\substack{S_0 \sim \text{ext}(D, \ell-1), \\ B_0 \sim D^m}} [\text{RSOA}(S_0 \circ B_0) = f_0] \cdot \Pr_{\substack{S_1 \sim \text{ext}(D, \ell-1), \\ B_1 \sim D^m}} [\text{RSOA}(S_1 \circ B_1) = f_1, \; f_1 \notin \mathcal{T}(11\zeta, f_0)]$$

$$\geq (1 - \zeta^d) \sum_{f_0 \in \mathcal{R}} \Pr_{\substack{S_0 \sim \text{ext}(D, \ell-1), \\ B_0 \sim D^m}} [\text{RSOA}(S_0 \circ B_0) = f_0] = 1 - \zeta^d,$$

where the first equality is because 'success' is defined as $|f_0(x) - f_1(x)| > 11\zeta$ at some $x$, equivalently $f_1 \notin \mathcal{T}(11\zeta, f_0)$, and we used Eq. (C.1.2) in the inequality.

Furthermore, sampling from $\text{ext}(D, \ell)$ involves sampling from $\text{ext}(D, \ell - 1), \ldots, \text{ext}(D, 0)$. Therefore, the number of examples drawn to sample from $\text{ext}(D, \ell)$, $M_\ell$, is a function of $M_{\ell-1}, \ldots, M_0$. Let $M_\ell^{(j)}$ be the number of examples drawn during the $j$th attempt at sampling from distribution $\text{ext}(D, \ell)$ and write $M_\ell = \sum_{j=1}^{\infty} M_\ell^{(j)}$. While sampling from distribution $\text{ext}(D, \ell)$, if we succeed prior to the $j$-th attempt, $M_\ell^{(j)} = 0$; otherwise, if the first $j - 1$ attempts end in failure, we have to draw two examples from $\text{ext}(D, \ell - 1)$ and two examples from $D^m$. Therefore, we may define the recursive equation

$$\mathbb{E}\left[M_\ell^{(j)}\right] = (1 - \theta)^{j-1} \cdot (2\mathbb{E}[M_{\ell-1}] + 2m), \tag{15}$$

since each attempt involves drawing two examples from $\text{ext}(D, \ell - 1)$ and two examples from $D^m$ and we used the fact that the probability of failure is $(1 - \theta)^{j-1}$. Therefore, we have

$$\begin{aligned}
\mathbb{E}[M_\ell] = \sum_j \mathbb{E}\left[M_\ell^{(j)}\right] &= \sum_{j=1}^{\infty} (1 - \theta)^{j-1} \cdot (2\mathbb{E}[M_{\ell-1}] + 2m) \\
&= \frac{1}{\theta} \cdot (2\mathbb{E}[M_{\ell-1}] + 2m) \\
&\leq \frac{1}{1 - \zeta^d} \cdot (2\mathbb{E}[M_{\ell-1}] + 2m) \leq 4 \cdot (\mathbb{E}[M_{\ell-1}] + m),
\end{aligned} \tag{16}$$

where we have used the fact that $\zeta < 1/2$ to obtain the last inequality. Using that $\mathbb{E}[M_0] = 0$ and using induction on Eq. (16) gives us the lemma statement. $\qquad \square$

### C.1.3 Final algorithm

Putting together these pieces, we now prove our main theorem.

**Theorem C.6** (Globally stable learner from online learner)**.** *Let $\alpha > 0$. Let $\mathcal{C} \subseteq \{f : \mathcal{X} \to [0, 1]\}$ be a concept class with $\text{sfat}_{2\zeta}(\mathcal{C}) = d$. Let $c \in \mathcal{C}$ be the target concept. Let*

$$T = \left(2 \cdot (4/\zeta)^{d+1} + 1\right) \cdot \frac{d \ln(1/\zeta)}{\alpha}.$$

*Let $D : \mathcal{X} \to [0, 1]$ be a distribution. There exists a randomized algorithm $G : (\mathcal{X} \times [0, 1])^T \to [0, 1]^{\mathcal{X}}$ that satisfies the following: given $T$ many examples $S = \{(x_i, \widehat{c}(x_i))\}$ where $x \sim D$, there exists a hypothesis $f$ such that*

$$Pr[G(S) \in \mathcal{T}(11\zeta, f)] \geq \frac{\zeta^d}{2(d+1)} \text{ and } \text{Loss}_D(f, c, 12\zeta) \leq \alpha \tag{17}$$

---

**Algorithm 3** Final globally-stable algorithm $G$ to learn concept class $\mathcal{C} \subseteq \{f : \mathcal{X} \to [0,1]\}$.

---

1. Draw $k \in \{0, 1, \ldots, d\}$ uniformly at random.

2. Let $\mathsf{ext}(D, k)$ be the distribution described in Algorithm 2 but additionally imposing a cutoff $T$ on sample complexity (i.e. we output 'fail' if the number of examples drawn in sampling from $\mathsf{ext}(D, k)$ ever exceeds $T$), where the auxiliary sample size is set to $m = d\ln(1/\zeta)/\alpha$ and cutoff $T = 2 \cdot (4/\zeta)^{d+1} \cdot m$.[7]

   Let $B \sim D^m$ and $S \sim \mathsf{ext}(D, k)$ and output $h = \mathsf{RSOA}_\zeta(S \circ B)$.

---

*Proof.* The algorithm $G$ in the theorem statement is exactly the algorithm we defined in the previous two sections along with a cutoff at $T$ examples.

Note that because we have enforced the cutoff at $T$ examples in drawing $S \sim \mathsf{ext}(D, k)$, the sample complexity of $G$ is $|S| + |B| \leq T + m = \left(2 \cdot (4/\zeta)^{d+1} + 1\right) \cdot \frac{d\ln(1/\zeta)}{\alpha}$ as stated in the theorem statement. Lemma C.3 guarantees that there exists $k \leq d$ and $f^*$ such that Eq. (12) holds. Let $k^*$ be the smallest $k$ such that Lemma C.3 holds with the constant $5\zeta$ replaced by $11\zeta$, and

$$\Pr_{\substack{S \sim \mathsf{ext}(D, k^*), \\ B \sim D^m}} [\mathsf{RSOA}_\zeta(S \circ B) \in \mathcal{T}(11\zeta, f^*)] \geq \zeta^d. \tag{18}$$

Then Lemma C.4 (with a simple modification for the new constant) implies that $\mathsf{Loss}_D(f, c, 12\zeta) \leq d\ln(1/\zeta)/m \leq \alpha$.

We now show that the probability that $G$ outputs some function in $\mathcal{T}(11\zeta, f^*)$ is $\frac{1}{2(d+1)} \cdot \zeta^d$. Firstly, with probability $\frac{1}{d+1}$, the randomly drawn $k$ in step 2 is $k^*$. Conditioned on this, we now show that with high probability, the loop in Steps $2(i)$ to $2(iii)$ will terminate after drawing $T = 2 \cdot (4/\zeta)^{d+1} \cdot m$ examples.

$$\Pr\left[M_{k^*} > 2 \cdot (4/\zeta)^{d+1} \cdot m\right] \leq \Pr\left[M_{k^*} > 2 \cdot \zeta^{-d} \cdot 4^{k^*+1} \cdot m\right] \leq \zeta^d/2, \tag{19}$$

where the first inequality used $k^* \leq d$ and the second inequality is by Markov's inequality and Lemma C.5. Putting together Eq. (18) and (19) the probability that $\mathsf{RSOA}(S \circ B)$ outputs a function in $\mathcal{T}(11\zeta, f^*)$ and also Algorithm 2 terminates before the cutoff $T$ is

$$\Pr_{\substack{S \sim \mathsf{ext}(D, k^*), \\ B \sim D^m}} \left[\mathsf{RSOA}(S \circ B) \in \mathcal{T}(11\zeta, f^*) \text{ and } M_{k^*} \leq 2 \cdot (4/\zeta)^{d+1} \cdot m\right] \geq \zeta^d - \zeta^d/2 = \zeta^d/2$$

$$\tag{20}$$

Multiplying this together with $1/(d+1)$ yields our claim. □

### C.2 Quantum stability does not imply quantum approximate DP (without a domain-size dependence)

In the previous section we showed that if a concept class $\mathcal{C}$ can be learned in the quantum online learning framework, then there exists a globally stable learner (with appropriate parameters) for $\mathcal{C}$ as well. This implication was first pointed out by [7] for Boolean-valued $\mathcal{C}$s. In fact, they went one step further and created a *approximately differentially-private* learner from a stable learner. In this sense, stability can be viewed as an intermediate property between online learnability and approximate differential privacy in the Boolean setting. Jung et al. [4] used the same technique to show that stability implies approximate differential privacy in the *multiclass* learning setting as well (i.e., when the concept class to be learned maps to a discrete set $\{1, \ldots, k\}$), but they do not show that an analogous implication holds for real-valued learning, which they mention briefly. Note that their real-valued learning setting is less general than ours, as they assume that they receive exact feedback on each example (we discuss this at the end of this section).

A natural question is: does this result still hold in the quantum learning setting, i.e. does quantum stability imply quantum differential privacy? In this section, we show that the [7] method for showing

---

[7]For simplicity in notation, we assume $cd/\alpha$ is an integer. If not, one can set $m = \lceil cd/\alpha \rceil$.

this implication for Boolean functions – which held up in the case of learning multiclass functions – fails for learning real-valued functions with imprecise feedback. Unlike in the former two cases, the transformation from stable learner to approximate DP learner necessarily incurs a domain-size dependence in the sample complexity. This is undesirable because, when $\mathcal{X}$ is a real-interval or if it is unbounded, this quantity could potentially be infinite.

### C.2.1 Lower-bounding the sample complexity of the stability $\rightarrow$ privacy transformation

In the Boolean setting, [7] showed that one could use the stable histograms algorithm [14] and the Generic Private Learner of [11], to convert a Boolean globally-stable learner, in a black-box fashion, to a private learner. This learner's sample complexity depends on $\mathsf{Ldim}(\mathcal{C})$ and the privacy and accuracy parameters of the stable learner, but *not* the domain size of the function class. We now show that this technique cannot possibly yield a domain size-independent sample complexity for quantum learning.

Our stable learner $G$ has the following guarantees (given in Theorem C.6): there exists some *function ball* (around the target concept) such that the collective probability of $G$ outputting its member functions is high. Contrast this with the global stability guarantee for learning Boolean functions [7], which says that $G$ outputs some *fixed function* with high probability. The stability guarantees differ because, in our setting, the learner only obtains $\varepsilon$-accurate feedback from the adversary. Hence the learner cannot uniquely identify the target concept $c$, since all functions that are in the $\varepsilon$-ball of $c$ would be consistent with the feedback of the adversary, and we thus allow the learner to output a function in the $\varepsilon$-ball around the target concept. However, this difference critically prevents us from using the [7] technique to transform a stable learner into a private learner in the quantum case. We sketch this argument below[8], which relies on ideas from classical fingerprinting codes [16] (which were also used earlier by Aaronson and Rothblum [17] in order to give lower bounds on gentle shadow tomography).

[7]'s transformation from stable learner to private learner, applied to our setting, would be as follows: generate a list of functions in $\mathcal{C}$ by running the stable learner $G(S)$ of Theorem C.6, $n$ many times, each of which outputs a single $f_i \in \mathcal{C}$. By Theorem C.6 and a Chernoff bound, one can show that with high probability, an $\eta = \zeta^d$-fraction of the list should be in $\mathcal{T}(\zeta, f^*)$ for some $f^*$. Next one would like to *privately* output some function in $\mathcal{T}(\zeta, f^*)$. We now cast this in terms of the following problem:

**Problem C.7** (Query release for function balls). *Given a list of $n$ functions $\{f_i : \mathcal{X} \to \mathbb{R}\}_{i \in [n]}$, an $\eta$-fraction of which are in $\mathcal{T}(\zeta, f^*)$ for some $f^* : \mathcal{X} \to \mathbb{R}$, output some function $g \in \mathcal{T}(\zeta, f^*)$.*

We could also consider the following problem of clique identification on a discrete domain.

**Problem C.8** (Clique identification on a discrete domain ). *Given a symmetric, reflexive relation $R \subseteq \mathcal{Y} \times \mathcal{Y}$ and a dataset $D \in \mathcal{Y}^n$ under the promise that $(x, y) \in R$ for every $x, y \in D$, find any point $z \in \mathcal{Y}$ such that $(x, z) \in R$ for every $x \in D$.* Clique identification on a discrete domain *is* clique identification *with $\mathcal{Y} = [4]^d$ and $R = \{(x, y) \in \mathcal{Y} \times \mathcal{Y} : \|x - y\|_\infty \leq 1\}$.*

Problem C.8 reduces to Problem C.7. To see this, note that when we choose the functions $f$ in Problem C.7 to be of the form $f : [d] \to [4]$, $\eta = 1$ and $\zeta = 1/2$, and let $D$ consist of the $n$ vectors $[f_i(1), \ldots f_i(d)]$, $i \in [n]$, we recover Problem C.8. Hence, any DP algorithm for query release for function balls is also a DP algorithm for clique identification on a discrete domain. However, we claim the following:

**Claim C.9.** *For $\delta < 1/500$, any $(1, \delta < 1/n)$-DP algorithm[9] solving Problem C.8 with probability at least $1499/1500$ requires $n \geq \tilde{\Omega}(\sqrt{d})$.*

We will prove the claim later, but we first explain why it implies a necessary domain size dependence in the transformation we hope to achieve. Noting that $d = |\mathcal{X}|$ in the translation from Problem C.7 to Problem C.8, we conclude from Claim C.9 that any $(1, \delta)$-DP algorithm for Problem C.7 requires $n \geq \tilde{\Omega}(\sqrt{|\mathcal{X}|})$. Hence, any algorithm to convert the stable real-valued learner $G$ of Theorem C.6 into an approximate-DP learner that also solves Problem C.7, also requires to run the stable learner

---

[8]The following argument was communicated to us by Mark Bun [15].

[9]It is not hard to modify this proof so as to allow an $\varepsilon$ privacy parameter.

611    $n$-many times, each of which consumes $T$ examples. Hence the total number of examples needed is

$$\tilde{\Omega}\left(\sqrt{|\mathcal{X}|}\left(2 \cdot (4/\zeta)^{d+1} + 1\right) \cdot \frac{d\ln(1/\zeta)}{\alpha}\right). \tag{21}$$

612    In particular, this lower bound is also optimal for query release up to poly-logarithmic factors, i.e.,
613    using $\tilde{O}(\sqrt{|\mathcal{X}|})$ examples one can solve Problem C.7 using the Private Multiplicative Weights method
614    by Hardt and Rothblum [18] (as also referenced in the work of Bun et al. [16]).

615    To prove Claim C.9, we first need to first define weakly-robust fingerprinting codes (first introduced
616    by Boneh and Shaw [19], then developed in [16]).

617    **Definition C.10.** *An $(n, d)$-fingerprinting code with security $s$ and robustness $r$ is a pair of random*
618    *variables $(G, T)$ where $G \in \{2, 3\}^{n \times d}$ and $T : \{2, 3\}^d \to 2^{[n]}$ that satisfy the following. We say*
619    *that a column $j \in [d]$ is marked if there exists $b \in \{2, 3\}$ such that $x_{i;j} = b$ for all $i \in [n]$. Similarly,*
620    *we say a string $w \in \{2, 3\}^d$ is feasible for $G$ if for at least a $1 - r$ fraction of the marked columns*
621    *$j \in G$, the entry $w_j$ agrees with the common value in that column. Moreover, we need the notion of*
622    *soundness and completeness as follows:*

- **Completeness** *For every $A : \{2, 3\}^{n \times d} \to \{2, 3\}^d$, we have*
  $\Pr_{w \leftarrow A(G)}[w \text{ is feasible for } G \text{ and } T(w) = \emptyset] \leq s$

- **Soundness** *For every $i \in [n]$ and algorithm $A : \{2, 3\}^{n \times d} \to \{2, 3\}^d$, we have*
  $\Pr_{w \leftarrow A(G_{-i})}[T(w) \ni i] \leq s$

627    We will also need the following theorem by [20] who gave explicit construction of fingerprinting
628    codes.

629    **Theorem C.11** ([20]). *Then, for every $s \in (0, 1)$, there exists an $(n, d)$-fingerprinting code with*
630    *security $s$ and robustness $r = 1/25$ with $d = \tilde{O}(n^2 \log(1/s))$.*

631    With this we now prove our main claim.

632    *Proof of Claim C.9.* The idea is to construct, from any $(\varepsilon = 1, \delta = 1/4n)$-DP clique identification
633    algorithm with success probability at least $1499/1500$, an adversary $A : \{2, 3\}^{n \times d} \to \{2, 3\}^d$ for
634    any $(n, d)$ fingerprinting code with robustness $1/25$, such that the code cannot be $1/20n$-secure
635    against the adversary. However, because Theorem C.11 guarantees the existence of a sound and
636    complete $(n, d)$-fingerprinting code with $(s = 1/20n, r = 1/25)$-parameters as long as $n < \tilde{\Omega}(\sqrt{d})$,
637    the claimed clique identification algorithm $M$ must have $n \geq \tilde{\Omega}(\sqrt{d})$. We now go into more detail
638    about how to construct the adversary.

Let $M$ be the alleged DP algorithm for clique identification, and let $G \in \{2, 3\}^{n \times d}$ be the $G$
corresponding to the fingerprinting code. If we regard each of the rows of $G$ as being a point in
$\mathcal{Y} = [4]^d$, then taking $D$ to be the set of all rows of $G$, $D$ fulfils the promise of Problem C.8.
Then the adversary $A$ is constructed out of $M$ as follows: on input $D$, run $M(D)$ producing a
string $w \in [4]^d$. Return the string $w' \in \{2, 3\}^d$ where $w_i' = 2$ if $w_i \in \{1, 2\}$ and $w_i' = 3$ if
$w_i \in \{3, 4\}$. A proof by contradiction, which we omit, shows that the string $w'$ produced in this
manner is feasible for the fingerprinting code with probability at least $2/3$. By completeness of the
code, $\Pr[T(A(D)) \in [n]] \geq 2/3 - s \geq 1/2$. In particular, there exists some $i^* \in [n]$ such that
$\Pr[T(A(D)) = i^*] \geq 1/2n$. Now by differential privacy,

$$\Pr[T(A(D_{-i^*})) = i^*] \geq e^{-\varepsilon}(\Pr[T(A(D)) = i^*] - \delta) \geq e^{-1}\left(\frac{1}{2n} - \frac{1}{4n}\right) \geq \frac{1}{20n}.$$

639    This contradicts the soundness of the code.      □

### 640   C.2.2   A quadratically worse upper bound on the sample complexity of privacy

641    The previous subsection showed that going from a stable learner to a private learner of real-valued
642    function classes should incur a sample complexity at least the root of domain size. Now we mention
643    an explicit algorithm for pure-DP learning real-valued function classes over a finite domain – with no

need for the stability intermediate step – that needs at most linear-in-$|\mathcal{X}|$ examples. (This was also pointed out in the Appendix of Jung et al. [4].)

The private algorithm that accomplishes this is the Generic Private Learner of [11, 7]. We give its guarantees in the lemma below. Intuitively, this lemma states that given a collection of hypotheses, one of which is guaranteed to have low loss $\alpha$ with respect to some unknown distribution and target concept, by adding Laplace noise, one can *privately* output with high probability a hypothesis with loss at most $2\alpha$ with respect to the unknown target concept and distribution.

**Lemma C.12** (Generic Private Learner [11, 7]). *Let $\mathcal{H} \subseteq \{h : \mathcal{X} \to [0, 1]\}$ be a set of hypotheses. For*

$$m = O\left(\frac{\log |\mathcal{H}|}{\alpha\varepsilon}\right)$$

*there exists an $(\varepsilon, 0)$-differentially private generic learner $\mathsf{GL} : (\mathcal{X} \times [0, 1])^m \to \mathcal{H}$ such that the following holds. Let $D : \mathcal{X} \times [0, 1] \to [0, 1]$ be a distribution, $c : \mathcal{X} \to [0, 1]$ be a target function, $\zeta$ be a distance parameter and $h^* \in \mathcal{H}$ be such that with $\mathsf{Loss}_D(h^*, c, \zeta) \leq \alpha$. Then on input $S \sim D^m$, algorithm $\mathsf{GL}$ outputs, with probability at least $2/3$, a hypothesis $\hat{h} \in \mathcal{H}$ such that $\mathsf{Loss}_D(\hat{h}, c, \zeta) \leq 2\alpha$.*

For every real-valued function class $\mathcal{C}$, one could discretize the $[0, 1]$-range of its functions $h : \mathcal{X} \to [0, 1]$ into bins of size $\zeta$. This obtains a discretized function class $\mathcal{H}$ with at most $(1/\zeta)^{|\mathcal{X}|}$ functions. Plugging this bound into the lemma above, we obtain a private learner with sample complexity

$$m = O\left(\frac{|\mathcal{X}| \log(1/\zeta)}{\alpha\varepsilon}\right). \tag{22}$$

## C.3  The quantum implications

We now turn to the quantum implications of the results in the previous sections. While we have stated all our results for the case of learning real-valued functions with imprecise adversarial feedback, we now expressly translate them to the setting of learning quantum states. Recall that, as stated in Section A, in quantum learning we are given $\mathcal{U}$, a class of $n$-qubit quantum states from which the state to be learned is drawn; $\mathcal{M}$, a set of 2-outcome measurements and $D : \mathcal{M} \to [0, 1]$, a distribution on the set of measurements.[10] Our results apply to quantum learning by associating, to every $\rho \in \mathcal{U}$, the real-valued function $c_\rho : \mathcal{M} \to [0, 1]$ defined as $c_\rho(M) = \mathsf{Tr}(M\rho) \in [0, 1]$ for every $M \in \mathcal{X}$, and taking the function class to be $\mathcal{C}_\mathcal{U} = \{c_\rho\}_{\rho \in \mathcal{U}}$.

Section C.1 implies that given a $\mathcal{C}_\mathcal{U}$ with bounded sfat dimension, a stable learner for $\mathcal{C}_\mathcal{U}$ also exists. To translate this result into the quantum learning setting, we define quantum stability as follows:

**Definition C.13** (Quantum stability). *A quantum learning algorithm $\mathcal{A} : (\mathcal{M} \times [0, 1])^T \to \mathcal{U}$ is $(T, \varepsilon, \eta)$-stable with respect to distribution $D : \mathcal{M} \to [0, 1]$ if, given $T$ many labelled examples $S = \{(E_i, y_i)\}_{i \in [T]}$ where $|\mathsf{Tr}(\rho E_i) - y_i| < \zeta$, there exists a state $\sigma$ such that*

$$\Pr[\mathcal{A}(S) \in \mathcal{B}_\mathcal{M}(\varepsilon, \sigma)] \geq \eta, \tag{23}$$

*where the probability is taken over the examples in $S$ and $\mathcal{B}_\mathcal{M}(\varepsilon, \sigma) := \{\rho : |\mathsf{Tr}(E\rho) - \mathsf{Tr}(E\sigma)| \leq \varepsilon\}$, that is to say, the ball of states within distance $\varepsilon$ of $\sigma$ on $\mathcal{M}$.*

In other words, quantum stability means that up to an $\varepsilon$-distance on the measurements in $\mathcal{M}$, there is some $\sigma$ that is output by $\mathcal{A}$ with "high" (at least $\eta$) probability. Then the quantum version of Theorem C.6 is the following:

**Theorem C.14** (Quantum-stable learner from online learner). *Let $\mathcal{U}$ be a class of quantum states with $\mathsf{sfat}_{2\zeta}(\mathcal{C}_\mathcal{U}) = d$, let $\mathcal{M}$ be a set of orthogonal 2-outcome measurements and let $D : \mathcal{M} \to [0, 1]$ be a distribution over measurements. There exists an algorithm $\mathcal{G} : (\mathcal{M} \times [0, 1])^T \to \mathcal{U}$ that satisfies the following: for every $\rho \in \mathcal{U}$, given*

$$T = \left(2 \cdot (4/\zeta)^{d+1} + 1\right) \cdot \frac{d \ln(1/\zeta)}{\alpha}.$$

---

[10]To be more clear, $D$ can be viewed as a distribution over $\{(E_i, \mathbb{I} - E_i)\}_i$ where $\{E_i\}_i$ is an orthogonal basis for the space of operators on $n$-qubits satisfying $\|E_i\| \leq 1$.

682    *many labelled examples $S = \{(E_i, y_i)\}_{i \in [T]}$ where $|\mathsf{Tr}(\rho E_i) - y_i| < \zeta$ and $E_i \sim D$, there exists a $\sigma$*

683    *such that $\Pr_{S \sim D^T}[\mathcal{G}(S) \in \mathcal{B}_{\mathcal{M}}(11\zeta, \sigma)] \geq \frac{\zeta^d}{2(d+1)}$ and $\Pr_{E \sim D}\left[|\mathsf{Tr}(\rho E) - \mathsf{Tr}(\sigma E)| \leq 12\zeta\right] \geq 1 - \alpha.$*

684    Namely, $\mathcal{G}$ is $(T, 11\zeta, \frac{\zeta^d}{2(d+1)})$-stable and furthermore, the state $\sigma$ at the center of its 'output ball' has

685    loss $\alpha$.

686    Section C.2 now gives a no-go result for going from the above-mentioned quantum-stable learner to

687    an approximate-DP one. It shows that the technique of [7] to convert a stable learner to a private one

688    necessarily incurs a domain-size dependence in the sample complexity.

689    We say a few words about the implications of this on quantum learning. As explained earlier, it is

690    often of most interest to choose $\mathcal{M}$ to be some orthogonal set of measurements. If, say, we choose it

691    to be the orthogonal basis of $n$-qubit Paulis, then $|\mathcal{M}| = 4^n$ and so Equation (21) implies that one

692    needs sample complexity $\tilde{\Omega}(4^{n/2})$ in order to go from stability to approximate differential privacy,

693    whereas Equation (22) implies that even without stability, there exists a simple (pure) private learner

694    for $\mathcal{C}_{\mathcal{U}}$ whose sample complexity is $\tilde{O}(4^n)$, which is quadratically worse.

695    **Comparison to prior work [4].**   After completion of this work, we were made aware by an

696    anonymous referee of the paper by Jung, Kim and Tewari [4] that extends the work of Bun et al. [7] to

697    two other classical settings – namely, multi-class learning (i.e., when the concept class to be learned $\mathcal{C}$

698    maps to a discrete set $\{1, \ldots, k\}$) and real-valued learning (when $\mathcal{C}$ takes on values in $[-1, 1]$). The

699    latter is relevant, because learning an unknown quantum state $\rho$ amounts to learning the real-valued

700    function $\mathsf{Tr}(\cdot \rho)$. Despite this similarity, our quantum learning setting and resulting analysis differs

701    from theirs in several crucial ways, which we now outline.

702    Firstly, [4]'s notion of stability for learning real-valued functions resembles our definition, *however*

703    in order to prove that online learnability implies stability, they *modify* the definition of Littlestone

704    dimension and use this modified notion in their work (in fact, we couldn't find a version of the paper

705    that spells out the proof that online learnability implies a stable real-valued learner, but this seems

706    implicit from their proof for the multi-class case). In this work, we use the standard notion of $\mathsf{sfat}(\cdot)$

707    – which we also bound in the case of quantum states – and still show this implication. Secondly,

708    for both PAC learning and online learning settings, [4] assume that the feedback received by the

709    learner is *exact*, i.e. for online learning, on input $x$, the adversary produces $c(x) \in [0, 1]$; for PAC

710    learning, the examples are of the form $(x, c(x))$. By contrast, in this work, we only assume that the

711    feedback in all learning models we consider (which includes both these settings) is a $\varepsilon$-approximation

712    of $c(x)$. This generalizes the previous settings and arises from the fact that, in quantum learning,

713    the feedback comes from some quantum estimation process or quantum measurement. Thus, all

714    implications proven in this work are robust to such adversarial imprecision. This imprecision crucially

715    bars the usage of [7]'s technique, developed for Boolean functions, to conclude that quantum stability

716    implies approximate differential private PAC learning with sample complexity independent of domain

717    size. Finally, one important contribution in this paper is to provide the implications of these real-

718    valued results in the quantum setting, for example the connections to shadow tomography, quantum

719    information theory, quantum one-way communication complexity.

## D   Pure differential privacy implies online learnability

721    In this section we will prove the converse direction of the implication we showed in the previous

722    section, namely that DP PAC learnability of a concept class $\mathcal{C}$ implies online learnability of $\mathcal{C}$. To be

723    more precise, we will show that the sample complexity of *pure* DP PAC learning $\mathcal{C}$ is linearly related

724    to the $\mathsf{sfat}(\cdot)$ dimension of $\mathcal{C}$. Combining this with Theorem B.4 implies learnability in the pure DP

725    PAC setting implies online learnability of $\mathcal{C}$ in the strong feedback setting. The implications we will

726    show are summarized in the diagram below:

727    This section is organized as follows. In Section D.1 we show that the sample complexity of pure

728    DP PAC is linearly related to the communication complexity of one-way public communication. As

729    shown in Figure 3, the link between these two notions goes through representation dimension. In

730    Section D.2 we show that one-way communication complexity is, in turn, characterized by $\mathsf{sfat}(\cdot)$.

731    Additionally, we know from Theorem B.4 that this combinatorial dimension upper-bounds the mistake

732    bound of online learning $\mathcal{C}$, and this completes the chain of implications shown in Figure 3.

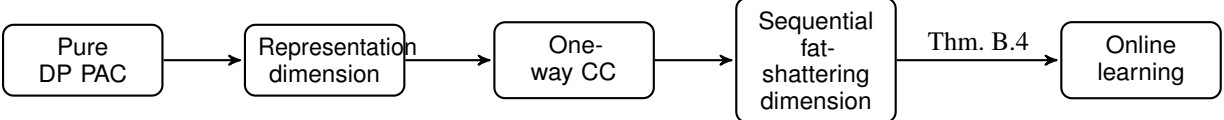

Figure 3: Sample complexity of pure DP PAC upper-bounds $\mathsf{sfat}(\cdot)$.

## D.1 **Pure** DP PAC **implies one-way communication**

In this section we prove that the sample complexity of pure DP PAC learning upper bounds one-way communication complexity of a concept class $\mathcal{C}$.

### D.1.1 **Pure differential privacy and** PRdim

We start by relating the sample complexity of differentially-private PAC (PPAC) learning (see Definition A.5) a concept class $\mathcal{C}$, to the probabilistic representation dimension of $\mathcal{C}$. As in the previous section, we use the shorthand $S \sim D^m$ to mean that the sample $S$ is of the form $\{(x_i, \widehat{c}(x_i))\}_{i=1}^m$ where each $x_i \sim D$ and for all $i$, $\widehat{c}(x_i)$ satisfies $|\widehat{c}(x_i) - c(x_i)| < \zeta/5$.

**Lemma D.1** (Sample complexity of $(\zeta, \alpha, \varepsilon, 0)$-PPAC learning and PRdim). *Let $\alpha < 1/4$. Suppose there exists an algorithm $\mathcal{A}$ that $(\zeta, \alpha, \varepsilon, 0)$-PPAC learns a real-valued concept class $\mathcal{C} \subseteq \{f : \mathcal{X} \to [0,1]\}$ with sample size $m$, then there exists a set of concept classes $\mathscr{H}$ and a distribution over their indices $\mathcal{P}$, such that $(\mathscr{H}, \mathcal{P})$ $(\zeta, 1/4, 1/4)$-probabilistically represents $\mathcal{C}$, with $\mathsf{size}(\mathscr{H}) = O(m\varepsilon\alpha)$. This implies that the sample complexity of $(\zeta, \alpha, \varepsilon, 0)$-PPAC learning $\mathcal{C}$ is*

$$\Omega\left(\frac{1}{\alpha\varepsilon}\mathsf{PRdim}_{\zeta,1/4,1/4}(\mathcal{C})\right). \tag{24}$$

*Proof.* Our proof extends the work of [13] to the case of robust real-valued PAC learning. We assume we are given a $(\zeta, \alpha, \varepsilon, 0)$-PPAC learner $\mathcal{A}$ of $\mathcal{C}$ that outputs some function in hypothesis class $\mathcal{F}$ with sample complexity $m$. The PAC guarantees hold whenever the feedback is a $\zeta/5$ approximation of $c(x_i)$, so for the rest of this proof, we will fix the examples $(x_i, \widehat{c}(x_i))$ to have feedback of the form: $\widehat{c}(x_i) := \lfloor c(x_i) \rfloor_{\zeta/5}$, where $\lfloor\ \rfloor_{\zeta/5}$ denotes rounding to the nearest point in $\mathscr{I}_{\zeta/5}$.

For every target concept $c \in \mathcal{C}$ and distribution $D$ on the input space $\mathcal{X}$, define the following subset of $\mathcal{F}$:

$$G_{D,\zeta}^{\alpha} = \{h \in \mathcal{F} : \mathsf{Loss}_D(h, c, \zeta) \leq \alpha\}, \tag{25}$$

where $\mathsf{Loss}_D(h, c, \zeta) := \Pr_{x \sim D}\left[|h(x) - c(x)| > \zeta\right]$, so $G_{D,\zeta}^{\alpha}$ may be interpreted as a set of probably-$\zeta$-consistent hypotheses in $\mathcal{F}$. In [13], they show that for every distribution $D$, there exists another distribution $\tilde{D}$ on the input space, defined as

$$\tilde{D}(x) = \left\{ \begin{array}{ll} 1 - 4\alpha + 4\alpha \cdot D(x), & x = 0 \\ 4\alpha \cdot D(x), & x \neq 0 \end{array} \right\} \tag{26}$$

(where 0 is some arbitrary point in the domain) which has the property

$$\Pr_{S \sim \tilde{D}^m, \mathcal{A}}\left[\mathcal{A}(S) \in G_{D,\zeta}^{1/4}\right] \geq \frac{3}{4} \tag{27}$$

where $\mathcal{A}(S)$ means $\mathcal{A}$ is fed with the sample $S$. The property in Eq. (27) follows from the fact that $\Pr_{\tilde{D}}[x] \geq 4\alpha \cdot \Pr_D[x]\ \forall x \in \mathcal{X}$ by Eq. (26) which implies $G_{\tilde{D},\zeta}^{\alpha} \subseteq G_{D,\zeta}^{1/4}$, and the assumption that $\mathcal{A}$ is $(\zeta, \alpha)$-PAC which can be re-written as $\Pr_{\tilde{D},\mathcal{A}}[\mathcal{A}(S) \in G_{\tilde{D},\zeta}^{\alpha}] > 3/4$.

Let us now call a sample $S$ 'good' if $\vec{x}$ has at least $(1 - 8\alpha)m$ occurrences of 0. Eq. (27) may be rewritten as

$$\Pr_{S \sim \tilde{D}, \mathcal{A}}\left[\mathcal{A}(S) \in G_{D,\zeta}^{1/4}\right] \tag{28}$$

$$= \Pr_{S \sim \tilde{D}, \mathcal{A}}\left[\mathcal{A}(S) \in G_{D,\zeta}^{1/4} \wedge S \text{ is good}\right] + \Pr_{S \sim \tilde{D}, \mathcal{A}}\left[\mathcal{A}(S) \in G_{D,\zeta}^{1/4} \wedge S \text{ is not good}\right] \geq \frac{3}{4} \tag{29}$$

Letting the random variable $X_S$ denote the number of occurrences of $0$ in $S$, Eq. (26) shows that $\mathbb{E}[X_S] \geq (1 - 4\alpha)m$. With this we upper bound the term $\Pr_{S \sim \tilde{D}, \mathcal{A}}\left[\mathcal{A}(S) \in G_{D,\zeta}^{1/4} \wedge S \text{ is not good}\right]$ by

$$\Pr_{S \sim \tilde{D}, \mathcal{A}}[S \text{ is not good}] = \Pr_{S \sim \tilde{D}, \mathcal{A}}[X_S < (1 - 8\alpha)m] \tag{30}$$

$$= \Pr_{S \sim \tilde{D}, \mathcal{A}}[X_S \leq (1 - \delta)(1 - 4\alpha)m] \leq e^{-\delta^2(1-4\alpha)m/2} = e^{-2\alpha^2 m/(1-4\alpha)}, \tag{31}$$

where the first inequality used $\delta = \frac{4\alpha}{1-4\alpha}$ and the second inequality follows from a Chernoff bound with $\mathbb{E}[X_S]$ replaced with the upper bound $(1 - 4\alpha)m$ on its expectation.

Therefore, one can bound the first term on the right hand side of Eq. (28) by

$$\Pr_{S \sim \tilde{D}, \mathcal{A}}\left[\mathcal{A}(S) \in G_{D,\zeta}^{1/4} \wedge S \text{ is good}\right] \geq \frac{3}{4} - e^{-2\alpha^2 m/(1-4\alpha)} \geq \frac{1}{4}. \tag{32}$$

Eq. (32) implies that there exists *some* sample, $S_{\text{good}}$ such that

$$\Pr_{\mathcal{A}}\left[\mathcal{A}(S_{\text{good}}) \in G_{D,\zeta}^{1/4}\right] \geq \frac{1}{4}. \tag{33}$$

Without loss of generality we may write down $S_{\text{good}}$ as

$$S_{\text{good}} := (\underbrace{(0, \lfloor c(0) \rfloor_{\zeta/5}), \ldots (0, \lfloor c(0) \rfloor_{\zeta/5})}_{k \text{ examples}}, (x_{k+1}, \lfloor c(x_{k+1}) \rfloor_{\zeta/5}) \ldots (x_m, \lfloor c(x_m) \rfloor_{\zeta/5})) \tag{34}$$

for some $k \geq (1 - 8\alpha)m$. Consider an alternative sample, $S_{\text{alt}}$, which takes the form

$$S_{\text{alt}} = (\underbrace{(0, \lfloor c(0) \rfloor_{\zeta/5}), \ldots, (0, \lfloor c(0) \rfloor_{\zeta/5})}_{m \text{ examples}}).$$

$S_{\text{alt}}$ differs from $S_{\text{good}}$ in exactly $m - k < 8\alpha m$ examples, and so by the $\varepsilon$-DP property of $\mathcal{A}$, we have

$$\Pr_{\mathcal{A}}[\mathcal{A}(S_{\text{alt}}) \in G_{D,\zeta}^{1/4}] \geq \exp(-8\alpha\varepsilon m) \Pr_{\mathcal{A}}[\mathcal{A}(S_{\text{good}}) \in G_{D,\zeta}^{1/4}] \geq \frac{1}{4}\exp(-8\alpha\varepsilon m). \tag{35}$$

For the remainder of this proof, we will use Eq. (35) to construct the pair $(\mathcal{H}, \mathcal{P})$. Define the examples

$$S_z = (\underbrace{(0, z), \ldots, (0, z)}_{m \text{ examples}}).$$

Now, for each $z \in \mathscr{I}_{\zeta/5}$, run $\mathcal{A}(S_z)$ repeatedly $4\ln(4)e^{8\alpha\varepsilon m}$ times. Store all the outputs in set $\mathcal{H}$, which has size $|\mathcal{H}| = 5/\zeta \cdot 4\ln(4)e^{8\alpha\varepsilon m}$. It is clear that for $z = \lfloor c(0) \rfloor_{\zeta/5}$, $S_z = S_{\text{alt}}$, and Eq. (35) therefore gives us guarantees on the output of $\mathcal{A}(S_z)$. We may conclude from Eq. (35) that for set $\mathcal{H}$ generated in the above fashion,

$$\Pr[\mathcal{H} \cap G_{D,\zeta}^{1/4} = \varnothing] \leq \left(1 - \frac{1}{4}e^{-8\alpha\varepsilon m}\right)^{4\ln(4)e^{8\alpha\varepsilon m}} \leq \frac{1}{4}. \tag{36}$$

Rearranging gives $m = \frac{1}{8\alpha\varepsilon}\left(\mathsf{PRdim}_{\zeta,1/4,1/4}(\mathcal{C}) - \ln(5/\zeta \cdot 4\ln 4)\right)$.

We may therefore define $\mathscr{H} := \left\{\mathcal{G} \subseteq \mathcal{F} : |\mathcal{G}| \leq 5/\zeta \cdot 4\ln(4)e^{8\alpha\varepsilon m}\right\}$ (note that $\mathcal{H} \in \mathscr{H}$) and further define $\mathcal{P}$ to be the distribution that puts all probability mass on $\mathcal{H}$. Comparing Eq. (36) with the definition of PRdim, Definition A.9, observe that $(\mathscr{H}, \mathcal{P})$ make up a $(\zeta, 1/4, 1/4)$-probabilistic representation for the class $\mathcal{C}$. Hence $\mathsf{PRDim}_{\zeta,1/4,1/4} \leq \ln(5/\zeta \cdot 4\ln(4)) + 8\alpha\varepsilon m$. $\qquad\square$

The following lemma is an immediate corollary of [21] who proved it for Boolean functions and the exact same proof carries over for our definition of PRdim and randomized one-way communication model in the real-valued setting.

**Lemma D.2** (PRdim $\asymp$ Randomized Communication Complexity for real-valued functions)**.** *Let $\mathcal{C}$ be a concept class of real-valued functions. The following relations hold:*

1. $\mathsf{PRdim}_{\zeta,\varepsilon,\delta}(\mathcal{C}) \leq R_{\zeta,\varepsilon\delta}^{\to,pub}(\mathcal{C})$

2. $R_{\zeta,\varepsilon+\delta-\varepsilon\delta}^{\to,pub}(\mathcal{C}) \leq \mathsf{PRdim}_{\zeta,\varepsilon,\delta}(\mathcal{C})$

 **D.2 One-way communication is characterized by** $\mathsf{sfat}(\cdot)$

We next prove that for every real-valued concept class $\mathcal{C} \subseteq \{f : \mathcal{X} \to [0,1]\}$, the sequential fat-shattering dimension lower bounds the randomized communication complexity of $\mathcal{C}$. Namely, we prove the following lemma:

**Lemma D.3.** *Let* $\mathcal{C} \subseteq \{f : \mathcal{X} \to [0,1]\}$ *be a concept class. Then* $R^{\to}_{\zeta,\varepsilon}(\mathcal{C}) \geq (1 - H(\varepsilon)) \cdot \mathsf{sfat}_\zeta(\mathcal{C})$.

With this lemma, we complete our chain of implications, and obtain the conclusion of this section, that the sample complexity of pure DP PAC learning upper-bounds the $\mathsf{sfat}(\cdot)$ dimension. We remark that the statement above is the real-valued version of the relationship exhibited in [21], wherein the Littlestone dimension (Boolean analog of $\mathsf{sfat}(\cdot)$) lower-bounds the randomized communication complexity of Boolean function classes. The proof of Lemma D.3 proceeds in two steps. First, we define the communication problem $\mathsf{AugIndex}_d$ and show that $R^{\to}_{\zeta,\varepsilon}(\mathcal{C}) \geq R^{\to}_{\varepsilon}(\mathsf{AugIndex}_d)$ for $d$ the $\mathsf{sfat}$ dimension of $\mathcal{C}$. (We refer the reader to Section A.2.2 for the definitions of the quantities $R^{\to}_{\zeta,\varepsilon}(\cdot)$ and $R^{\to}_{\varepsilon}(\cdot)$ which pertain respectively to real- and Boolean-function communication complexity.) Next, we use the known relation $R^{\to}_{\varepsilon}(\mathsf{AugIndex}_d) > (1 - H(\varepsilon))d$ where $H : [0,1] \to [0,1]$ is the binary entropy function $H(x) := -x \log x - (1-x) \log(1-x)$.

To do the first of the two steps, we will relate the one-way classical communication complexities of two communication tasks. The first is the task $\mathsf{AugIndex}_d$ for $d \in \mathbb{Z}_+$ which is defined as follows: Alice gets string $x \in \{0,1\}^d$, while Bob gets $x_{[i-1]}$ for some $i \in [d]$, which is the length-$(i-1)$ prefix of $x$. The task is for Bob to output the bit $x_i$ and we say that $\mathsf{AugIndex}_d(x, i) = x_i$. The second is the task $\mathsf{Eval}_\mathcal{C}$, defined in Section A.2.2, for some real-valued function class $\mathcal{C} \subseteq \{f : \mathcal{X} \to [0,1]\}$. We repeat the definition for convenience: Alice is given a function $f \in \mathcal{C}$ and Bob a $z \in \mathcal{X}$ and Bob's goal is to approximately compute $f(z)$, i.e. Bob has to compute $b \in [0,1]$ satisfying

$$\Pr\left[|b - f(z)| \leq \zeta\right] \geq 1 - \varepsilon, \tag{37}$$

where the probability is taken over the local randomness of Alice and Bob respectively. We denote the one-way randomized communication complexity of $\mathsf{Eval}_\mathcal{C}$ as $R^{\to}_{\zeta,\varepsilon}(\mathcal{C})$ for short.

**Lemma D.4.** *If* $\mathcal{C} \subseteq \{f : \mathcal{X} \to [0,1]\}$ *satisfies* $\mathsf{sfat}_\zeta(\mathcal{C}) = d$, *then* $R^{\to}_{\zeta,\varepsilon}(\mathcal{C}) \geq R^{\to}_{\varepsilon}(\mathsf{AugIndex}_d)$.

*Proof.* The idea of the proof is to show that a a one-way communication protocol for $\mathsf{Eval}_\mathcal{C}$ can also be used to compute $\mathsf{AugIndex}_d$ for $d = \mathsf{sfat}_\zeta(\mathcal{C})$. The protocol for $\mathsf{AugIndex}_d$ is as follows:

1. Alice and Bob agree on the $\zeta$-fat-shattering tree for the concept class $\mathcal{C}$ ahead of time.

2. Upon being given an instance of the $\mathsf{AugIndex}_d$ problem, Alice (who has the $d$-bit string $x$) identifies some function in $\mathcal{C}$ as follows: she follows the $\zeta$-fat-shattering tree down the path of left-right turns defined by string $x$. This takes her to a leaf $\ell$ which is associated with some unique function $c_{\mathrm{Alice}} \in \mathcal{C}$. Bob (who has the $(i-1)$-bit string $x_{[i-1]}$) identifies some $z_{\mathrm{Bob}} \in \mathcal{X}$, $a_{\mathrm{Bob}} \in [0,1]$ as follows: he follows the $\zeta$-fat-shattering tree down the path of left-right turns defined by $x_{[i-1]}$. This takes him to some node $w$ at level $i-1$ and Bob sets $z_{\mathrm{Bob}}, a_{\mathrm{Bob}}$ to be the domain point and threshold associated with that node.

3. Alice and Bob use their protocol $\pi$ for $\mathrm{Eval}_\mathcal{C}$ on the inputs $c_{\mathrm{Alice}}, z_{\mathrm{Bob}}$, and following this protocol allows Bob to compute a $b$ that satisfies

$$\Pr\left[|b - c_{\mathrm{Alice}}(z_{\mathrm{Bob}})| \leq \zeta\right] \geq 1 - \varepsilon. \tag{38}$$

4. If $b > a_{\mathrm{Bob}}$, Bob outputs 1; else output 0.

We now prove the correctness of this protocol. Eq. (38) states that with probability $1 - \varepsilon$, $b$ is a $\zeta$-approximation of $c_{\mathrm{Alice}}(z_{\mathrm{Bob}})$. Condition on this. In parallel, observe that the Alice's leaf $\ell$ associated with the function $c_{\mathrm{Alice}}$ is a descendent of Bob's node $w$ associated with the values $(z_{\mathrm{Bob}}, a_{\mathrm{Bob}})$, therefore one of the following two statements must be true by definition of $\zeta$-fat-shattering tree and by the procedure outlined in Step 2:

- $\ell$ is in the right subtree of $w$ i.e. $c_{\mathrm{Alice}}(z_{\mathrm{Bob}}) > a_{\mathrm{Bob}} + \zeta$, and $x_i = 1$. By Eq. (38), this implies $b > a_{\mathrm{Bob}}$. By Step 4, Bob outputs 1, which is also the value of $x_i = \mathsf{AugIndex}_d(x, i)$.

833    • $\ell$ is in the left subtree of $w$ i.e. $c_{\mathrm{Alice}}(z_{\mathrm{Bob}}) < a_{\mathrm{Bob}} - \zeta$, and $x_i = 0$. By Eq. (38), this implies
834      $b < a_{\mathrm{Bob}}$. By Step 4, Bob outputs 0, which is also the value of $x_i = \mathsf{AugIndex}_d(x, i)$.

835    This means that the output of Bob in Step 4, $\tilde{b}$, satisfies

$$\Pr[\tilde{b} = \mathsf{AugIndex}_d(x, i)] \geq 1 - \varepsilon, \tag{39}$$

836    where again the probability is taken over the randomness of Alice and Bob. Hence, the protocol
837    above is a valid protocol for computing $\mathsf{AugIndex}_d$. □

838    Finally we can prove the lemma stated at the beginning of the section.

839    *Proof of Lemma D.3.* Follows from Lemma D.4 combined with the inequality $R_\varepsilon^\rightarrow(\mathsf{AugIndex}_d) \geq$
840    $(1 - H(\varepsilon))d$ which was proven in [21]. □

841    In fact, below we strengthen the above into a bound on the one-way *quantum* communication
842    complexity of computing real-valued concept classes.

843    **Corollary D.5.** *Let* $\mathcal{C} \subseteq \{f : \mathcal{X} \to [0, 1]\}$ *be a concept class. Then* $Q_{\zeta, \varepsilon}^\rightarrow(\mathcal{C}) \geq (1 - H(\varepsilon)) \cdot \mathsf{sfat}_\zeta(\mathcal{C})$.

844    *Proof of Corollary D.5.* In the proof of Lemma D.4, simply replace the classical one-way random-
845    ized protocol to compute $\mathrm{Eval}_C$ with the quantum one-way randomized protocol. This gives that
846    $Q_{\zeta, \varepsilon}^\rightarrow(C) \geq Q_\varepsilon^\rightarrow(\mathsf{AugIndex}_d)$. Next, [22] provides a bound for the complexity of quantum *serial*
847    *encoding* that amounts to the statement $Q_\varepsilon^\rightarrow(\mathsf{AugIndex}_d) \geq (1 - H(1 - \varepsilon))d$. Combining the two
848    yields the claim. □

849    We remark that a similar corollary for *Boolean* valued concept classes was proven earlier by [23]
850    (where the RHS of Corollary D.5 is replaced by Littlestone dimension). Our proof technique is easily
851    generalized to the Boolean setting and significantly simplifies his proof [23].

852    # E   Applications of our results

853    We now present a few applications of the results we established in the previous sections. For the
854    rest of this section, let $\mathcal{U}$ be a class of quantum states on $n$ qubits, and let $\mathcal{U}_n$ refer to the set of
855    *all* quantum states on $n$ qubits. So far, we have shown that the complexity of learning the quantum
856    states from the class $\mathcal{U}$, in two models of learning (pure DP PAC and online learning in the mistake
857    bound model), depends on the sequential fat shattering dimension of the real-valued function class $\mathcal{C}_{\mathcal{U}}$
858    associated with $\mathcal{U}$: here $\mathcal{C}_{\mathcal{U}} := \{f_\rho : \mathcal{X} \to [0, 1]\}_{\rho \in \mathcal{U}}$, where $\mathcal{X}$ is the set of all possible two-outcome
859    measurements, and $f_\rho$ is given by $f_\rho(E) = \mathsf{Tr}(E\rho)$ for every $E \in \mathcal{X}$.

860    In the online learning work of [2] they consider the setting where $\mathcal{U}$ is the set of all $n$-qubit states
861    $\mathcal{U}_n$. Let us denote the corresponding function class as $\mathcal{C}_n$. In this case, [2] showed that $\mathsf{sfat}_\varepsilon(\mathcal{C}_n) \leq$
862    $O(n/\varepsilon^2)$, thus effectively upper-bounding the $\mathsf{sfat}(\cdot)$ dimension of the class of all $n$-qubit quantum
863    states by $n$. This section asks what happens when we allow $\mathcal{U} \subseteq \mathcal{U}_n$ – for instance, when $\mathcal{U}$ is a
864    special class of states that may be of particular interest or more experimentally feasible to prepare.
865    Are there any meaningful such classes for which we can improve this bound? We first answer this
866    affirmatively for a few classes of quantum states and finally improve the sample complexity of gentle
867    shadow tomography for these classes of states.

868    ## E.1   Holevo information and sequential fat shattering dimension

869    In this subsection we provide an upper bound on $\mathsf{sfat}(\mathcal{C}_{\mathcal{U}})$ in terms of the Holevo information of
870    an ensemble defined on the class of states $\mathcal{U}$. Using this new upper bound leads to improved upper
871    bounds on $\mathsf{sfat}(\cdot)$ for many classes of quantum states $\mathcal{U}$, and hence improved upper bounds on the
872    sample complexity of learning $\mathcal{U}$. Previously for $\mathcal{U} = \mathcal{U}_n$, [24, 2] observed that one could use
873    arguments from quantum random access code by [22] to obtain a *combinatorial* upper bound on
874    learning. In this section we show that a better upper bound can be achieved by maximizing the Holevo

information, $\chi(\{p_i, \rho_i\}_{\rho_i \in \mathcal{U}})$ (over all possible distributions $\vec{p}$ on $\mathcal{U}$), where Holevo information is defined as

$$\chi\left(\{p_i, \rho_i\}_{\rho_i \in \mathcal{U}}\right) = S(\bar{\rho}) - \sum_{i:\rho_i \in \mathcal{U}} p_i S(\rho_i), \quad \bar{\rho} = \sum_{i:\rho_i \in \mathcal{U}} p_i \rho_i, \tag{40}$$

where $\vec{p}$ is a distribution and $S$ is the von Neumann entropy $S(\rho) := -\mathsf{Tr}[\rho \log \rho]$.

### E.1.1 Quantum Random Access Codes

We first define random access codes and serial random access codes over the set $\mathcal{U}$, modifying the definition in [22] so that $\mathcal{U}$ – the set of states from which the code states may be chosen – is part of the definition of these codes.

**Definition E.1** (Random access codes and serial random access codes). *Let $\mathcal{U}$ be a class of quantum states over $n$ qubits. A $(k, n, p, \mathcal{U})$-random access code (RAC) consists of a set of $2^k$ code states $\{\rho_s\}_{s \in \{0,1\}^k} \subseteq \mathcal{U}$ such that, for every $i \in [k]$ and $s \in \{0,1\}^k$, there exists a 2-outcome measurement $\mathcal{O}_i$ such that*

$$\mathsf{Pr}[\mathcal{O}_i(\rho_s) = s_i] \geq p. \tag{41}$$

*A $(k, n, p, \mathcal{U})$-serial random access code (SRAC) consists of $2^k$ code states $\{\rho_s\}_{s \in \{0,1\}^k} \subseteq \mathcal{U}$ such that, for every $i \in [k]$, and for all $s \in \{0,1\}^k$, there exists a measurement with outcome 0 or 1, possibly depending on the last $k - i$ bits $x_{i+1}, \ldots, x_k$, such that Eq. (41) holds.*

In words, a RAC over $\mathcal{U}$ is a way of encoding $k$ classical bits into $n$-qubit states from $\mathcal{U}$, such that for every $i \in [k]$ and $x \in \{0,1\}^k$, the probability of 'recovering' the bit $x_i$ by performing the 2-outcome measurement $\mathcal{O}_i$ on $\rho_x$ is at least $p$. A serial RAC (denoted SRAC) is defined similarly except that one is allowed to use information from decoding the *subsequent bits* to decode $x_i$. [22] showed the following relation between the number of encodable classical bits and the number of qubits in the code states

$$\text{Every } (k, n, p, \mathcal{U}_n)\text{-RAC or } (k, n, p, \mathcal{U}_n)\text{-SRAC satisfies } n \geq (1 - H(p))k. \tag{42}$$

Here, $H(\cdot)$ is the binary entropy function, and note that the statement applies to code states drawn from the *entire* class of $n$-qubit states.

[2] in a recent work showed the surprising connection that a $p$-sequential fat-shattering tree for $\mathcal{U}$ of depth $k$ can be used to construct a $(k, n, p, \mathcal{U})$-SRAC.[11] As a corollary of this observation, we have

$$\mathsf{sfat}_p(\mathcal{C}_{\mathcal{U}}) \leq \max\{k : \text{ there exists } (k, n, p, \mathcal{U}) - \mathsf{SRAC}\}. \tag{43}$$

Combining Eq. (42), (43) yields $\mathsf{sfat}_p(\mathcal{C}_{\mathcal{U}}) \leq n/(1 - H(p))$. In this section, we consider the scenario where $\mathcal{U} \subseteq \mathcal{U}_n$ and show that this bound can be improved to the following.

**Theorem E.2** (Bounding $\mathsf{sfat}(\cdot)$ by the Holevo information). *Let $p \in [0, 1]$ and $\mathcal{U}$ be some class of quantum states over $n$ qubits. Then*

$$\mathsf{sfat}_p(\mathcal{C}_{\mathcal{U}}) \leq \frac{1}{1 - H(p)} \max\left\{\chi\left(\{(q_i, \sigma_i)\}_{\sigma_i \in \mathcal{U}}\right) : \sum_i q_i = 1\right\}.$$

To do so, we tighten the argument of [22] which was originally derived for $\mathcal{U} = \mathcal{U}_n$. To prove our result, we make use of the following lemma.

**Lemma E.3** ([22]). *Let $\sigma_0, \sigma_1$ be density matrices and $\sigma = \frac{1}{2}(\sigma_0 + \sigma_1)$. If $\mathcal{O}$ is a measurement with $\{0, 1\}$-outcome such that making the measurement on $\sigma_b$ yields the bit $b$ with probability $p$, then*

$$S(\sigma) \geq \frac{1}{2}[S(\sigma_0) + S(\sigma_1)] + (1 - H(p)).$$

We now state and prove our main lemma.

---

[11] We remark that such a connection between RAC and learnability was established in an earlier work by [24] to understand PAC learnability of quantum states.

**Lemma E.4.** *Let $\mathcal{U}$ be some class of quantum states over $n$ qubits. Every $(k, n, p, \mathcal{U})$-RAC or $(k, n, p, \mathcal{U})$-SRAC satisfies*

$$(1 - H(p))k \leq \max\left\{\chi\big(\{(q_i, \sigma_i)\}_{\sigma_i \in \mathcal{U}}\big) : \sum_i q_i = 1\right\}, \tag{44}$$

*where $H(\cdot)$ is the binary entropy function and $\chi$ is the Holevo information $\chi\big(\{(q_i, \sigma_i)\}_{\sigma_i \in \mathcal{U}}\big) = S(\sum_i p_i \sigma_i) - \sum_i p_i S(\sigma_i)$ and $S(\cdot)$ is the von Neumann entropy function.*

*Proof.* Using Definition E.1, a $(k, n, p, \mathcal{U})$-RAC consists of a set of code states $\{\rho_x\}_{x \in \{0,1\}^k} \subseteq \mathcal{U}$ and measurements $\{\mathcal{O}_i\}_{i \in [k]}$ satisfying $\Pr[\mathcal{O}_i(\rho_x) = x_i] \geq p$. Proceeding as in [22], we first define the following states which are derived from the code states: For every $0 \leq \ell \leq k$ and $y \in \{0,1\}^\ell$, let

$$\sigma_y = \frac{1}{2^{k-\ell}} \sum_{z \in \{0,1\}^{k-\ell}} \rho_{zy}.$$

In words, for a $\ell$-bit string $y$, let $\sigma_y$ be a uniform superposition over all $2^{n-\ell}$ code states with the suffix $y$. Let $\psi = \frac{1}{2^n} \sum_{z \in \{0,1\}^n} \rho_z$ be the uniform superposition over *all* code states. Then we have

$$S(\psi) \geq \frac{1}{2^k} \sum_{z \in \{0,1\}^k} S(\rho_z) + k(1 - H(p)). \tag{45}$$

To see this, first one can use Lemma E.3 to show $S(\psi) \geq \frac{1}{2}(S(\sigma_0) + S(\sigma_1)) + 1 - H(p)$ and recursively applying this lemma to each of the $S(\cdot)$ quantities, we get the equation above (observe that each application of the lemma is justified because for every $y \in \{0,1\}^\ell$, we may write $\sigma_y = \frac{1}{2}(S(\sigma_{0y}) + S(\sigma_{1y}))$; and by assumption of a $(k, n, p, \mathcal{U})$-RAC, $\mathcal{O}_{\ell+1}$ can distinguish $\sigma_{0y}, \sigma_{1y}$ with success probability $p$ and thus is a measurement that meets the conditions of Lemma E.3.) Using Eq. (45) it now follows that

$$k(1 - H(p)) \leq S(\psi) - \frac{1}{2^k} \sum_{z \in \{0,1\}^k} S(\rho_z) = \chi\Big(\Big\{\frac{1}{2^k}, \rho_x\Big\}_{x \in \{0,1\}^k}\Big) \leq \max_{T \subseteq \mathcal{U}} \chi\Big(\Big\{\frac{1}{|T|}, \sigma_i\Big\}_{\sigma_i \in T}\Big). \tag{46}$$

where the last inequality follows because the uniform ensemble of code states $\big\{\frac{1}{2^k}, \rho_x\big\}_{x \in \{0,1\}^k}$ is precisely of the form $\{p_i, \sigma_i\}_{\sigma_i \in \mathcal{U}}$ with zero weight on non-code states in $\mathcal{U}$. In Eq. (44), to get a simpler-looking bound, we further relax this inequality by taking the optimization over arbitrary probability distributions on the code states, not just the ones that are uniform on a subset. Eq. (44) also holds for SRAC by noting that the argument above doesn't change by allowing $\mathcal{O}_i$ to depend on bits $x_{i+1}, \ldots, x_k$. $\square$

The proof of Theorem E.2 follows immediately from combining Lemma E.4 and Observation (43).

An interesting consequence of our result is the following. As far as we are aware, there is no way of computing $\mathsf{sfat}(\cdot)$ directly, but there exist algorithms to compute our bound in Theorem E.2. For a set $\mathcal{U}$ of states, performing the maximization $\max\{\chi\big(\{(q_i, \sigma_i)\}\big) : \sum_i q_i = 1\}$ is a convex optimization problem which can be solved using the Blahut-Arimoto algorithm[25]. However, for certain special classes of states, one can present simple bounds on the maximal Holevo information which we present next.

### E.1.2 Classes of states with bounded $\mathsf{sfat}(\cdot)$ dimension

A natural question is, how does the new upper bound on $\mathsf{sfat}(\mathcal{U})$ in Theorem E.2 compare to the previous upper bound $\mathsf{sfat}(\mathcal{U}_n) < n/\varepsilon^2$ given in [2]. Observe that that the $\varepsilon$ dependence comes about from a Taylor expansion of $1 - H((1 - \varepsilon)/2)$ and our new bounds do not change this dependence, hence for the remainder of this section we set $\varepsilon = 1$ for simplicity. We now mention a few classes of states for which our new bound improves the $n$ dependence of the previous bound.

- Suppose our quantum states are "$k$-juntas", i.e., each $n$-qubit quantum state lives in the same *unknown* $k$-dimensional subspace of the $2^n$-dimensional Hilbert space. Then clearly, the right-hand-side of Eq. (44) is upper-bounded by $\log k < n$. In particular for $n$-juntas the $\mathsf{sfat}(\cdot)$ dimension is $O(\log n)$, hence the sample complexity of learning scales as $O(\log n)$ which is exponentially better than the prior upper bounds of $n$.

- $\mathcal{U}$ consists of a small set of states with small pairwise trace distance; in [26] and [27] they showed that

$$\chi(\{p_i, \rho_i\}) \leq v_m \log |\mathcal{U}| \tag{47}$$

where $v_\mathrm{m} = \frac{1}{2} \sup_{i,j} \|\rho_i - \rho_j\|_1$ is the maximal trace norm distance between the states in the class $\mathcal{U}$. This bound could be significantly better than the trivial $\log |\mathcal{U}|$ if $v_m$ is sufficiently small.

- Let $\mathcal{U} = \mathcal{N}(\mathcal{U}_n)$ be the set of all $n$-qubit states obtained after passing the states in $\mathcal{U}_n$ through the channel $\mathcal{N}$. That is, we would like to learn some arbitrary $n$-qubit state that has been passed through an *unknown* quantum channel $\mathcal{N}$. This is the case in many experimentally-relevant settings and is in fact one way to understand the effect of experimental noise (which can be modelled by a quantum channel during state preparation). The Holevo information of the quantum channel $\mathcal{N}$ is the following quantity

$$\chi(\mathcal{N}) := \max_{\vec{p}, \rho_i} S\Big(\sum_i p_i \mathcal{N}(\rho_i)\Big) - \sum_i p_i S(\mathcal{N}(\rho_i)), \tag{48}$$

where the maximization is over (arbitrary-sized) ensembles $\{(p_i, \rho_i)\}$. Observe that using Eq. (44) one can upper bound $\mathsf{sfat}(\cdot)$ dimension of the set $\mathcal{U} = \mathcal{N}(\mathcal{U}_n)$ in terms of $\chi(\mathcal{N})$. A centerpiece of quantum Shannon theory is the Holevo-Schumacher-Westmoreland (HSW) theorem [28], which states that (see for example [29] for a pedagogical proof) $\chi(\mathcal{N}) \leq C(\mathcal{N})$ where $C(\mathcal{N})$ is the classical capacity of the channel. Putting these two bounds together gives

$$\mathsf{sfat}(\mathcal{N}(\mathcal{U}_n)) \leq C(\mathcal{N}). \tag{49}$$

Now, using the connection above one can upper bound $\mathsf{sfat}(\cdot)$ of noisy quantum states using results developed in quantum Shannon theory to bound the classical channel capacity. For a depolarizing channel acting on $d$-dimensional states with parameter $\lambda$ for instance (a common noise model), one can upper bound $C(\mathcal{N})$ in Eq. (49) by a result of [30] as follows

$$\log d - S_{\min}(\Delta_\lambda) \tag{50}$$

where $S_{\min}(\Delta_\lambda) = -\left(\lambda + \frac{1-\lambda}{d}\right)\log\left(\lambda + \frac{1-\lambda}{d}\right) - (d-1)\left(\frac{1-\lambda}{d}\right)\log\left(\frac{1-\lambda}{d}\right)$ and the sub-tractive quantity in the quantity above makes this bound strictly better than [2]. Similar upper bounds on channel capacity are also known for Pauli channels [31] and generalized Pauli channels [32].

- Interestingly, we may now also bound $\mathsf{sfat}(\cdot)$ of the class of quantum Gaussian states. Since these states are infinite-dimensional, the previous bound of [2] is not useful. However, our channel capacity upper-bound on $\mathsf{sfat}(\cdot)$ yields a finite bound: It is known from [33] that the channel capacity of a pure-loss bosonic channel with transmissivity $\eta \in [0, 1]$,[12] when the input Gaussian states have photon number at most $N_p$ (and hence bounded energy, which is physically realistic), is $g(\eta N_p)$ where $g(x) \equiv (x + 1)\log_2(x + 1) - x \log_2 x$. In particular, the case $\eta = 1$ corresponds to zero loss, hence $g(N_p)$ bounds $\mathsf{sfat}(\cdot)$ for the *entire* class of Gaussian states with $N_p$ photons.

Alternatively, one might be interested in states prepared through phase-insensitive bosonic channels. These model other kinds of noise, such as thermalizing or amplifying processes. A recent breakthrough [34] allows one to bound the capacities of these channels, and hence the $\mathsf{sfat}(\cdot)$ dimensions of these noisy Gaussian states.

## E.2 Faster online shadow tomography

We now discuss how our results can also improve *shadow tomography*, a learning framework recently introduced by Aaronson [35]. This is a variant of quantum state tomography in which the goal is not

---

[12]This channel is a simple model for communication over free space or through a fiber optic link, where $\eta$ models how much noise is 'mixed' into the states.

to learn $\rho$ completely, but to learn its 'shadows', i.e., the expectation values of $\rho$ on a fixed (known) set of measurements.

To be precise, let $\mathcal{U}$ be a subset of $n$-qubit states. Given $T$ copies of an unknown state $\rho \in \mathcal{U}$, and a set of known two-outcome measurements $E_1, \ldots, E_m$. The goal is to learn (with probability at least $2/3$) $\mathsf{Tr}(E_i\rho)$ upto additive error $\varepsilon$ for every $i \in [m]$. A trivial learning algorithm uses $T = O((2^n + m) \cdot \varepsilon^{-2})$ many copies of $\rho$ to solve the task, and surprisingly Aaronson showed how to solve this task using $T = \mathrm{poly}(n, \log m, \varepsilon^{-1})$ copies of $\rho$, exponentially better than the trivial algorithm. An intriguing open question left open by Aaronson [35] and others is, is the $n$ dependence necessary? There have been follow up results by [36] that improved Aaronson's procedure when the goal is obtain 'classical shadows' and more recently [37] gave a procedure which has the best known dependence on all parameters for standard shadow tomography.

Subsequently [17] considered *gentle* shadow tomography a (stronger) variant of shadow tomography (we do not define gentleness here and refer the interested reader to [17]). Here, we show that suppose we were performing gentle shadow tomography with the prior knowledge that the unknown state $\rho$ came from a class of states $\mathcal{U}$, then the $n$-dependence in the sample complexity can be replaced with $\mathsf{sfat}(\mathcal{C}_\mathcal{U})$. As we discussed in the previous section, clearly $\mathsf{sfat}(\mathcal{C}_\mathcal{U}) \leq \tilde{O}(n/\varepsilon^2)$, but for many class of states $\mathsf{sfat}(\mathcal{C}_\mathcal{U})$ could be much lesser than $n$, giving us a significant improvement over Aaronson's result. We first state our main statement.

**Theorem E.5** (Faster gentle shadow tomography). *The complexity of gentle shadow tomography on a class of states $\mathcal{U}$ is*

$$O\left(\frac{\mathsf{sfat}_\varepsilon(\mathcal{C}_\mathcal{U})^2 \log^2 m \log(1/\delta)}{\varepsilon^2 \min\{\alpha^2, \varepsilon^2\}}\right). \tag{51}$$

*where $\alpha, \delta$ are gentleness parameters and the goal is to learn $\Pr[E_i(\rho) \text{ accepts}]$ to within an additive error of $\varepsilon$ for every $i \in [m]$.[13] Moreover, there exists an explicit algorithm that achieves this.*

Indeed the parameter $\mathsf{sfat}(\mathcal{C}_\mathcal{U})$ in this bound means that for the classes of states mentioned in Section E.1.2, the sample complexity of shadow tomography is better than the complexity in [35] (in terms of $n$). We now prove Theorem E.5. The connection comes from the implication in [17] that under certain conditions, an online learner for quantum states can be used as a black box for what they term 'Quantum Private Multiplicative Weights', an algorithm that performs shadow tomography in both an online and a gentle manner. We now state the precise setting in which this black box online learner must operate. As usual, we are concerned with the function class $\mathcal{C}_\mathcal{U} := \{f_\rho\}_{\rho \in \mathcal{U}}$ where the domain $\mathcal{X}$ is the set of all possible two-outcome measurements $E$ on the states in $\mathcal{U}$ and the functions in the class are defined as $f_\rho(E) = \mathsf{Tr}(E\rho)$ for every $E$. The unknown state $\rho$ defines some target function $c \in \mathcal{C}_\mathcal{U}$.

1. Adversary provides input point in the domain: $x_t \in \mathcal{X}$.

2. Learner outputs a prediction $\hat{y}_t \in [0, 1]$.

3. If the learner makes a mistake, i.e., if $|\hat{y}_t - c(x_t)| > \varepsilon$, then adversary provides strong feedback $\hat{c}(x_t) \in [0, 1]$ where $\hat{c}(x_t)$ is an $\varepsilon/10$-approximation of $c(x_t)$, i.e., $|\hat{c}(x_t) - c(x_t)| < \varepsilon/10$, and the learner is allowed to update its hypothesis. Else, the adversary does not provide any feedback, and the learner must use the same hypothesis on the next round.

4. Learner suffers loss $|\hat{y}_t - c(x_t)|$.

Observe that this is a close variant of our setting in Section A.1, the only difference being that the adversary here only gives feedback on rounds where the learner makes a mistake (i.e., when the learner's prediction is grossly wrong). This means that the learner updates her hypothesis if and only if it makes a mistake. Given an online learner $\mathcal{A}$ in the above setting that makes at most $\ell$ updates, [17] shows that there exists a randomized algorithm $\mathcal{B}$ for shadow tomography using

$$n = O\left(\frac{\ell^2 \log^2 m \log(1/\delta)}{\varepsilon^2 \min\{\alpha^2, \varepsilon^2\}}\right). \tag{52}$$

many examples of the unknown state $\rho$ where such that algorithm $\mathcal{B}$'s error is bounded by $\varepsilon$ with probability at least $1 - \beta$. Moreover, this algorithm is $(\alpha, \delta)$-gentle. We are now equipped with all

---

[13]Implicitly in the complexity above we have assumed that the algorithm succeeds with probability at least $2/3$.

we need to prove Theorem E.5. The proof boils down to the observation that for any concept class $\mathcal{C}$, we can always construct an online learner that is guaranteed to make at most $\mathsf{sfat}(\mathcal{C})$ mistakes, and therefore $\ell = \mathsf{sfat}(\mathcal{C})$ in Eq. (52). The online learner we construct is a variant of the proper version of our RSOA Algorithm 1.

*Proof of Theorem E.5.* The proof follows from the Quantum Private Multiplicative Weights algorithm in [17] and its accompanying Theorem 39, simply by exhibiting an online learner $\mathcal{A}$ for $\mathcal{U}$ in the setting described above, that makes at most $\ell = \mathsf{sfat}_\varepsilon(\mathcal{C}_S)$ mistakes. In the rest of this proof, we exhibit just such an algorithm, which is a variant of the proper version of RSOA.

---

**Algorithm 4** Alternative Robust Standard Optimal Algorithm

**Input:** Concept class $\mathcal{C} \subseteq \{f : \mathcal{X} \to [0,1]\}$, target (unknown) concept $c \in \mathcal{C}$, and $\varepsilon \in [0,1]$.
**Initialize**: $V_1 \leftarrow \mathcal{C}$

1: **for** $t = 1, \dots, T$ **do**
2:     A learner receives $x_t$ and maintains set $V_t$, a set of "surviving functions".
3:     For every super-bin midpoint $r \in \tilde{\mathscr{I}}_{2\varepsilon/5}$ the learner computes the set of functions $V_t(r, x_t)$.
4:     A learner finds the super-bin which achieves the maximum $\mathsf{sfat}(\cdot)$ dimension

$$R_t(x_t) := \left\{ \underset{r \in \tilde{\mathscr{I}}_{2\varepsilon/5}}{\arg\max} \, \mathsf{sfat}_{2\varepsilon/5}\left(V_t(r, x_t)\right) \in \tilde{\mathscr{I}}_{2\varepsilon/5} \right\}$$

5:     The learner computes the mean of the set $R_t(x_t)$, i.e., let

$$\hat{y}_t := \frac{1}{|R_t(x_t)|} \sum_{r \in R_t(x_t)} r.$$

6:     The learner outputs $\hat{y}_t$, receives feedback $\hat{c}(x_t)$ if it has made a mistake, i.e., if $|\hat{y}_t - c(x_t)| > \varepsilon$.
7:     If the learner received feedback, update $V_{t+1} \leftarrow \{g \in V_t \mid g(x_t) \in B_{\varepsilon/5}(\hat{c}(x_t))\}$; else $V_{t+1} \leftarrow V_t$.
8: **end for**

**Outputs:** The intermediate predictions $\hat{y}_t$ for $t \in [T]$, and a final prediction function/hypothesis which is given by $f(x) := R_{T+1}(x)$.

---

The difference between Algorithm 4 and RSOA is that in RSOA, the learner is allowed to update the set $V_t$ on all rounds $t \in [T]$, while in Algorithm 4, the update happens only on the rounds for which it made a mistake ('mistake rounds'). Because the learner's current hypothesis for the target concept is computed based on the 'set of surviving functions' $V_t$, updating $V_t$ amounts to updating the algorithm's hypothesis. We thus aim to show that Algorithm 4 has no more than $\mathsf{sfat}(\cdot)$ mistake rounds. However, we observe that we may directly import the proof of Theorem B.4 to do so. This is because that proof is independent of what happened on the non-mistake rounds, which are the only rounds that differ between RSOA and Algorithm 4. Rather, it argues that on the rounds on which RSOA made a mistake, $\mathsf{sfat}(V_t)$ decreases by at least 1 due to the update on $V_t$, and having initialized $V_1 = \mathcal{C}$, no more than $\mathsf{sfat}(\mathcal{C})$ updates may happen in total. Exactly the same argument can be used to bound the mistakes of Algorithm 4, though note that for the constants to work out, the $\varepsilon$ of RSOA must be multiplied by 5. $\qquad\square$