# OpenReview forum: "Private learning implies quantum stability"
_NeurIPS.cc/2021/Conference — NeurIPS 2021 Spotlight_

### Official Review · Reviewer_C1db · 2021-07-16

**Rating:** 7
**Confidence:** 4

**Summary:**

This paper presents a number of connections between private learning, online learning and stability in the quantum and real-valued settings, generalizing recent work of Bun, Livni and Moran '20 in the Boolean-valued setting. The main results are:
 - An algorithm (termed RSOA) that accomplishes online learning of real-valued functions (requiring feedback only within an $\epsilon$ precision) with a mistake bound given by the sequential fat-shattering (sfat) dimension of the class
 - A notion of real-valued and quantum stability (generalizing Bun et al's "global stability"), along with a proof that online learnability implies this kind of stability
 - A no-go result for (one approach for) converting stability to approximate PAC learnability
 - An improved shadow tomography result with sample complexity in terms of the sfat dimension
 - Bounds on the sfat dimension of quantum state classes in terms of Holevo information

Many of the key notions and proofs are natural generalizations of the Boolean and (exact) real-valued analogs as studied e.g. in Bun et al '20 and Rakhlin et al '10. The concrete applications to quantum state learning are novel and interesting, and address some questions raised in earlier works such as Aaronson et al '18.

**Limitations And Societal Impact:**

I am not aware of any potential negative societal impact of this work, which is primarily theoretical.

I am a bit confused about the no-go theorem between quantum/approximate stability and approximate DP PAC learnability. The claim is that the approach of BLM20 cannot work (in a domain size-independent way) because it amounts to "query release for function balls" (Problem C.7), which is sufficiently general that clique identification (Problem C.8) (shown to be hard in Claim C.9) can be reduced to it. But this reduction seems to obscure the fact that the type of access to the input seems quite different between the two problems. In clique identification, the algorithm is presumably only told that a size-$n$ clique $D$ _exists_ and must find it using access only to the relation graph, whereas in the instance of "query release for function balls" naturally generated by the BLM20 approach, the set of functions $\\{ f_i \\}_{i \in [n]}$ is very much at hand. If this were the clique (and note that the reduction has $\eta = 1$, so all the functions are in the clique), then one could just output any one of these functions (uniformly at random to ensure privacy, I would think) and one has found the clique trivially. Put differently, "query release for function balls" seems trivial in the $\eta = 1$ regime generated by the reduction; the reduction would only seem to make sense if the query release algorithm could not see the functions as part of the input. I may be misunderstanding something, and would appreciate it if the authors could clarify exactly what is happening. (Incidentally, it would be good to be more careful about the use of the word "given" in Claim C.8 so as to distinguish the assumptions made in the problem versus the input the algorithm is literally given. Similarly with the term "special case": a more explicit reduction would be helpful.)

At a higher level, probably my main (subjective) quibble with the paper is that it feels a little unfocused and is perhaps attempting too many different things. It feels a little hard to take away a set of clean insights, and I do think that despite the collection of nice results this lack of focus is a limitation. I wonder if the paper would have been well served by being split into two roughly-standalone parts, one for the real-valued generalizations and another for the quantum applications. It is likely difficult to make any such changes for this conference, but it could be something to keep in mind when making edits.

**Main Review:**

The motivation for this work is primarily to obtain an improved understanding of online learning and shadow tomography of quantum states, as well as to see what the consequences of Bun et al's breakthrough are for the quantum setting. Along the way, one is naturally forced to first consider learning of real-valued functions, and much of the paper deals with generalizing existing results to the real-valued setting with "imprecise feedback". This requires technical care in some places, but to a large extent seems to be reasonably straightforward. These generalizations are arguably not too surprising, but it is still good to have them established.

The authors define a kind of private analog of PAC learning quantum states that is a natural theoretical extension of classical private learning, but privacy for quantum states does not seem particularly well-motivated in its own right, or at any rate could use some more discussion and motivation (other than that it implies other things such as one-way communication and online learning, which is certainly nice).

The RSOA algorithm and its mistake bound in terms of the sfat dimension seems to follow similar ideas to Rakhlin et al's FAT-SOA and indeed Littlestone's original SOA. The proof that online learning implies stability roughly follows Bun et al's elegant approach, except with some definite extra care to handle the real-valued setting properly. The characterization of one-way communication in terms of sfat dimension seems quite interesting, although I do not have too much knowledge of this area. The applications to quantum state learning, especially shadow tomography and online learning, are definitely interesting and make concrete progress in directions suggested by Aaronson et al. The no-go result for online learning implying (approximate) private learning is a little unfortunate, as this was perhaps the main point of Bun et al 20 (although see my question under Limitations below).

The writing is mostly reasonable but not always fully clear, although I do appreciate that the authors have a tricky job on their hands with the scope of this paper and the space constraints. There is some looseness with technical language in a few places. For instance, the definition of stability says "$A$ is stable with respect to distribution $D$ if, given a sample $S \sim D^T$, there exists a state $\sigma$ such that ..." -- but this makes it sound as though $\sigma$ depends on $S$ when in fact it only depends on $D$. That is, it should be something like "$A$ is stable wrt $D$ if there exists $\sigma$ such that whp over sample $S$ ...". (I had to read the original definition in Bun et al to clarify this for myself.) Also, it is a bit confusing that $\mathcal{M}$ is used sometimes to denote all 2-outcome measurements and sometimes just an orthogonal basis; it would be nice to use different letters just for clarity.

Overall, there are many results in this paper, and many of them answer natural questions. Some of them are unsurprising and not necessarily highly original, but have value regardless. With everything added up, this paper definitely makes significant contributions to the area. I did not verify every proof but did try to look over the main ones; I also do not claim familiarity with all of the areas touched upon.

UPDATE: Following discussion with the authors,  I understand the reduction better now but definitely still think it calls for more careful exposition. Overall I continue to think this is a good paper and merits acceptance.

**Time Spent Reviewing:**

6

---

> ### Author Response · Authors · 2021-08-10
> **Reply to questions raised by reviewer**
>
>
> Thank you for your review, we appreciate your thoughtful comments! Here we address your questions/comments:
> On the question about the reduction: Let us clarify that in the clique identification problem, the algorithm is literally given as input all the points in D, and not just the relation graph. This is evident also in how we prove the lower bound -- by using the clique ID algorithm to construct an adversary for a fingerprinting code. An adversary is given the entire codebook (see Def 3.1 in https://arxiv.org/pdf/1311.3158.pdf), which consists of the codewords, or the points in D. You are right in that the reduction works in the η=1 regime. However, we believe the your chain of reasoning breaks down at the statement “one could just output any one of these functions (uniformly at random to ensure privacy, I think)”. In fact, outputting them uniformly at random only ensures privacy up to a delta parameter of 1/n, which is not differentially private. An often unspoken assumption in DP is that the delta parameter must be much smaller than 1/n, precisely because we would like to rule out algorithms that simply pick a point in their input at random and then do something non-private to them. Indeed, our lower bound argument only applies when delta = o(1/n). Thank you for bringing up this intriguing point!
>
> On the use of ‘given/special case’: We have noted the potential confusion. We will change it as we are editing for the final version (if this paper gets accepted).
>
> Finally, we very much understand your concern about the paper seeming unfocussed. We felt that the quantum setting gave the most compelling motivation for the real-valued learning results, but the quantum implications themselves could not stand alone. Thus the best way we could think of to present both sets of results was to present them in the same paper, but clearly demarcate them, which we hope is done in our current format.

---

> > ### Comment · Reviewer_C1db · 2021-08-18
> > **Seeking further clarification regarding reduction**
> >
> > Sorry but I'm still confused about the reduction. In the clique identification problem (Problem C.8), is it correct to say that D itself is a clique in the relation R, and the goal is to find something connected (i.e. related) to everything in D, so that in particular any element in D itself suffices? And you are given D as part of the input? Is that not just "given a list as input, output something from the list"? This definitely seems quite a peculiar problem. (I did not really have time to dive into the Bun et al reference on fingerprinting codes; note that that paper does not itself mention this clique identification problem.) It would be saying that algorithmically this problem is totally trivial, and it is perhaps only somewhat nontrivial when DP enters the picture. And even then I don't really see how the delta = 1/n issue is the critical one. If D has size n and is the size of the largest clique then any randomized algorithm for this problem must put mass at least 1/n on one of the points of D, so that delta = 1/n is the best achievable parameter -- no? IOW, it seems that the trivial algorithm of "output something uniformly at random from D" solves this problem with best-possible DP parameters -- it is a (0, 1/n)-DP algorithm that always succeeds. Is that not right?
> >
> > And finally, even if delta = 1/n is an issue, isn't Claim C.9 actually stated for delta = O(1), which is weaker? That is, the trivial (0, 1/n)-DP algorithm is certainly (1, 1/1500)-DP. It's a bit confusing since the proof words it differently: it uses an "$(\epsilon=1, \delta=1/(4n))$-DP clique identification algorithm with success probability $1499/1500$" to construct the fingerprinting adversary.

---

> > > ### Author Response · Authors · 2021-08-20
> > > **The trivial algorithm doesn't attain the smallest possible value of delta**
> > >
> > > The reviewer is correct to notice that the DP requirement is what makes clique identification nontrivial. Indeed, we may paraphrase the problem as “Given a set $S$ and a subset $s \in S$, output an element of $s$”. However, the subtlety here, is that an algorithm for clique ID knows $S$, even the points outside of $s$, and can put non-zero probability mass on such points. DP algorithms often do precisely that.
> > >
> > > Let us illustrate by discussing the reviewer’s proposed “trivial algorithm” of putting probability mass $1/n$ on all $n$ points in $s$. The reviewer is right that this algorithm has delta parameter $1/n$, however it isn’t true that no DP algorithm can have delta<1/n (here think of delta=1/(4n) for example). To see why, let’s go back to the meaning of $(\epsilon,\delta)$-DP, and set $\epsilon=0$ for simplicity.
> > > “If D has size $n$ and is the size of the largest clique then any randomized algorithm for this problem must put mass at least $1/n$ on one of the points of D.” Indeed, let $p^{\ast}$ be any point which has mass at least 1/n in the output of algorithm A given input $s$. Now consider changing $s$ to $s’$ by replacing any point $p$ in $s$ with a different one $p’ \in S$. $A$ being $(0, \delta)$-DP simply implies that, in the output of $A(s’)$, the probability mass on $p^{\ast}$ cannot differ by more than $\delta$ from what it was in the output of $A(s)$.
> > >
> > > The “trivial algorithm” must have $\delta=1/n$, because $\delta< 1/n$ is violated when we choose $p^{\ast} \in \{p,p’\}$: $A(s)$ puts probability 1/n on $p$ and probability $0$ on $p’$, while $A(s’)$ puts probability 1/n on $p’$ and probability $0$ on $p$. However, nothing excludes the existence of a $(0,\delta< 1/n)$-DP algorithm $A’$, that when run on input $s$, has some probability of outputting $p’$, even though $p’$ is not in $s$. In fact, this is typically achieved by adding some sort of noise to the probability of outputting every point in the universe $S$ which means that even some points not in the input have some finite probability of being output. However, this cannot be done when the data domain is infinite. This gives some intuition as to why our $n$ has a $d$ dependence ($d$ is the size of each data point).
> > >
> > > Finally, the reviewer is right in that in Claim 4.9, we should be more explicit about the range of $\delta$ we are allowing. We neglected to mention (though we should have) that $\delta$ additionally must be $<1/n$. This is used in proving the contradiction as the reviewer pointed out -- see the first equation of the proof. So the new claim should be:
> > > ``For $\delta< 1/500$, any $(1,\delta <1/n)$-DP algorithm solving Problem C.8 with probability at least $1499/1500$ requires $n \geq \tilde{\Omega}(\sqrt{d})$.”
> > > With this, everything works out. We thank the eagle-eyed reviewer for pointing this out and we will make this change in the final version!

---

> > > > ### Comment · Reviewer_C1db · 2021-08-21
> > > > **Review update**
> > > >
> > > > Hm interesting. Let me make sure I understand the relationship between clique identification and your paraphrasing. In clique identification (C.8), you are given $D$ with the guarantee that it is fully connected (ie is a clique), and must find something connected to everything in $D$. That is, if we denote the largest clique containing $D$ by $D'$ (which may or may not be $D$ itself!), then the task is really to output anything in $D'$. In your paraphrasing, $s$ would then be referring to $D$ and $S$ to $D'$. What the algorithm is given is $s$; what it can output is anything in $S$. (Note that this is a little different from being given $S$ itself, which doesn't seem to make as much sense as a paraphrasing.) As I see it, this is precisely what makes the problem interesting and subtly different from merely "given a list, output anything in it" -- rather, it is "given a list, either output anything in it or anything else connected to everything in it". IIUC, this is precisely where the actual clique structure of the problem comes in. Because otherwise one would have to ask what the point of introducing this problem even was, if it all it was was a roundabout way of asking "given a list, output anything in it". Is this right?
> > > >
> > > > I do think this is all a bit subtle and demands careful exposition, so I would urge the authors to do that. (In fact, I confess that I am still not ultra-confident here, but I suppose reasonably confident.)
> > > >
> > > > I thank the authors for their responses but I will keep my review score at 7 all things considered (the clarification of the reduction was not my only issue).

---

### Official Review · Reviewer_uKof · 2021-07-16

**Rating:** 7
**Confidence:** 4

**Summary:**

The paper develop a series of relationship between quantum learning model.
(1) Pure DP PAC learnability implies finite sequential fat shattering dimension.
(2) Finite sequential fat shattering dimension implies online learnability
(3) Online learnability implies certain notion of quantum stability.
(4) quantum stability may not implies approximate PAC learnability.

The work (especially point 3,4 above) is inspired by the recent breakthrough of Bun, Livni and Moran (Journal of the ACM'21) on proving online learnability implies private PAC learnability, extends the result to real-valued setting.



**Limitations And Societal Impact:**

looks good

**Main Review:**

Strength: It is certainly exciting to see the relationship between quantum learning class, and as far as I know, this is the first paper that intends to show such equivalence between quantum learning class (although the classical counterpart has been well-studied) The proof is also non-trivial.

Drawback: From my view, the major unsatisfactory of this paper is that the implication (from online learnability) stops at the level of quantum stability, but the major breakthrough of Bun, Livni and Moran (Journal of the ACM'21) actually takes a step further and showing (classical) stability implies private PAC learning. Though the later step is not the technical core of BLM'21 and this paper proves such implications do not hold in quantum regime, it weakens the conceptual contribution of this paper, as it is not clear afterwards why quantum stability is interesting.

Summary: Overall, I think this is a nice paper, but I hope the author could response to my comment above (drawback), and it might be good to put this into discussion or open questions.


I have read the author's response and my evaluation stays the same.

**Time Spent Reviewing:**

1

---

> ### Author Response · Authors · 2021-08-09
> **Reply to the drawback comment of the review**
>
>
> Thank you for your review, we appreciate your thoughtful comments! Here we address your question:
>
> Although, we agree with your comment that it is unfortunate that the complete chain of implications do not go through for real-valued functions, we view our counterexample for this as a key contribution of our paper since it points out the unexpected fact that it is not possible to straightforwardly (i.e. extending the method of Bun et al to the real-valued case) derive the implication that quantum stability implies approximately private PAC learning.
>
>  Independently, we suspect that quantum stability (and our notion of real-valued stability) may be interesting because classically this notion appears in many algorithms, e.g., follow-the-perturbed-leader and follow-the-regularized-leader. Also our real-valued/quantum stability has a very conceptually appealing interpretation of an algorithm’s output always being in some epsilon ball of a function/state, which naturally allows for robustness in the algorithm. Overall, we hope that our definition of stability contributes to the still-evolving discussion of what it means to be differentially private in the real-valued/quantum setting, given the close connections between the two in the classical learning setting.

---

### Official Review · Reviewer_XTWD · 2021-07-19

**Rating:** 8
**Confidence:** 3

**Summary:**

This paper derives relationships between several models of approximately learning quantum states. Specifically, it extends results in learning boolean functions to learning real-valued functions with imprecise feedback, and shows that the complexity of learning in Pure DP PAC upper bounds quantum CC and the sequential fat-shattering dimension. The Sfat dimension in turn implies online learnability. A conceptual contribution of this paper is quantum stability, and the authors show that online learnability implies the existence of stable learners, but contrary to results on boolean functions, quantum stability does not imply Approximate DP PAC. The authors also propose the RSOA algorithm, which is central to proving the results.

**Limitations And Societal Impact:**

Yes.

**Main Review:**

The paper makes solid technical contributions towards addressing the motivating question of relating different learning models. The learning models are designed with learning quantum states in mind and extend previous settings on exact feedback to imprecise feedback. The results answer an important question and make progress in understanding the complexity of learning quantum states. Overall the paper is well-written, and the following minor points might improve the clarity of the paper: 1) can the authors comment on the computational complexity of the RSOA algorithm? 2) does the improved bound on sfat() use techniques in the previous sections of the paper? It’s not clear to me how this is an application of the previous results, but it is a well-motivated task.

Post rebuttal: I have read the authors' response and I intend to keep my score. I agree with reviewer C1db's point on clarifying the reduction, but overall I think this paper is above the acceptance threshold.

**Time Spent Reviewing:**

4

---

> ### Author Response · Authors · 2021-08-09
> **Reply to questions raised by reviewer**
>
> Thanks for your time, we appreciate your thoughtful comments! Here we address your questions:
>
> 1) Indeed our RSOA is not efficient since it requires the computation of the sfat dimension of each super-bin. In fact this is also true for the well-known Standard Optimal Algorithm since it requires computing the Littlestone dimension which is not known to be computable in polynomial time: https://arxiv.org/pdf/1705.09517.pdf. In contrast, the main selling point of our work is, information-theoretically many models and parameters can be related and we leave it as open to prove these implications time-efficiently.
>
> 2) The improved bound on sfat is independent of our equivalences statement (in terms of techniques). We simply use that sfat characterizes many models of learning quantum states (as proved in the main body). Acknowledging that it is difficult to compute sfat of a class of quantum states exactly, we provide a way to upper-bound it. Thus we can use our upper-bound on sfat to upper-bound the complexities of those quantum learning tasks.

---

### Official Review · Reviewer_fFb9 · 2021-07-24

**Rating:** 8
**Confidence:** 4

**Summary:**

This paper is motivated by the problem of learning n-qubit quantum states in different learning models, such as differentially private PAC learning and online learning with imprecise feedback, and proves information-theoretic equivalences and implications between them. One of the main results of the paper is proving that pure DP PAC learning of quantum states implies learning of quantum states in the online learning model with imprecise feedback.

The line of implications goes as follows: a bound on the sample complexity of Pure DP PAC learning implies a bound on the probabilistic representation dimension of the class, which implies a bound on the one-way communication complexity of quantum learning, which bounds the *sequential-fat-shattering* dimension of the class (a generalization of the Littlestone dimension from boolean to real-valued function classes).
If the sequential-fat-shattering dimension is bounded then there exists an online learning algorithm which is robust to imprecise feedback (as must be the case for the training samples-measurements of quantum states), called Robust-SOA.

The authors then also prove that this online learner implies the existence of a learning algorithm that satisfies a type of stability, called quantum stability, which says that there exists a state such that the algorithm’s output is within a ball of this state with high probability.

Among others, a significant implication included in the results is a new bound on the complexity of shadow tomography which includes the sequential-fat-shattering dimension (sfat) instead of the number of qubits n (and sfat<=n for any class but there exist classes with sfat significantly smaller than n).


**Limitations And Societal Impact:**

Sufficiently discussed.

**Main Review:**

The reasoning behind the “pure DP PAC learning->online learning of quantum states->stability “ result follows a recent line of papers which prove that for the case of classification (Boolean function classes), DP PAC learning implies online learning which implies a type of stability, called global stability, which in turn, implies DP PAC learning, with the main quantity in these implications being the Littlestone dimension of the class. However, learning of quantum states seems to come with additional difficulties, since the classes are real-valued and the measurements are noisy (which leads to the online learning with imprecise feedback model and to the proposed “quantum stability” being an approximate version of global stability). Getting over these challenges seems to require many new (intermediate) results and connections, which could also be of independent interest, especially given that the paper considers many different types of learning models and combinatorial quantities, not necessarily tied to quantum learning.

=============
I have read the authors' responses and still think that overall this paper's results merit acceptance. I also found the paper a bit hard to read in places but cannot come up with concrete suggestions, since the issue seems to be the breadth of results and the effort to compile them together is clear. I do second adding clarifications to the reduction as per Reviewer C1db's comments. Finally, (again, not having changed my opinion on the paper) I think it is beneficial to refer to the recent paper "Differentially Private Nonparametric Regression Under a Growth Condition" by Noah Golowich which appeared at COLT 2021 : it studies private learnability in the real-valued domain and gives a relaxed sufficient condition, compared to Jung et al, which includes the eta-sequential-fat-shattering dimension.

**Time Spent Reviewing:**

8

---

### Decision · Program_Chairs · 2021-09-27

**Decision:**

Accept (Spotlight)

**Comment:**

Bun, Livni and Moran recently proved that finite littlestone dimension implies private (approximate) PAC learning of a Boolean concept class. This paper generalizes it to 1) learning real valued functions instead of Boolean valued functions and 2) to learning quantum states. In the process, the paper discovers several interesting relationships between different models of learning and points out an interesting "no-go" theorem for relating a (natural notion of ) quantum stability to online learnability.

The paper makes fundamental connections between learning models: both positive (by translating algorithmic results) and negative (by exhibiting counter-examples for what might be natural conjectured relationships). The reviewers had a high opinion of the paper and authors' response and the ensuing discussion clarified what makes the no-go theorem between quantum stability and private learning non-trivial.   We recommend acceptance.